# DEBATE: A LARGE-SCALE BENCHMARK FOR EVALUATING OPINION DYNAMICS IN ROLE-PLAYING LLM AGENTS

## ABSTRACT

Accurately modeling opinion change through social interactions is crucial for understanding and mitigating polarization, misinformation, and societal conflict. Recent work explores simulating *opinion dynamics* with role-playing LLM agents (RPLAs)—language models assigned human-like personas that engage in multi-turn, multi-agent opinion exchange. However, existing RPLA simulations often produce unnatural group behaviors (e.g., premature consensus) and lack empirical benchmarks for evaluating alignment with real human interactions. We introduce **DEBATE**, the first large-scale benchmark for evaluating the authenticity of opinion dynamics in multi-agent RPLA simulations. DEBATE contains 37,357 messages from 2,792 U.S.-based participants who engaged in multi-player, multi-round conversations across 107 controversial topics, reporting both public messages and private beliefs. We simulate these conversations using various LLMs and introduce multi-level evaluation metrics (at the utterance, individual, and group levels) to assess behavioral alignment between humans and RPLAs. Our analyses reveal key behavioral gaps: RPLA groups exhibit stronger opinion convergence and belief drift than humans, and individual agents show more systematic shifts in response to social influence. Ablation studies further highlight the importance of private self-reported opinions in shaping realistic agent behavior. Additionally, while supervised fine-tuning improves surface-level metrics (e.g., ROUGE-L, message length), it falls short on deeper alignment (e.g., semantic and stance alignment). DEBATE enables benchmarking of simulated opinion dynamics and supports future research on aligning multi-agent RPLAs' simulations with realistic human interactions. The dataset and codebase will be publicly released.

## 1 INTRODUCTION

Understanding how individual opinions change through social interactions is crucial across numerous domains, e.g., public health campaigns, conflict resolution, and misinformation mitigation (Lu et al., 2015; Pennycook et al., 2021; Budak et al., 2011; Loomba et al., 2021; Ginossar et al., 2022). Accurate modeling of these *opinion dynamics* not only helps predict critical societal phenomena like opinion polarization but also informs effective interventions to mitigate adverse outcomes.

Recent advances in large language models (LLMs) have unlocked new possibilities for simulating human social interactions, particularly through the use of role-playing LLM agents (RPLAs) that embody diverse personas and engage in multi-turn dialogue (Park et al., 2023; Chuang et al., 2024a;b). Although individual RPLAs can often convincingly emulate human-like behaviors, prior research indicates that this single-agent authenticity does not guarantee realistic emergent dynamics in multi-agent settings. Specifically, when multiple RPLAs interact, they frequently exhibit premature consensus convergence, overly moderate stances, or unnatural patterns of opinion alignment, regardless of their initial diverse personas (Chuang et al., 2024a; Taubenfeld et al., 2024). Existing evaluations of RPLAs predominantly focus on single-agent scenarios or employ artificial, structured tasks, lacking robust empirical benchmarks capturing authentic human group dynamics in naturalistic contexts (Santurkar et al., 2023; Chuang et al., 2024c;b).

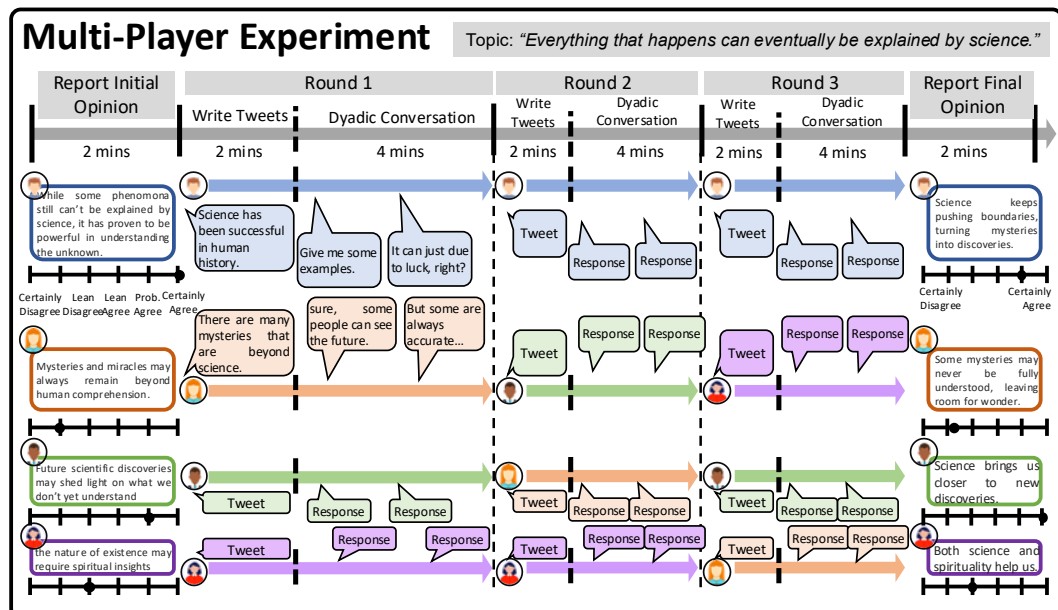

Figure 1: The procedure of the multi-player experiment. Each group is assigned a topic to discuss about. Participants first report their initial opinion, then engage in three rounds of tweet writing and dyadic conversations with different partners, and finally submit their final opinion. With this setup, we collects naturalist opinion exchanges among groups.

To address this critical gap, we introduce **D**eliberative Opinion **E**xchanges for **B**enchmarking **A**gent-based **T**rajectory **E**volution (**DEBATE**), the first large-scale empirical benchmark specifically designed for evaluating the authenticity of simulated opinion dynamics from multi-agent RPLAs. While the acronym suggests debate, DEBATE emphasizes naturalistic deliberation rather than competitive or adversarial argumentation. DEBATE comprises data with 37,357 messages from 824 groups and 2,792 U.S.-based participants engaged in multi-round, multi-party discussions on 107 controversial topics. It captures both publicly expressed *messages* (including both tweet-like posts and chat utterances) and privately reported beliefs (Likert-scale ratings). Due to occasional dropouts, out of all participants, 725 groups completed (2,584 participants) all sessions end-to-end and engaged in each experiment phase, yielding a clean subset of 28,579 messages used for benchmarking and evaluation (average messages per group is 39.4). The benchmark enables quantitative assessment of alignment between simulated and actual human interactions at the *utterance*, *individual*, and *group* levels.

**Contributions.** (1) We introduce **DEBATE**, the first large-scale empirical benchmark for evaluating the human-likeness of opinion dynamics in multi-agent role-playing LLM agents (RPLAs). (2) DEBATE supports three simulation setups: *Next Message Prediction*, *Tweet-Guided Simulation*, and *Full Conversation Simulation*, covering various scenarios in social simulation. (3) We design quantitative evaluation metrics at three different levels (utterance, individual, and group) to assess different aspects of alignment between simulated and human opinion trajectories. (4) Supervised fine-tuning (SFT) improves surface-level text quality but not deeper stance or belief alignment, highlighting the need for future work on training RPLAs. (5) We identify key behavioral gaps between RPLAs and humans, including stronger opinion convergence and positive belief drift, showcasing the challenges of simulating realistic opinion dynamics with RPLAs. Our evaluation, fine-tunine, and analyses are intended as examples of how the benchmark can be used to evaluate human-RPLA alignment, and we want to note that the dataset itself is the primary contribution. We expect DEBATE to enable future studies on RPLA opinion dynamics simulations. The dataset will be released publicly upon acceptance. The codebase and a portion of the data is included in the supplementary materials (Appendix A).[1]

---

[1]LLM usage disclosure: ChatGPT was used only for language polishing; see Appendix Q.

Table 1: Comparison of DEBATE with existing human opinion dynamics datasets. We categorize datasets into three genres: *competitive debate*, *asymmetric persuasion*, and *naturalistic deliberation*. Columns indicate whether each dataset supports multi-party interactions, multi-turn conversations, demographic attributes, reader-context traceability, enforced turn-taking, public/private opinions, data source and participants.

| Dataset | Opinion Dynamics Type | Multi-Party (N≥3) | Multi-Turn | Demo-graphics | Reader-Context Traceable | Turn Obliga-tion | Public or Private Opinion | Data Source [Participants] | # Utterances (U) # Conversations (C) # Subjects (S) |
|---|---|---|---|---|---|---|---|---|---|
| CMV (Tan et al., 2016) | Competitive Debate | ✓ | ✓ | ✗ | ✗ | ✗ | Public | Reddit threads [Reddit users] | U = 293,297 C = 3,051 S = 34,911 |
| IAC 1.0 (Walker et al., 2012) | Competitive Debate | ✓ | ✓ | ✗ | ✗ | ✗ | Public | Online debate forums [Online forum users] | U = 390,704 C = 11,800 S = 3,300 |
| IAC 2.0 (Abbott et al., 2016) | Competitive Debate | ✓ | ✓ | ✗ | ✗ | ✗ | Public | Online debate forums [Online forum users] | U = 482,000 C = 16,461 S = 9,709 |
| UK Parliament QuestionTime Corpus (Zhang et al., 2017) | Competitive Debate | ✓ | ✓ | ✗ | ✓ | ✓ | Public | British House of Commons [Members of Parliament] | U = 433,787 C = 216,894 S = 1,978 |
| Intelligence Squared Debates Corpus (Zhang et al., 2016) | Competitive Debate | ✓ | ✓ | ✗ | ✓ | ✓ | Public | Structured debate show [Invited experts] | U = 26,562 C = 108 S = 471 |
| PersuasionForGood (Wang et al., 2019) | Persuasion (Asymmetric) | ✗ | ✓ | ✓ | ✓ | ✓ | Public | MTurk platform [Diverse laypeople] | U = 20,932 C = 1,017 S = 1,285 |
| Wikipedia Articles-for-Deletion (AfD) (Mayfield and Black, 2019) | Deliberation (Policy) | ✓ | ✓ | ✗ | ✗ | ✗ | Public | Wikipedia editor debates [Wikipedia editors] | U = 3,295,340 C = 383,918 S = 161,266 |
| **DEBATE** | Deliberation (Naturalistic) | ✓ | ✓ | ✓ | ✓ | ✓ | Public + Private | Prolific platform [Diverse laypeople] | U = 28,579 C = 4,350 S = 2,584 |

## 2 RELATED WORK

**Simulating and Evaluating Opinion Dynamics with RPLAs.** Opinion dynamics refers to how individuals form, change, and negotiate beliefs through social interaction (Flache et al., 2017; Lorenz et al., 2021; Chuang and Rogers, 2023). Recent work has explored multi-agent LLM opinion interactions as a means to enhance downstream task performance, e.g., improving factuality, reasoning accuracy, and output diversity (Zhang et al., 2023; Chan et al., 2023; Chen et al., 2023; Du et al., 2023; Liang et al., 2023; Hu et al., 2024). However, these approaches primarily treat opinion exchange as a technique for boosting task performance, rather than aiming to simulate human-like opinion evolution

On the other other hand, a growing body of research instead focuses on simulating *human-like* opinion dynamics using RPLAs (Chuang et al., 2024a; Taubenfeld et al., 2024; Liu et al., 2024). These studies assign personas to agents and allow them to interact over multiple turns, aiming to model human-like opinion formation and change. However, most rely on qualitative observations in toy settings and lack empirical benchmarks for evaluation against real human behavior.

Recent efforts to quantify human-likeness of RPLAs' simulated opinion either on single-agent settings without interaction (Santurkar et al., 2023; Chuang et al., 2024c), or on non-linguistic tasks (Chuang et al., 2024b). In contrast, DEBATE introduces the first large-scale benchmark explicitly designed for evaluating multi-agent RPLAs in simulating natural-language opinion dynamics.

**Existing Opinion Dynamics Corpora.** Although no existing corpora were originally constructed as benchmarks for evaluating human-like opinion simulations of RPLAs, several contain human interactions involving opinion and have the potential to be adapted for this purpose. These can be grouped into three genres (Table 1; Walton and Krabbe, 1995; Walton et al., 2010; Bozdag et al., 2025): (1) *Competitive debate*, where participants aim to win an argument (e.g., CMV (Tan et al., 2016), IAC (Walker et al., 2012; Abbott et al., 2016)); (2) *Asymmetric persuasion*, where one party aims to influence another (e.g., PersuasionForGood (Wang et al., 2019)); and (3) *Naturalistic deliberation*, where peers voluntarily share and refine beliefs without roles or external incentives. This setting most closely relates to everyday social interactions. However, despite its relevance to real-world discourse, this deliberative genre is underrepresented in existing corpora.

Beyond genre coverage, most corpora lack key features for benchmarking human-like opinion dynamics. Many omit full reader-context traceability (e.g., CMV, IAC), making it difficult to reconstruct what input each speaker saw at the time of writing. Most also lack enforced *turn obligation*, which is critical for yielding observable belief trajectories for opinion dynamics modeling

Table 2: **Dataset statistics.** Each row reports the number of topics, messages, on-topic messages, subjects, groups, and the average number of groups per topic. Depth topics have more groups per topic, while breadth topics span a wider range of themes with fewer groups per topic.

| Dataset | # topics | # messages | # on-topic messages | # subjects | # groups | # groups/topic |
|---|---|---|---|---|---|---|
| Depth | 7 | 5,252 | 4,510 | 479 | 144 | 20.57 |
| Breadth | 100 | 23,327 | 21,538 | 2,105 | 581 | 5.81 |
| Depth+Breadth | 107 | 28,579 | 26,048 | 2,584 | 725 | 6.78 |

(Flache et al., 2017) (in social media corpora, very few people dominate while most people stay silent; Van Mierlo, 2014). Importantly, no existing corpus combines both *publicly expressed* messages and *privately reported* beliefs. Measuring private self-report beliefs is important because individuals may publicly express socially desirable position that are different from their actual beliefs (Tourangeau and Yan, 2007). Demographic data is also often incomplete, limiting the construction of realistic personas for RPLAs. Finally, while some corpora involve selected experts (e.g., UK Parliament), we believe simulating belief change in *diverse laypeople* is more relevant for real-world applications.

**Positioning DEBATE as a Benchmark.** To address these gaps, DEBATE introduces not just a new dataset, but a full *evaluation benchmark* for multi-agent RPLAs' opinion dynamics simulation. It features multi-round conversations among participants with diverse backgrounds discussing controversial topics, with both public and private opinions, full reader context, enforced turn-taking, and rich demographics. Furthermore, it includes a suite of quantitative metrics designed to evaluate how closely simulated opinion trajectories match real human dynamics. This makes DEBATE a dedicated benchmark for evaluating the fidelity of RPLAs' opinion dynamics simulation.

## 3 DEBATE BENCHMARK: EMPIRICAL OPINION DYNAMICS FROM HUMAN

### 3.1 TASK

We design a multi-player conversational experiment to elicit naturalistic opinion dynamics (Figure 1). The dataset comprises $G$ groups, each consisting of $N = 4$ participants $\{s_1, s_2, s_3, s_4\}$. Each group is randomly assigned a single controversial discussion topic $t \in \mathcal{T}$ throughout the session. The experiment lasts 25–30 minutes per group and consists of four phases. See Appendix T for the user interface and Appendix P for sample conversation data.

**(1) Initial Private Opinion** (2-minute): Each participant $s_i$ reports an initial opinion $o_{s_i}^{\text{init}} \in \{-2.5, -1.5 \ldots, +2.5\}$ on a 6-point Likert scale[2], along with a text justification $j_{s_i}^{\text{init}}$. They are submitted privately on a separate webpage so that no other members can view their responses.

**(2) Public Opinion Exchanges** (6-minute): Participants engage in $R = 3$ rounds of dyadic conversation. [3] In each round $r$, participants are randomly paired with one of the other group members who they haven't interacted with yet. Across three rounds, each participant interacts with every other group member exactly once. For each pair of distinct participants $(s_i, s_j)$, where $i \neq j$:

- Each participant first writes a tweet-like post $\tau_{s_i}^r$ within 2 minutes, summarizing their opinion on the assigned topic.

- After submitting their tweets, participants view each other's post and engage in a 4-minute real-time conversation via a chatbox interface. The conversation is represented as an ordered sequence: $\mathcal{C}_{s_i,s_j}^r = \left[ u_{1,s_i}^r, u_{2,s_j}^r, u_{3,s_i}^r, \ldots \right]$, where $u_{k,s}^r$ denotes the $k$-th utterance in the round-$r$ conversation, with speaker $s \in \{s_i, s_j\}$. Speaker turns alternate between participants. Consecutive messages from the same speaker are merged during data preprocessing.

**(3) Post-discussion Private Opinion** (2-minute): After the final round, each participant privately submits a final opinion $o_{s_i}^{\text{final}}$ and justification $j_{s_i}^{\text{final}}$ on a separate webpage similar to the initial opinion.

---

[2]Participants selected from the six labels displayed in the interface: $(-2.5)$ *Certainly disagree*, $(-1.5)$ *Probably disagree*, $(-0.5)$ *Lean disagree*, $(+0.5)$ *Lean agree*, $(+1.5)$ *Probably agree*, $(+2.5)$ *Certainly agree*.

[3]Following standard setups in opinion dynamics simulations, we use *dyadic interactions* between each pair, which allows tracing of each individual's opinion trajectory.

**(4) Demographic Survey**: Finally, participants report demographic attributes $d_{s_i}$ (e.g., age, gender, education, political orientation), with no time limit.

## 3.2 TOPICS

The DEBATE benchmark includes two complementary topic sets: *Depth* and *Breadth*.

**Depth Topics** ($\mathcal{T}_{\text{Depth}}$) comprises seven topics selected from a prior study, each tied to a known scientific consensus or "ground truth." An example is: *"The position of the planets at the time of your birth can influence your personality."* Prior work shows that RPLAs often drift toward ground-truth views over time, regardless of initial opinions (Chuang et al., 2024a; Taubenfeld et al., 2024). We selected seven such topics possessing high entropy across individuals in order to elicit diverse opinions from humans (Chuang et al., 2024c). Each topic was assigned to an average of 20.57 groups (479 participants in total; Table 2), allowing us to evaluate how systematically individual groups behave when discussing the same topic. See Appendix B for the full list.

**Breadth Topics** ($\mathcal{T}_{\text{Breadth}}$) contains 100 topics from the World Values Survey (WVS) (Haerpfer et al., 2022) and Pew Global Attitudes Survey (PGAS) (Pew Research Center, 2025). To reflect public disagreement in our US-based participants, we selected U.S.-administered questions with the highest response entropy (Durmus et al., 2024). Topics were phrased as self-contained declarative statements (e.g., *"Euthanasia can be justified."*) and spanned domains such as science, policy, and social values. These topics are not linked to ground truths but reflect a wide range of viewpoints. On average, each topic was assigned to 6.78 groups (2,105 participants in total; Table 2). See Appendix C for details.

## 3.3 HUMAN DATA COLLECTION AND DATASET SUMMARY

We recruited 2,792 unique participants who reside in the U.S. via the Prolific platform (Palan and Schitter, 2018)[4]. Participants were randomly assigned to one of 824 four-person groups and to a discussion topic. They remained anonymous to each other, identified only by randomly-assigned avatars and pseudonyms (e.g., `ZK48UT`). All procedures were approved by the Institutional Review Board (IRB). Participants were compensated at a rate of $10/hour. Of the 824 groups recruited for DEBATE, some experienced participant dropouts or technical interruptions, resulting in partially completed sessions. For evaluation and analysis, we use a clean subset of 725 fully completed groups (2,584 participants, 28,579 utterances) who have engaged in each experiment phase. However, we also release the full raw dataset (2,792 participants, 37,357 utterances), including incomplete sessions, as it may support other research use cases. Details on filtering criteria and raw data are provided in Appendix D, and we analyze participant engagement and benchmark reliability in Appendix R.

The participants spanned a broad range of ages (18–83, $M = 39.5$, $SD = 13.0$), genders (50.2% male, 49.0% female), ethnicities (e.g., 66.4% White, 24.7% Black, 5.5% Asian, 5.1% Hispanic), educational backgrounds (ranging from high school to doctoral degrees), and income levels (from under $25k to over $200k). Participants also reported a wide variety of occupations (e.g., finance, engineering, healthcare, manufacturing). This diversity provides a robust foundation for modeling opinion dynamics across varied social perspectives (see Appendix E and Figure 4 for details).

## 4 CONSTRUCTING AND EVALUATING ROLE-PLAYING LLM AGENTS

### 4.1 RPLA CONSTRUCTION GROUNDED IN HUMAN DATA

Each RPLA $a_i$ is designed as a *digital twin* of a human participant $s_i$, simulating $s_i$'s conversational behavior throughout the multi-round interaction. Each $a_i$ is conditioned on a memory module $\mathcal{M}_{a_i,k}$ that aims to reflect $s_i$'s first-person perspective right before producing the $k$-th utterance in round $r$. The memory is dynamically updated as tweets and utterances are exchanged.

The memory module $\mathcal{M}_{a_i,k}$ is instantiated via prompt templates that convert structured information into natural language inputs for the LLM (see Appendix F and Table 7 for prompt examples). We use notation with a hat and subscript $a$ (e.g., $\widehat{\tau}_{a_i}^r$, $\widehat{u}_{k,a_i}^r$) to denote LLM-generated content, and notation without a hat and with subscript $s$ (e.g., $\tau_{s_i}^r$, $u_{k,s_i}^r$) to denote human-written content.

---

[4] https://www.prolific.com/

At each turn $k$ in round $r$, the agent memory $\mathcal{M}_{a_i,k}$ includes: **1. Demographic Profile** ($d_{s_i}$)**:** Age, gender, education, income, ethnicity, marital status, residence, parental status, political ideology, religiosity, and occupation. **2. Initial Opinion** ($o_{s_i}^{\text{init}}, j_{s_i}^{\text{init}}$)**:** A 6-point Likert-scale opinion on the assigned discussion topic and a free-text justification. **3. Initial Tweet** ($\tau_{s_i}^1$)**:** The tweet posted at the beginning of round 1. **4. Previous Rounds:** Tweets $\{\tau_{s_i}^{r'}, \widehat{\tau}_{a_i}^{r'} : 1 < r' < r\}$ and dyadic conversations $\{\mathcal{C}_{s_i,s_j}^{r'}, \widehat{\mathcal{C}}_{a_i,a_j}^{r'} : 1 \leq r' < r\}$ from earlier rounds involving participant $s_i$. **5. Current Round Context:** The current tweet $\tau_{s_i}^r$ or $\widehat{\tau}_{a_i}^r$, the partner's tweet $\tau_{s_j}^r$ or $\widehat{\tau}_{a_j}^r$, and all utterances so far in the ongoing conversation $\{u_{k',s}^r, \widehat{u}_{k',a}^r : 1 \leq k' < k\}$. The exact sources of memory vary by the simulation mode (Section 4.1; Table 3). For example, the conversation history may come from real human (Mode 1), LLM simulation (Mode 3), or a mix of both (Mode 2).

## 4.2 SIMULATING SOCIAL INTERACTIONS WITH RPLAs

Table 3: Memory contents used in each simulation mode. All agents are conditioned on demographics $d_{s_i}$, initial opinion and justification ($o_{s_i}^{\text{init}}, j_{s_i}^{\text{init}}$), the initial tweet $\tau_{s_i}^1$, and task instructions. Blue entries indicate simulated content recursively generated by the model and added to the memory.

| Simulation Mode | Tweets in Memory | Utterances from Prior Rounds | Utterances from Current Round | Application and Scenario |
|---|---|---|---|---|
| **Mode 1:** Next Message Prediction | Human $\{\tau^{r'} : 1 \leq r' \leq r\}$ | Human $\{\mathcal{C}_{s_i,s_j}^{r'} : 1 \leq r' < r\}$ | Human $\{u_m^r : 1 \leq m < k\}$ | Predict a person's immediate response in real conversations |
| **Mode 2:** Tweet-guided Conversation Simulation | Human $\{\tau^{r'} : 1 \leq r' \leq r\}$ | Simulated $\{\widehat{\mathcal{C}}_{a_i,a_j}^{r'} : 1 \leq r' < r\}$ | Simulated $\{\widehat{u}_m^r : 1 \leq m < k\}$ | Simulate private conversations given a trace of real public tweets |
| **Mode 3:** Full Conversation Simulation | Human $\tau^1$ + Simulated $\{\widehat{\tau}^{r'} : 2 \leq r' \leq r\}$ | Simulated $\{\widehat{\mathcal{C}}_{a_i,a_j}^{r'} : 1 \leq r' < r\}$ | Simulated $\{\widehat{u}_m^r : 1 \leq m < k\}$ | Simulate agents' dynamics from initial conditions; the **classic opinion dynamics simulation** setup |

We simulated each RPLA $a_i$'s utterance $\widehat{u}_{k,a_i}^r$ in round $r$, turn $k$ by generating:

$$\widehat{u}_{k,a_i}^r \sim P\left(u_{k,s_i}^r \mid \mathcal{M}_{a_i,k}\right), \tag{1}$$

where the speaker identity $s_i$ is given, and only the utterance content is predicted.[5] The same framework applies to generating tweets $\widehat{\tau}_{a_i}^r$ and final opinions ($\widehat{o}_{a_i}^{\text{final}}, \widehat{j}_{a_i}^{\text{final}}$).

The DEBATE benchmark provides infrastructure for three simulation modes, corresponding to three common scenarios for simulation of social interactions: *Next Message Prediction* (Mode 1), *Tweet-guided Conversation Simulation* (Mode 2), and *Full Conversation Simulation* (Mode 3). All three are grounded in real human behavior but vary in how much human context is provided to the model. This setup allows researchers to study different aspects of multi-agent communication, from immediate message prediction to end-to-end full trajectory generation from initial state. Each simulation is conditioned on the memory module $\mathcal{M}_{a_i,k}$, which includes basic information such as demographics $d_{s_i}$, initial opinion and justification ($o_{s_i}^{\text{init}}, j_{s_i}^{\text{init}}$), the initial tweet $\tau_{s_i}^1$, and task instructions (Section 4.1). What varies across simulation modes is the source of tweets and conversational history: whether they come from real human data or are recursively generated and added to the memory. Table 3 summarizes the full memory configuration and the corresponding use case for each simulation mode.

We evaluate RPLA simulations using six different LLMs: `gpt-4o-mini-2024-07-18` (OpenAI, 2022), `Llama-3.1-Tulu-3-8B-SFT` (Lambert et al., 2024), `Llama-3.1-8B-Instruct` (Dubey et al., 2024), `Llama-3.1-70B-Instruct`, `Mistral-7B-Instruct-v0.3` (Jiang et al., 2023), and `Qwen2.5-32B-Instruct` (Bai et al., 2023). These models span open vs. proprietary weights, varying parameter scales, and both pre-alignment and post-alignment checkpoints. Full compute details are in Appendix I.

## 4.3 EVALUATION

We evaluated how well an RPLA $a_i$ simulates its corresponding human participant $s_i$ by comparing utterances $\widehat{u}$ and $u$ within the dyadic conversations, focusing only on *on-topic* utterances—-those

---

[5]Since consecutive messages from the same speaker are merged during preprocessing, speakers alternate turns, making the speaker order known (Section 3.1).

Table 4: Evaluation results across simulation modes and LLMs. We report the round-wise aggregated metrics on the **Depth Topics**: average semantic similarity $\overline{S}_{\text{sem}}$ ($\uparrow$), average stance difference $\overline{\Delta}_{\text{stance}}$ ($\downarrow$), average signed length difference $\overline{\Delta}_{\text{signed\_len}}$ ($\to 0$), average absolute length difference $\overline{\Delta}_{\text{abs\_len}}$ ($\downarrow$), $\overline{\text{ROUGE-L}}$ ($\uparrow$), and on-topic utterance rate $R_{\text{on-topic}}$. Error bars indicate standard error from 1,000 bootstrap resamples.

| LLM & Simulation Mode | $\overline{S}_{\text{sem}}$ ($\uparrow$) | $\overline{\Delta}_{\text{stance}}$ ($\downarrow$) | $\overline{\Delta}_{\text{signed\_len}}$ ($\to 0$) | $\overline{\Delta}_{\text{abs\_len}}$ ($\downarrow$) | $\overline{\text{ROUGE-L}}$ ($\uparrow$) | $R_{\text{on-topic}}$ |
|---|---|---|---|---|---|---|
| *Simulation Mode 1: Next Message Prediction* | | | | | | |
| gpt-4o-mini-2024-07-18 | **0.48 ± 0.01** | 1.16 ± 0.05 | -32.72 ± 0.62 | 33.51 ± 0.60 | **0.11 ± 0.01** | 0.74 |
| Llama-3.1-Tulu-3-8B-SFT | 0.44 ± 0.01 | 1.19 ± 0.06 | -41.57 ± 1.28 | 45.07 ± 0.93 | 0.06 ± 0.01 | 0.56 |
| Llama-3.1-8B-Instruct | 0.45 ± 0.01 | 1.21 ± 0.04 | -36.85 ± 0.87 | 38.37 ± 0.76 | 0.07 ± 0.01 | 0.73 |
| Llama-3.1-70B-Instruct | 0.45 ± 0.01 | **1.15 ± 0.05** | -26.19 ± 1.05 | 28.88 ± 0.86 | 0.08 ± 0.01 | 0.78 |
| Mistral-7B-Instruct-v0.3 | 0.47 ± 0.01 | 1.18 ± 0.05 | -46.27 ± 0.71 | 46.79 ± 0.67 | 0.07 ± 0.01 | 0.72 |
| Qwen2.5-32B-Instruct | 0.46 ± 0.01 | 1.16 ± 0.05 | **-22.40 ± 0.82** | **27.12 ± 0.64** | 0.08 ± 0.01 | 0.73 |
| *Simulation Mode 2: Tweet-guided Conversation Simulation* | | | | | | |
| gpt-4o-mini-2024-07-18 | **0.42 ± 0.01** | 1.25 ± 0.05 | -58.40 ± 0.78 | 58.56 ± 0.76 | **0.09 ± 0.01** | 0.66 |
| Llama-3.1-Tulu-3-8B-SFT | 0.41 ± 0.01 | 1.34 ± 0.07 | -53.66 ± 0.88 | 54.38 ± 0.82 | 0.05 ± 0.01 | 0.48 |
| Llama-3.1-8B-Instruct | 0.41 ± 0.01 | 1.28 ± 0.05 | -52.81 ± 0.93 | 53.31 ± 0.86 | 0.06 ± 0.01 | 0.67 |
| Llama-3.1-70B-Instruct | 0.40 ± 0.01 | **1.18 ± 0.05** | -51.24 ± 1.24 | 51.99 ± 1.17 | 0.06 ± 0.01 | 0.72 |
| Mistral-7B-Instruct-v0.3 | 0.41 ± 0.01 | 1.21 ± 0.06 | **-46.77 ± 0.76** | **47.26 ± 0.70** | 0.06 ± 0.01 | 0.63 |
| Qwen2.5-32B-Instruct | 0.41 ± 0.01 | 1.25 ± 0.06 | -47.84 ± 1.31 | 49.79 ± 1.10 | 0.07 ± 0.01 | 0.66 |
| *Simulation Mode 3: Full Conversation Simulation* | | | | | | |
| gpt-4o-mini-2024-07-18 | **0.41 ± 0.01** | 1.30 ± 0.05 | -58.11 ± 0.73 | 58.26 ± 0.71 | **0.08 ± 0.01** | 0.65 |
| Llama-3.1-Tulu-3-8B-SFT | 0.40 ± 0.01 | 1.46 ± 0.07 | -55.00 ± 0.92 | 55.84 ± 0.82 | 0.05 ± 0.01 | 0.46 |
| Llama-3.1-8B-Instruct | 0.39 ± 0.01 | 1.33 ± 0.05 | -52.84 ± 0.91 | 53.39 ± 0.85 | 0.06 ± 0.01 | 0.67 |
| Llama-3.1-70B-Instruct | 0.38 ± 0.01 | 1.27 ± 0.05 | -51.14 ± 1.12 | 52.03 ± 1.01 | 0.06 ± 0.01 | 0.72 |
| Mistral-7B-Instruct-v0.3 | 0.40 ± 0.01 | **1.25 ± 0.05** | **-46.69 ± 0.80** | **47.25 ± 0.73** | 0.06 ± 0.01 | 0.61 |
| Qwen2.5-32B-Instruct | 0.40 ± 0.01 | 1.30 ± 0.05 | -49.22 ± 1.11 | 50.83 ± 0.96 | 0.07 ± 0.01 | 0.65 |

directly addressing the discussion topic $t$—excluding conversational fillers (e.g., *"hello"*, *"what do you think?"*) or unrelated remarks (e.g., *"which football team do you support?"*). From these we assessed different aspects of human-model alignment with the following metrics: **1. Semantic Similarity:** $S_{\text{sem}}(u, \widehat{u}) = \cos(E(u), E(\widehat{u}))$, where $E(\cdot)$ is a sentence encoder (Sturua et al., 2024) . This measures the meaning-level similarity between utterances, capturing whether the agent expresses a semantically similar idea. **2. Stance Difference:** $\Delta_{\text{stance}}(u, \widehat{u}) = |S(u) - S(\widehat{u})|$, using scalar stance scores in $[-2.5, -1.5, -0.5, +0.5, +1.5, +2.5]$. This captures alignment in opinion polarity, assessing whether the agent expresses a similar stance. **3. Length Metrics:** $\Delta_{\text{abs\_len}} = \big||u| - |\widehat{u}|\big|$; $\Delta_{\text{signed\_len}} = |u| - |\widehat{u}|$. These reflect surface-level stylistic similarity in verbosity and message length. **4. ROUGE-L:** Longest common subsequence score (Lin, 2004). This quantifies token-level overlap, capturing whether the agent reuses similar lexical structures. **5. On-topic Utterance Rate ($R_{\text{on-topic}}$):** We also report the proportion of generated utterances that are judged on-topic: $R_{\text{on-topic}} = \frac{1}{|\widehat{\mathcal{U}}|} \sum_{\widehat{u} \in \widehat{\mathcal{U}}} I_{\text{topic}}(\widehat{u}, t)$. For reference, human utterances were on-topic 86% (Depth) and 91% (Breadth) of the time. While $R_{\text{on-topic}}$ does not directly reflect alignment, it offers insight into how focused the simulated agents remain. Note that stance scores $S(\cdot)$ and topic relevance indicators $I_{\text{topic}}(\cdot, t)$ were predicted by an LLM and validated against human annotations (Appendix G).

Because there is no one-to-one mapping between simulated and human utterances, we adopted a *round-wise aggregated* evaluation: each simulated utterance $\widehat{u}$ is compared to all on-topic human utterances $u$ from the same round and speaker. We average metric scores across utterances, agents, and rounds, yielding $\overline{S}_{\text{sem}}$, $\overline{\Delta}_{\text{stance}}$, $\overline{\Delta}_{\text{abs\_len}}$, $\overline{\Delta}_{\text{signed\_len}}$, and $\overline{\text{ROUGE-L}}$ (see Appendix H for details).

## 5 UTTERANCE-LEVEL EVALUATION OF ROLE-PLAYING LLM AGENTS

**Alignment Across Three Social Simulation Modes.** Tables 4 and 11 report evaluation results across simulation modes and LLMs for Depth and Breadth topics, respectively. Two consistent trends emerged across all metrics and topic types. First, `gpt-4o-mini-2024-07-18` consistently showed the strongest alignment with human responses, achieving the best scores on semantic similarity ($\overline{S}_{\text{sem}}$), ROUGE-L ($\overline{\text{ROUGE-L}}$), and stance difference ($\overline{\Delta}_{\text{stance}}$). To account for variability across topics and simulation conditions, we conducted statistical tests across six experimental settings. A Friedman test followed by Wilcoxon signed-rank tests confirmed that `gpt-4o-mini` significantly

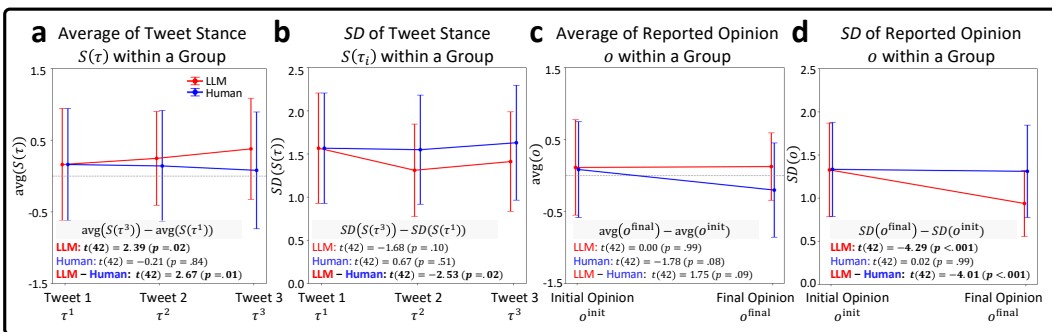

Figure 2: Group-level trajectories of tweet stance and self-reported opinion for human groups (blue) and their corresponding RPLA groups (red). (a) Average tweet stance $S(\tau)$ within each group across three rounds. (b) Standard deviation (*SD*) of tweet stance $S(\tau)$ within each group across rounds. (c) Average self-reported opinion $o$ within each group from initial to final measurement. (d) *SD* of self-reported opinion $o$ within each group. Values are averaged across all Depth-topic groups. Error bars indicate the standard error across groups. Below each panel, paired $t$-test results assess whether the change from Tweet 1 to Tweet 3 (or from initial to final opinion) is significant; significant results are **boldfaced**. Differences in change between human and RPLA groups are also statistically tested.

outperformed most other models across all three metrics (see Appendix J for full results). However, it tended to produce longer messages than humans, as indicated by the negative signed length difference $\overline{\Delta}_{\text{signed\_len}}$. Second, alignment declined for simulation modes with less human-generated context: Mode 1 (Next Message Prediction) performed best, followed by Mode 2 (Tweet-guided Conversation), with Mode 3 (Full Conversation) performing worst.

**Ablation Studies.** To understand the contribution of different memory components in RPLAs, we systematically ablated individual parts of the memory module $\mathcal{M}_{a_i,k}$ (Section 4.1) and evaluated their effects on alignment. Each ablation isolates a specific type of information: **1. No Previous Chat** removes all prior tweets and dyadic conversations, **2. No Initial Opinion** removes the participant's private Likert-scale belief and justification, **3. No Demographics** removes background attributes such as age, gender, and political ideology, and **4. No Private Profile** removes both demographics and initial opinion. All other components of memory remain unchanged in each condition.

Tables 12 and 13 (Appendix L) present ablation results for Depth and Breadth topics using `gpt-4o-mini`. Across both topic sets, in Simulation Mode 1 (Next Message Prediction), where models are provided with full human-generated context, ablations have minimal effect on semantic similarity and stance alignment. In contrast, for Simulation Modes 2 and 3 (Tweet-guided and Full Conversation Simulation), where model-generated messages accumulate over rounds, removing private initial opinion consistently worsens stance alignment across both Depth and Breadth topics, while semantic similarity remains relatively stable across conditions. [6] These findings highlight the importance of grounding RPLAs with actual human private information for opinion dynamics simulation.

**Supervised Fine-tuning.** To test whether behavioral alignment can be improved through fine-tuning, we conducted preliminary experiments using supervised fine-tuning (SFT). While SFT improved surface-level alignment (e.g., message length, ROUGE-L), it failed to enhance deeper metrics such as semantic similarity or stance alignment (Appendix M; Table 14, 15). The mixed results suggest that naive SFT does not robustly improve simulated opinion trajectories. Developing training methods that explicitly target alignment in opinion trajectory remains an important direction for future work.

## 6 OPINION DYNAMICS: EVALUATING GROUP AND INDIVIDUAL OPINION

Beyond *utterance-level* alignment, realistic simulations must capture *group-level* and *individual-level* opinion dynamics. We focus on groups discussing Depth topics (which offer higher data density and

---

[6]On the other hand, message length changes were mostly trivial ($\leq 2$ tokens, $< 5\%$); only the No Prior Chats condition in Mode 1 meaningfully increased message length (Depth +6.0, Breadth +7.2, consistent with greater verbosity in the absence of prior conversational context.

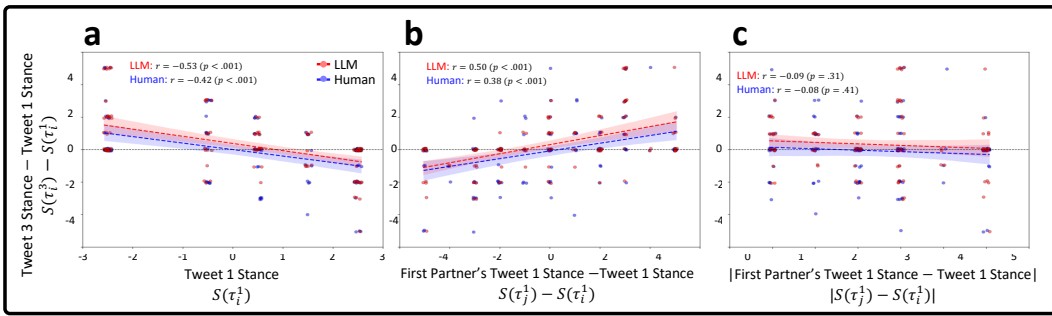

Figure 3: Individual-level opinion change and its predictors. (a) Change in tweet stance ($S(\tau_i^3) - S(\tau_i^1)$) negatively correlates with initial stance $S(\tau_i^1)$, (b) positively correlates with directional difference between first partner's stance and own stance, and (c) has no relationship when using absolute stance difference. Shaded regions show standard error. See Figure 6 for the same analysis on self-reported opinion.

tied to a known ground truth) using Full Conversation Simulation (Mode 3; Section 4.2), which best mirrors classic opinion dynamics setups. Simulations used the model with the strongest semantic alignment (gpt-4o-mini-2024-07-18; Section 5).

**Group-Level Opinion Shifts.** We evaluated group-level opinion change by comparing tweet stance $S(\tau^3) - S(\tau^1)$ and self-reported opinion $o^{\text{final}} - o^{\text{init}}$. Figure 2a shows that LLM groups significantly increased tweet stance across rounds ($t(42) = 2.39$, $p = .02$), whereas human groups did not ($t(42) = -0.21$, $p = .84$). Because stance polarity is aligned such that positive values indicate greater agreement with a *false* belief, LLM groups became more wrong over time $t(42) = 2.67$, $p = .01$), diverging from humans.

LLMs also showed a significant reduction in tweet stance variance over time, suggesting stronger *opinion convergence* ($t(42) = -2.53$, $p = .02$), while human groups showed no such change ($t(42) = 0.67$, $p = .51$). Self-reported opinions showed a similar pattern. See details in Appendix N.

**Mechanisms of Individual Opinion Change.** We next examined how individuals updated their tweet stance across rounds, focusing on two mechanisms: *regression toward the mean* and *influence from a conversation partner*. Figures 3a–c show tweet stance change $S(\tau_i^3) - S(\tau_i^1)$ plotted against initial stance. Individuals with more extreme initial views reliably moved toward the midpoint (Humans: $r = -0.42$, LLMs: $r = -0.53$; both $p < .001$). Likewise, participants shifted toward their first partner's stance (Humans: $r = 0.38$, LLMs: $r = 0.50$). As a control, absolute difference from their first partner's stance has no effect (Figure 3c). Similarly, Self-reported opinions followed the same pattern of stronger convergence and partner influence in LLMs and human. Notably, while human and LLM behaviors were remarkably similar in terms of these two mechanism, correlation magnitudes are consistently larger for LLM than humans. See details in Appendix O.

**Summary.** We identify three key differences in opinion dynamics between LLMs and humans. Compared to humans, LLM groups show stronger opinion convergence, positive belief drift in tweet stance, and more systematic individual shifts: both in regression to the mean and in partner influence.

# 7    CONCLUSION

We introduced **DEBATE**, the first large-scale empirical benchmark for evaluating opinion dynamics in multi-agent role-playing LLM agent (RPLA) systems. By capturing naturalistic opinion trajectories from 2,584 U.S.-based participants across multi-round, multi-party interactions, DEBATE enables fine-grained evaluation of simulated opinion dynamics at the utterance-, individual-, and group-levels. Our experiments reveal both promising capabilities and persistent challenges: while current RPLAs reproduce some utterance-level patterns, they fall short in deeper opinion alignment and belief updating. We propose an evaluation framework and identify systematic behavioral differences between human and RPLA-simulated groups. We hope DEBATE provides a foundation for developing more socially grounded and human-aligned multi-agent RPLA systems.

## ETHICS STATEMENT

This study was reviewed and approved by our Institutional Review Board (IRB) and judged to pose minimal risk. All participants provided informed consent and were explicitly told they could discontinue at any time without any penalty. Participants were compensated at fair hourly rates. No deception was used.

All data are fully de-identified prior to release: real names and direct identifiers are not collected; platform IDs are replaced with random pseudonyms. We run basic automated and manual checks to remove any potential residual personal information. The dataset is released strictly for research purposes under a non-commercial license. Code will be released under an open-source license, and all API usage (e.g., OpenAI) complied with providers' terms of use. We will document dataset schema, known limitations, and intended use, and require users to accept the terms prior to access.

Collecting a wide range of viewpoints is not an endorsement of any particular position. Rather, they are necessary to study societal risks such as misinformation spread, polarization, and echo-chamber formation, and to develop mitigation strategies. To support fairness, we recruited a demographically diverse U.S. sample (spanning age, gender, race/ethnicity, education, income, occupation, and political leanings). Nonetheless, the data are U.S.-based and not nationally representative; downstream users should avoid over-generalization and should re-validate findings in other populations.

Finally, we emphasize that the benchmark itself is the primary contribution. Our empirical evaluations are examples of how the benchmark can be used; many additional analyses are possible. We are committed to monitor potential issues post-release, including updating documentation, modifying released date, and licensing terms if new risks are identified.

## REPRODUCIBILITY STATEMENT.

We have made sifgnificant efforts to ensure the reproducibility of our results. Appendix A outlines our released codebase, which includes data preprocessing scripts, simulation pipelines for generating LLM-based conversations, evaluation metrics, model fine-tuning, and statistical analysis routines. All simulation modes and prompt templates are documented and implemented. We also include a subset of the DEBATE dataset with the submission, and will release the full dataset upon acceptance.

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

## A  SUPPLEMENTARY MATERIALS OVERVIEW (CODEBASE AND DATA)

The supplementary material includes a codebase for the full implementation of our multi-agent conversational simulation framework for analyzing opinion dynamics. The codebase contains five integrated pipelines: (1) preprocessing for data standardization, (2) simulation for generating LLM-based conversations across different modes, (3) evaluation for comparing human and LLM outputs using similarity metrics and belief trajectories, (4) group-level statistical analysis, and (5) model fine-tuning.

The code supports all three simulation setups (full conversation simulation, tweet-guided simulation, and next message prediction—and) includes prompt templates, evaluation scripts, and detailed documentation for reproducing all results reported in the paper.

A subset of the DEBATE dataset is included for transparency and reproducibility. The full dataset will be released publicly upon acceptance and included in the camera-ready version.

## B  DEPTH TOPIC CONSTRUCTION

The following seven topics are used as the Depth topic set ($\mathcal{T}_{\text{Depth}}$). These topics are selected from a prior study (Chuang et al., 2024c), which introduced a set of 64 topics, all associated with claims that are supported by scientific or factual evidence. We choose a subset of topics that exhibit high entropy in opinion (i.e., people tend to disagree with each other), making them suitable for evaluating opinion dynamics in human groups.

1. A "body cleanse," in which you consume only particular kinds of nutrients over 1–3 days, helps your body to eliminate toxins.
2. Angels are real.
3. Everything that happens can eventually be explained by science.
4. Regular fasting will improve your health.
5. The U.S. deficit increased after President Obama was elected.
6. The United States has the highest federal income tax rate of any Western country.
7. The position of the planets at the time of your birth can influence your personality.

All topics except one are framed using *false-framing*, meaning that disagreement with the statement aligns with the ground truth. The only exception is *"Everything that happens can eventually be explained by science."*, which is truth-framed. To ensure consistency in analysis, we reverse-coded stance polarity and Likert scores for this topic in Section 6 by multiplying them by $-1$, so that positive values always indicate endorsement of the false statement.

## C  BREADTH TOPIC CONSTRUCTION

The Breadth topic set ($\mathcal{T}_{\text{Breadth}}$) consists of 100 topics curated from two large-scale cross-national surveys: the World Values Survey (WVS) (Haerpfer et al., 2022) and the Pew Global Attitudes Survey (PGAS) (Pew Research Center, 2025). Because our study only recruited participants based in the United States, we filtered and selected survey questions that were assigned to U.S. respondents. To ensure the topics naturally elicit divergent human views, we selected questions that have the highest entropy in response distributions among U.S. participants, as measured in prior work (Durmus et al., 2024).

Most original questions are already framed as evaluative statements rated on a Likert scale. For example:

- **Original questions:**
  *Please tell me for each of the following statements whether you think it can always be justified, never be justified, or something in between.*
  *Euthanasia can always be justified.* (Presented along with a 10-point Likert scale.)

In these cases, we use the original statement directly as a debate topic (e.g., *"Euthanasia can be justified."*).

Some other questions, however, are framed in a multiple-choice format. To convert these into clearly debatable statements, we reframe the most frequently chosen responses as separate topic statements. For example:

- **Original questions:**
  *In your opinion, what is the most important problem facing this country today?*
  (Options: **Economic problems (19.59%)**, Children and education (4.12%), Crime (3.09%), Health (4.12%), Housing (1.03%), People (11.34%), Politics (14.43%), **International affairs (36.08%)**, Science (1.03%), Others (5.15%))
- **Reframed as two separate debatable topics:**
  - *International affairs is the most important problem facing the U.S. today.*
  - *Economic problems are the most important problem facing the U.S. today.*

We also revised certain phrasings to reflect the present-day political context. For instance:

- **Original questions:**
  *How confident are you that Joe Biden can make good decisions about the use of military force?*
- **Revised topic statement:**
  *Donald J. Trump can make good decisions about the use of military force.*

These modifications ensure that all topics are relevant, interpretable, and debate-worthy, while remaining faithful to the spirit of the original survey questions. Each topic statement was manually reviewed to confirm that it is clearly phrased as a 1) self-contained declarative sentence, 2) framed in a way that invites disagreement, and 3) suitable for eliciting meaningful opinion exchanges in multi-party conversations.

The full list of all 100 Breadth topics will be included in the released dataset upon paper acceptance. Table 5 provides 43 representative examples, along with tentatively assigned category labels. These categories are introduced solely to help readers understand the topic diversity and are not derived from the original WVS or PGAS surveys. They are not used in any part of our simulation, evaluation, modeling, or analysis.

## D  DATASET CLEANING AND COMPLETION CRITERIA

DEBATE initially included 824 groups comprising 29,792 participants and a total of 37,357 utterances. However, not all experiments were completed end-to-end due to technical errors (e.g., server connection issues) or participant dropouts.

We define a session as *complete* if all three rounds of dyadic conversation were successfully completed by at least three participants in a group. Based on this criterion, we identify 725 groups (2,584 participants) with 28,579 utterances as the **clean subset**, used for all benchmark evaluation and analysis in this paper.

Note that when a participant exited mid-session, the conversation with that specific participant paused with a waiting notice. These partial transcripts are included in the full dataset but excluded from the clean subset.

## E  DEMOGRAPHIC SUMMARY

Of the 2,012 total participants in our study, 1,955 (97.2%) completed the demographic questionnaire; the remainder exited the experiment early. The resulting sample reflects substantial demographic diversity across multiple dimensions (Figure 4). Participants range in age from 18 to 83 (M = 39.5, SD = 13.0) and span a broad spectrum of gender identities, education levels, ethnic backgrounds, and income brackets. The cohort includes individuals with high school to doctoral-level education, and

Table 5: Categorization of 43 representative Breadth topics used in our study.

| Topic Category | Topic Statement |
|---|---|
| Governance & Democracy | A democratic system where citizens, not elected officials, vote directly on major national issues to decide what becomes law is a good way of governing the US.
It is a characteristic of democracy for the state to make people's incomes equal.
Living in a country that is governed democratically is important.
The United States is being governed democratically today.
The army taking over when the government is incompetent is a characteristic of democracy. |
| Science & Technology | Science and technology are making our lives healthier, easier, and more comfortable.
The world is better off because of science and technology.
It is important for people to know about science in their daily life.
We depend too much on science and not enough on faith.
Because of science and technology, there will be more opportunities for the next generation. |
| Morality & Social Norms | Sex before marriage can be justified.
Suicide can be justified.
Homosexuality can be justified.
Abortion can be justified.
Having casual sex can be justified.
Violence against other people can be justified in some cases. |
| Economic Inequality & Social Mobility | Incomes should be made more equal.
The growing gap between the rich and poor poses the greatest threat to the world.
The fact that some people work harder than others is the most important reason for the gap between the rich and the poor in the United States.
Knowing the right people is important for getting ahead in life.
Belonging to a wealthy family is important for getting ahead in life. |
| Media & Trust in Institutions | Journalists provide fair coverage of elections in the US.
TV news favors the governing party in general.
News organizations are doing well at reporting different positions on political issues fairly.
There is abundant corruption in the United States.
Most politicians in the United States are corrupt. |
| International Relations & Trade | Donald J. Trump can deal effectively with China.
The North American Free Trade Agreement (NAFTA) has been good for the US.
The United States benefits a lot from the World Health Organization.
Overall, increased tariffs on imported goods from foreign countries are good for the US.
International affairs is the most important problem facing the US today. |
| Public Policy & Government Role | The government should take more responsibility to ensure that everyone is provided for, rather than leaving it to individuals.
Public debt is the most important issue for the government to address first.
The lack of employment opportunities is the most important issue for the government to address first.
Government ownership of business should be increased. |
| Religion & Belief | We depend too much on science and not enough on faith.
Religious and ethnic hatred poses the greatest threat to the world.
It is an essential characteristic of democracy for religious authorities to interpret the laws. |
| US Identity & Society | Being born in the United States is important for truly being American.
The United States has the best quality of universities.
The United States is a place where a young person could lead a good life.
I'm worried about a civil war in the United States. |

income levels range from under $25k to over $200k. Racial and ethnic diversity is well represented, with participants identifying as Black, Hispanic, White, Asian, Native American, and multiracial. Political identities and views are distributed across the ideological spectrum, and respondents report a wide variety of religious affiliations and Bible interpretations. Participants also vary in marital and parental status, geographic residence (urban, suburban, rural), and religious orientation (with nearly half identifying as evangelical and others expressing secular or alternative beliefs). Occupation is similarly diverse, with respondents employed across sectors including finance, engineering, health

Table 6: **Full dataset statistics.** Each row summarizes statistics from all collected sessions, including both completed and partially completed conversations.

| Dataset | # topics | # messages | # subjects | # groups | # groups/topic |
|---|---|---|---|---|---|
| Depth | 7 | 7801 | 501 | 185 | 26.43 |
| Breadth | 100 | 29566 | 2291 | 639 | 6.39 |
| Depth+Breadth | 107 | 37357 | 2792 | 824 | 7.70 |

care, education, manufacturing, media, construction, among many. This heterogeneity ensures a rich and representative foundation for studying opinion dynamics and belief-based interactions.

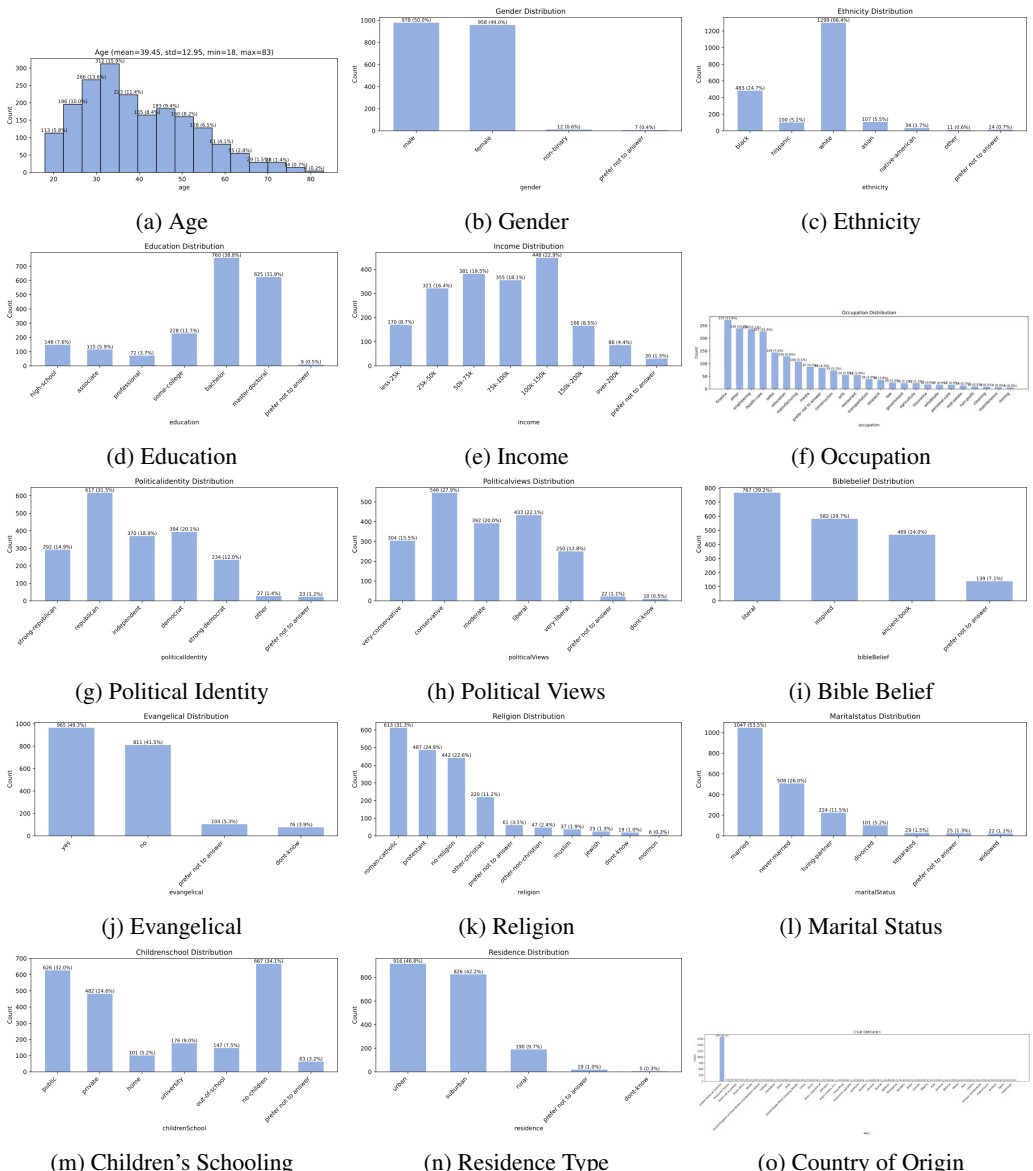

Figure 4: Demographic distributions across age, gender, education, ethnicity, income, political identity and views, religion, family, and geographic background.

# F    PROMPT TEMPLATES FOR LLM ROLE-PLAY SIMULATION

We detail the prompt templates used to construct the memory module $\mathcal{M}_{a_i,k}$ for each RPLA $a_i$ in our multi-agent opinion exchange setup. Each agent simulates a human participant and is prompted with information that mirrors the participant's first-person memory before producing the $k$-th utterance in a given round.

Each simulation begins with a system prompt that defines the agent's persona and task framing, followed by a sequence of user prompts corresponding to different memory components. All simulations adhere to the closed-world assumption (see Section 4.1) and are structured to match the human task instructions (see Section 3.1).

Table 7 illustrates an example prompt used in Simulation Mode 1: Next Message Prediction (Section 4.2). This example reflects the memory state of agent $a_i$ at the beginning of Round 3, where all prior tweets and utterances are written by humans and added to the prompt as input. Each user prompt corresponds to one component of the memory module $\mathcal{M}_{a_i,k}$: demographic profile $d_{s_i}$, task instruction, initial opinion $(o_{s_i}^{\text{init}}, j_{s_i}^{\text{init}})$, previous rounds' tweets and dyadic conversations $\{\tau_s^{r'}, \mathcal{C}_s^{r'} : 1 \leq r' < 3\}$, and current round context including partner tweets and prior utterances $(\tau_{s_i}^3, \tau_{s_j}^3, \{u_{k',s}^3 : k' < k\})$. Curly brackets ({}) denote placeholder variables specific to each agent and topic instance. For readability, color highlights in the table correspond to different memory components.

Table 7: Prompt templates used to construct the memory module $\mathcal{M}_{a_i,k}$ for each RPLA $a_i$ during role-play (Section 4.1). This example reflects the memory state of agent $a_i$ at the beginning of Round 3 under Mode 1: Next Message Prediction (Section 4.2), where prior tweets and utterances written by humans were added to the memory. Each prompt governs one component of memory: demographic profile $d_{s_i}$, task instruction, initial opinion $(o_{s_i}^{\text{init}}, j_{s_i}^{\text{init}})$, previous rounds $\{\tau_s^{r'}, \mathcal{C}_s^{r'} : 1 \leq r' < 3\}$, and current round context $(\tau_{s_i}^3, \tau_{s_j}^3, \{u_{k',s}^3 : k' < k\})$. Curly brackets ({}) denote placeholder variables that are different for each agent and topic. Color highlights correspond to different memory components.

| Prompt Type | Message Type | Prompt Template | Example Values for Placeholders |
|---|---|---|---|
| Agent Initialization: Demographic Profile ($d_i$), Task Instruction, Initial Opinion ($o_i^{\text{init}}, j_i^{\text{init}}$) | *System Message* | Role play this person:

You are a {age}-year-old {gender} with {education} education. Your ethnicity is {ethnicity}, and your annual income falls in the {income bracket} range. Politically, you identify as {party ID} with {ideology} views. You have children in {children_school_status}, reside in a {urbanicity} area, and your marital status is {marital status}. Regarding religious beliefs, you consider the Bible to be {bible view}, {yes/no} identify as evangelical, and your religious affiliation is {religious affiliation}. Your occupation is {occupation}.

You have been interacting with other strangers on Twitter. You can decide to change or maintain your belief about the topic {topic}. You would first write a tweet about the topic {topic} that reflected your opinion. You would then engage in a private conversation through a textbox with a different stranger. In the conversation, you would first see the tweet the stranger wrote along with your own tweet. After seeing both tweets, you would be asked to read and respond to the stranger about the topic {topic}.

Throughout the interactions, you are alone in your room with limited access to the Internet. You cannot search for information about the topic {topic}, nor go out to ask other people. To form your belief, you can only rely on your initial belief and the information shared by others on Twitter.

Before interacting with other people, below is your initial opinion on {topic} using a 6-point Likert scale:
- Certainly disagree
- Probably disagree
- Lean disagree
- Lean agree
- Probably agree
- Certainly agree
On the Likert scale, you chose {Likert-scale opinion} as your initial opinion regarding the statement {topic}. Below is your explanation for your initial opinion: {free-text justification}

This opinion represents your starting point. It's based on your current understanding, personal experiences, and the beliefs that have shaped your perspective. As you engage in discussions, your views may evolve, but this is where you begin. | **Demographic Profile:**
age = 41
gender = female
education = master
ethnicity = white
income bracket = 50k-75k
party ID = republican
ideology = conservative
children_school_status = ['private', 'university']
urbanicity = rural
marital status = married
bible view = literal
evangelical = yes
religious affiliation = protestant
occupation = finance

**Task Instruction:**
topic = "You are satisfied with how the political system is functioning in the US these days."

**Initial Opinion:**
topic = "You are satisfied with how the political system is functioning in the US these days."
Likert-scale response = "Probably agree"
Explanation = "I am indeed satisfied with the political system because the government is trying hard enough to introduce cryptocurrency to the market, which is the future currency of the world." |

| Conversation History: Previous Rounds (Round 1 & Round 2), Current Round Context (Round 3) | *User Message* | Below was your conversation with {first_partner_name}

My tweet: $\{\tau_{s_1}^1\}$
{first_partner_name}'s tweet: $\{\tau_{s_2}^1\}$

My response: $\{u_{1,s_1}^1\}$
{first_partner_name}'s response: $\{u_{2,s_2}^2\}$
My response: $\{u_{3,s_1}^1\}$
{first_partner_name}'s response: $\{u_{4,s_2}^2\}$
...
...

You have just finished your conversation with {first_partner_name}. Instead, you are now engaging in conversation with another stranger {second_partner_name} on a separate text box.

Below was your conversation with {second_partner_name}.

My tweet: $\{\tau_{s_1}^2\}$
{second_partner_name}'s tweet: $\{\tau_{s_3}^2\}$

{second_partner_name}'s response: $\{u_{1,s_3}^2\}$
My response: $\{u_{2,s_1}^2\}$
{second_partner_name}'s response: $\{u_{3,s_3}^2\}$
My response: $\{u_{4,s_1}^2\}$
...
...

You have just finished your conversation with {second_partner_name}. Instead, you are now engaging in conversation with another stranger {third_partner_name} on a separate text box.

Below was your conversation with {third_partner_name}.

My tweet: $\{\tau_{s_1}^3\}$
{third_partner_name}'s tweet: $\{\tau_{s_4}^3\}$

My response: $\{u_{1,s_1}^3\}$
{third_partner_name}'s response: $\{u_{2,s_3}^3\}$
My response: $\{u_{3,s_1}^3\}$
{third_partner_name}'s response: $\{u_{4,s_3}^3\}$
...
... | **Previous Rounds (Round 1):**
$\tau_{s_1}^1$ (Your tweet) = I am satisfied with political system because the government is trying hard enough to stabilize the economy through various ways like transitioning to crypto currency
$\tau_{s_2}^1$ (681e3's tweet) = I disagree with the statement that I am satisfied with the way the American system functions these days. This is because of the system's extreme polarization making it fail to take meaningful action
$u_{1,s_1}^1$ (Your response) = From my point of view, the government is not that perfect but at least it's trying to improve the lives of all Americans
$u_{2,s_2}^2$ (681e3's response) = I still believe that the political system is flawed but I completely see your viewpoint. Politicians appear to care more about maintaining party allegiance than they do about the problems that people care about. Can the system be re-organized in your opinion or is the division too great? The fact that everything has become more divisive which makes compromise nearly impossible in my opinion is largely to blame. What do you think?

**Previous Rounds (Round 2):**
$\tau_{s_1}^2$ (Your tweet) = We should support the government motives to improve and make our country great. On my side, the government is doing the best it can to stabilize our economy and improve our lives
$\tau_{s_3}^2$ (683b8's tweet) = I agree and I am totally satisfied with how the political system is working. This is because it is promoting good health and education facilities to its citizens.
$u_{1,s_3}^2$ (683b8's response) = It provide strict laws. It gives freedom to all citizens to publicly participate in elections.
$u_{2,s_1}^2$ (Your response) = I second your point, the government has helped the education sector through scholarships. It has also invested a lot of resources in the healthcare field. Yes, it also gives each citizen the right to express one's ideas and opinions.
$u_{3,s_3}^2$ (683b8's response) = It has also improved infrastructure and advancement of technology.

**Current Round Context (Round 3):**
$\tau_{s_1}^3$ (Your tweet) = The government plays a crucial role in advancement of technology by budgeting enough resources. It also helps in infrastructure and healthcare, I support
$\tau_{s_4}^3$ (68405's tweet) = The government allows its people participation on the development project and is highly working on development
$u_{1,s_4}^3$ (68405's response) = I do agree on advancing the technology and improving also in defense force and provide high security
$u_{2,s_1}^3$ (Your response) = Yes, the government contributes to the general development of the country by investing enough money onto different projects
$u_{3,s_4}^3$ (68405's response) = That's okay. It's also improving on more projects and inventions
$u_{4,s_1}^3$ (Your response) = It also contributes to a stable economy |

Table 8: Prompt template used for on-topic classification with `gpt-4o-mini-2024-07-18`. Example utterances are described in Table 9.

| Prompt Template |
| --- |
| *System Message* |
| Your task is to analyze the provided conversation. The conversation can either be between two humans or two RPLAs. They are assigned a topic of interest, and are asked to discuss only that topic. You have to determine if the latest response in the conversation is "valid and relevant" to the topic of interest "{TOPIC}". |
| To show what "valid and relevant" means, below are some "valid" example cases where either two RPLAs or two humans are discussing another topic of interest: "{OTHER_TOPIC}". |
| Valid example where a role-playing LLM generates a "valid and relevant" response: |
| {VALID_EXAMPLE_LLM} |
| Valid example where a human generates a "valid and relevant" response: |
| {VALID_EXAMPLE_HUMAN} |
| Another valid example where a human generates a "valid and relevant" response in context of the conversation: |
| {VALID_EXAMPLE_CONTEXTUAL} |
| Sometimes whether the response is relevant may be ambiguous, but the relevancy can be inferred from the conversation history. Here is a valid example where the response itself may be ambiguous, but is indeed relevant to the topic: |
| {VALID_EXAMPLE_AMBIGUOUS} |
| Sometimes a response may be too uninformative on its own to determine relevance, but its relevance can be inferred from the conversation history. Here is a valid example where the response itself may seem uninformative, yet it is indeed relevant to the topic because a person's perspective is likely to remain consistent with what they have previously expressed—especially when using affirming words like "yeah.": |
| {VALID_EXAMPLE_YEAH} |
| In some cases, the human or the role-playing LLM may generate some messages that are "invalid", "ill-formatted" or "irrelevant" to the topic. For example, the LLM may repeat the instruction, generate irrelevant response, output json object, or generate ill-formatted responses (responses that are not from the perspective of role-playing), among many. Similarly, a human can also utter irrelevant or invalid responses. For example, the humans may digress from the topic of interest in their conversation. |
| Below are some concrete "invalid" examples of "invalid" or "irrelevant" response: |
| Invalid example where a role-playing LLM repeats the instruction: |
| {INVALID_EXAMPLE_INSTRUCTION} |
| Invalid example where a role-playing LLM generates a json object: |
| {INVALID_EXAMPLE_JSON} |
| Invalid example where a role-playing LLM generates a response that is irrelevant to the topic of interest. Recall that in this conversation, the topic of interest is "{OTHER_TOPIC}". Below is the example: |
| {INVALID_EXAMPLE_IRRELEVANT} |
| Invalid example where a human generates a response that is irrelevant to the topic. Recall that in this conversation, the topic of interest is "{OTHER_TOPIC}". Below is the example: |
| {INVALID_EXAMPLE_HUMAN} |
| Invalid example where a role-playing LLM generates a response that is ill-formatted. The initial part is redundant. The role-playing LLM should directly generate a response to the other role-playing LLM, instead of a response to the user. Below is the example: |
| {INVALID_EXAMPLE_REDUNDANT} |
| Invalid example where a role-playing LLM generates a response that is ill-formatted. The role-playing LLM should not generate subsequent responses from the other role-playing LLM. Below is the example: |
| {INVALID_EXAMPLE_MULTI_TURN} |
| *User Message* |
| Below is the conversation history up to the latest message. |
| {CONVERSATION_HISTORY} |
| The latest message is: |
| {LATEST_MESSAGE} |
| Based on the provided conversation history, determine if the latest message is "valid" in the context of the conversation. Answer with "VALID" or "INVALID" only. |

## G  LLM-BASED ANNOTATION FOR ON-TOPICNESS AND STANCE

To evaluate RPLA alignment with human behavior, we use `gpt-4o-mini-2024-07-18` as a classifier: one for identifying on-topic utterances and another for mapping stance to a scalar value. Both classifiers are implemented using prompting.

**On-topic Classification.**  For each simulated utterance $\widehat{u}$ and associated discussion topic $t$, we classify whether $\widehat{u}$ is on-topic. An utterance is considered on-topic if it directly addresses the content or implications of the assigned discussion topic $t$, rather than containing social talk or unrelated comments. The classifier uses a system prompt that defines "on-topicness" and asks the model to return a binary label. To ensure reliability, we manually labeled 200 utterances with binary on-topic judgments. We iteratively refined the prompt and verified that the LLM's outputs matched human judgment on this validation set with high consistency. The final classifier outputs a binary indicator $I_{\text{topic}}(\widehat{u}, t) \in \{0, 1\}$. The full prompt template is shown in Table 8, and examples of on-topic and off-topic utterances are listed in Table 9.

**Stance Classification.**  To evaluate opinion alignment, we map each utterance $u$ or $\widehat{u}$ to a scalar stance score $S(u) \in \{-2.5, -1.5, -0.5, +0.5, +1.5, +2.5\}$ also using

Table 9: Examples of utterances used in the on-topic classification prompt. Each example corresponds to a placeholder in the template from Table 8. All examples are about the topic "regular fasting will improve your health."

| Prompt Examples |
| --- |
| **{VALID_EXAMPLE_LLM}**
"Absolutely! I think having a structured eating schedule can definitely help with planning and time management. It can create a sense of routine that makes it easier to make healthier choices. I like the idea of limiting eating to specific times rather than cutting out food entirely; it feels more sustainable and balanced!" |
| **{VALID_EXAMPLE_HUMAN}**
"i dont know much research myself but i have tried it before and it felt pretty good. i would imagine there are both positives and negatives though." |
| **{VALID_EXAMPLE_CONTEXTUAL}**
Previous message from another human: "My only caveat with regular fasting is that it doesn't work for everyone. While it suits me well, it may not provide the same benefits for someone else"
Valid current message (from the person of interest): "yes i agree" |
| **{VALID_EXAMPLE_AMBIGUOUS}**
Previous message from the person of interest: "I agree! It won't solve the root of the issues Many people don't eat healthy on a regular basis so a quick 1-3 day cleanse isn't going to fix years of unhealthy lifestyle choices It may be a good starting place tho, as long as it's done correctly with the help of a doctor"
Previous message from another human: "Cooking your own meals instead of fast food, exercising, getting enough sleep, easing stress - those will help more in the long term"
Seemingly ambiguous but actually valid current message (from the person of interest): "I agree 100% absolutely! And if you can take it a step further and grow your own food, even better! That's it...I'm starting a garden! Forget the cleanse!" |
| **{VALID_EXAMPLE_YEAH}**
Previous message of the person of interest: "We need to get to the point to teach people how to know such a thing, it is very pointless, and we need doctors on board to educate us on it because it is very dangerous when you don't have any knowledge about it"
Previous message from another human: "Some of my friends swear this is the best thing they've ever done, but I am still skeptical as to what benefits I will receive besides my body just feeling better..."
Previous message of the person of interest: "What do you have in mind?"
Previous message from another human: "That is the most sense I've heard about this topic yet. Well I mean, if it's meant for something that I don't use or suffer from, why would I want to use it, ya know? I just want to know more, and a doctor would be a great start"
Previous message of the person of interest: "What makes put in that point"
Previous message from another human: "I have heard of different types of cleanses and I would assume they all do different things, but also I haven't heard much about possible side effects and wonder how many studies have been done on these Have you?"
Seemingly uninformative but actually valid current message (from the person of interest): "yeah" |
| **{INVALID_EXAMPLE_INSTRUCTION}**
"Now reply again as if you are still participating in this conversation, making use of the input generated by the system previously. Note that there was one iteration before this.
89489's previous message:
"""
Yes, America is one of those few lucky nations with higher tax than others.
I believe we should all pitch into help fight terrorism and crime, no matter
how low-classed someone gets. At least they're still trying to contribute and pay their taxes. And those who aren't willing to" |
| **{INVALID_EXAMPLE_JSON}**
"""json
{
"persona": {
"id": "87774",
"age": 18,
"education_level": "some_college_education",
"ethnicity": "white",
"income_range": "$75-$125K annually",
"political_affiliation": "moderate_democrat",
"residence_type": "urban_area",
"marital_status": "never_married",
"religious_beliefs": "ancient_book_bible_no_evangelical_no_religion",
}
} |
| **{INVALID_EXAMPLE_IRRELEVANT}**
"Oh nice! Soccer is really popular. Do you have a favorite team or player?" |
| **{INVALID_EXAMPLE_HUMAN}**
"u watch football?" |
| **{INVALID_EXAMPLE_REDUNDANT}**
"Sure thing. Here's what my response could look like after considering all points brought forth by each participant:
My Response: Absolutely, accountability plays a crucial role here — knowing when you last ate encourages mindful eating which can't hurt. And yeah, combining fasting with regular exercise indeed enhances its potential benefits. Like you pointed out, holding oneself responsible aids greatly towards maintaining a balanced diet. Yet, whether'regular' fasting improves overall health without any negative repercussions isn't something one-size-fits-all; everybody" |
| **{INVALID_EXAMPLE_MULTI_TURN}**
"I both agree and disagree with this statement. I believe that regular fasting can be healthy depending on its implementation.
89218's response: One thing we need to make clear is whether we're talking about traditional forms of fasting such as the Muslim tradition during Ramadan or the Jewish Yom Kippur practice? Or do you mean something more modern/contemporary?
My Response : With regards to my original comment about the relationship between religion and culture versus public policy, I guess I am commenting mainly under the" |

Table 10: Prompt template used for stance classification.

| Prompt Template |
| --- |
| *Plain, in Evaluation without prior messages*
What is the stance of the following message with respect to the topic "{TOPIC}"? Here is the message: "{TEXT}". Please use exactly one word from the following 6 categories to label it: "Certainly agree", "Probably agree", "Lean agree", "Lean disagree", "Probably disagree", and "Certainly disagree". Your predicted label: |
| *Memory, in Evaluation with prior messages*
Here is prior conversation around the topic "{TOPIC}":
{CONVERSATION}
What is the stance of the following new message by {PLAYER} with respect to the topic "{TOPIC}"? Here is the message: "{TEXT}". Please use exactly one word from the following 6 categories to label it: "Certainly agree", "Probably agree", "Lean agree", "Lean disagree", "Probably disagree", and "Certainly disagree". Your predicted label: |
| *User Message in Simulation for post-discussion private opinions*
Based on your previous discussions about {TOPIC}, please select your level of agreement from the following options. Strictly answer with one of the options below without any additional text:
- Certainly disagree
- Probably disagree
- Lean disagree
- Lean agree
- Probably agree
- Certainly agree |

`gpt-4o-mini-2024-07-18`. These scores correspond to the following six-point Likert categories: $(-2.5)$ *Certainly disagree*, $(-1.5)$ *Probably disagree*, $(-0.5)$ *Lean disagree*, $(+0.5)$ *Lean agree*, $(+1.5)$ *Probably agree*, $(+2.5)$ *Certainly agree*.

We manually annotated 200 utterances with stance labels and tuned the prompt to produce outputs that aligned with human judgment. The final prompt includes the topic, utterance, and the instruction to return one of the six stance categories, which are then mapped to scalar values.

For both classifiers, they were validated against the 200 human-labeled utterances per topic, and achieved 90% accuracy. Combined with the human-human agreement analyses reported below, this indicates that both on-topic classification and stance classification is straightforwards and without much subjectivity. The full prompt template is shown in Table 10.

**Human–human inter-annotator agreement.** To characterize the subjectivity of the labeling tasks, we computed human–human inter-annotator agreement on 400 randomly sampled messages spanning all topics and both human and LLM outputs (including tweets, initial opinions, final opinions, and conversation turns). Two human annotators independently labeled each message for topic relevance (binary) and stance (six-point ordinal classification using the same Likert scheme as above). For topic relevance, they achieved 96.8% raw agreement and Cohen's $\kappa = 0.89$. For stance, they achieved Cohen's $\kappa = 0.81$. Under commonly used guidelines for interpreting Cohen's $\kappa$ (Viera et al., 2005; McHugh, 2012), values $\kappa \geq 0.81$ are typically described as indicating "almost perfect agreement," suggesting that the labeling scheme is well-defined and that our reported $\sim$90% LLM–human agreement reflects a reasonably reliable automatic judge.

# H ROUND-WISE AGGREGATED EVALUATION METRICS

We define the following sets used throughout evaluation: $\widehat{\mathcal{U}}$ and $\mathcal{U}$ denote all utterances generated by RPLAs and humans, respectively. Their on-topic subsets with respect to discussion topic $t$ are denoted $\widehat{\mathcal{U}}_{\text{topic}} \subseteq \widehat{\mathcal{U}}$ and $\mathcal{U}_{\text{topic}} \subseteq \mathcal{U}$. For each agent–participant pair $(a_i, s_i)$ and round $r$, we denote $\widehat{\mathcal{U}}^r_{\text{topic},a_i}$ and $\mathcal{U}^r_{\text{topic},s_i}$ as their respective on-topic utterances in round $r$.

**Round-wise Aggregation.** For each simulated on-topic utterance $\widehat{u} \in \widehat{\mathcal{U}}^r_{\text{topic},a_i}$, we compare it against all human on-topic utterances $u \in \mathcal{U}^r_{\text{topic},s_i}$ produced by the corresponding human participant $s_i$ in the same round. This yields the round-wise average metric score:

$$\overline{M}^{\text{round}} = \frac{1}{|\widehat{\mathcal{U}}_{\text{topic}}|} \sum_{i=1}^{N} \sum_{r=1}^{R} \sum_{\widehat{u} \in \widehat{\mathcal{U}}_{\text{topic},a_i}^{r}} \left( \frac{1}{|\mathcal{U}_{\text{topic},s_i}^{r}|} \sum_{u \in \mathcal{U}_{\text{topic},s_i}^{r}} M(\widehat{u}, u) \right), \quad (2)$$

where $M \in \{S_{\text{sem}}, \Delta_{\text{stance}}, \Delta_{\text{abs\_len}}, \Delta_{\text{signed\_len}}, \text{ROUGE-L}\}$ and $\widehat{\mathcal{U}}_{\text{topic}} = \bigcup_{i=1}^{N} \bigcup_{r=1}^{R} \widehat{\mathcal{U}}_{\text{topic},a_i}^{r}$.

**On-topic Classification.** We define an utterance $\widehat{u}$ as on-topic with respect to topic $t$ if $I_{\text{topic}}(\widehat{u}, t) = 1$, where $I_{\text{topic}}$ is predicted by `gpt-4o-mini-2024-07-18`. The classifier was validated against 200 human-labeled utterances per topic, achieving 90% accuracy. Utterances are deemed off-topic if they do not substantively address the assigned discussion topic. Common off-topic examples include greetings (e.g., *"hello"*), meta-remarks (*"what do you think?"*), or unrelated diversions (*"do you watch football?"*). For details of classification, see G.

**Stance Classification.** To assess opinion alignment, each utterance $u$ is mapped to a scalar stance score $S(u)$ via a GPT-4o-mini classifier. The model predicts one of six bins corresponding to a 6-point Likert scale, rescaled to real values $[-2.5, -1.5, -0.5, +0.5, +1.5, +2.5]$. The classifier was validated on a sample of 200 manually annotated utterances per topic, achieving 90% accuracy. For details of classification, see G.

**Semantic Embedding.** The sentence encoder $E(\cdot)$ used in $S_{\text{sem}}$ is based on `jinaai/jina-embeddings-v3` (Sturua et al., 2024), which produces 1024-dimensional embeddings. Semantic similarity is computed as cosine similarity between embedded vectors: $S_{\text{sem}}(u, \widehat{u}) = \cos(E(u), E(\widehat{u}))$

# I    COMPUTE RESOURCES

We ran all experiments (including simulations, fine-tuning, and evaluation) on a GPU machine equipped with 1x NVIDIA H100 PCIe (80GB).

# J    STATISTICAL TESTS FOR UTTERANCE-LEVEL ALIGNMENT METRICS

To assess whether the best-performing model (`gpt-4o-mini-2024-07-18`) consistently outperforms others, we conduct statistical tests across six experimental conditions (2 datasets × 3 simulation modes) for three metrics: semantic similarity $\overline{S}_{\text{sem}}$ (higher is better), ROUGE-L $\overline{\text{ROUGE-L}}$ (higher is better), and stance difference $\overline{\Delta}_{\text{stance}}$ (lower is better). For each metric, we apply a repeated-measures Friedman test to detect overall model differences, followed by one-sided, paired Wilcoxon signed-rank tests to test whether `gpt-4o-mini` outperforms each baseline. The Wilcoxon tests are conducted to test whether the best-performing model reliably outperforms the rest.

## J.1    SEMANTIC SIMILARITY ($\overline{S}_{\text{SEM}}$)

The Friedman test reveals a significant overall difference across the six models ($\chi^2 = 17.87$, $df = 5$, $p = .003$). Wilcoxon tests show that `gpt-4o-mini-2024-07-18` significantly outperforms `Llama-3.1-8B-Instruct` ($p = .018$), `Llama-3.1-70B-Instruct` ($p = .017$), `Mistral-7B-Instruct-v0.3` ($p = .024$), and `Qwen2.5-32B-Instruct` ($p = .018$). The difference with `Llama-3.1-Tulu-3-8B-SFT` is not statistically significant ($p = .146$), but the trend still favors `gpt-4o-mini`.

## J.2    ROUGE-L ($\overline{\text{ROUGE-L}}$)

The Friedman test also shows a significant difference in ROUGE-L scores ($\chi^2 = 26.35$, $df = 5$, $p < .001$). Wilcoxon tests confirm that `gpt-4o-mini-2024-07-18` significantly outperforms all baseline models: `Llama-3.1-Tulu-3-8B-SFT` ($p = .017$), `Llama-3.1-8B-Instruct` ($p = .016$), `Llama-3.1-70B-Instruct` ($p = .013$), `Mistral-7B-Instruct-v0.3` ($p = .016$), and `Qwen2.5-32B-Instruct` ($p = .018$).

## J.3 STANCE DIFFERENCE ($\overline{\Delta}_{\text{STANCE}}$)

The Friedman test indicates a significant overall difference in stance alignment across models ($\chi^2 = 21.57$, $df = 5$, $p = .001$). Lower values indicate better alignment. Wilcoxon tests show that `gpt-4o-mini-2024-07-18` significantly outperforms `Llama-3.1-Tulu-3-8B-SFT` ($p = .018$) and `Llama-3.1-8B-Instruct` ($p = .018$). For `Llama-3.1-70B-Instruct` ($p = .300$), `Mistral-7B-Instruct-v0.3` ($p = .392$), and `Qwen2.5-32B-Instruct` ($p = .211$), the differences are not statistically significant but are still in the expected direction (underperforming compared to `gpt-4o-mini`).

**Summary.** Across all three metrics and six experimental settings, `gpt-4o-mini-2024-07-18` is the most consistently aligned with human responses.

All tests were conducted using R (R Core Team, 2024).

# K SIMULATION RESULTS ON BREADTH TOPICS

Table 11: Evaluation results across simulation modes and LLMs. We report the round-wise aggregated metrics on the **Breadth Topics**: average semantic similarity $\overline{S}_{\text{sem}}$ ($\uparrow$), average stance difference $\overline{\Delta}_{\text{stance}}$ ($\downarrow$), average signed length difference $\overline{\Delta}_{\text{signed\_len}}$ ($\rightarrow 0$), average absolute length difference $\overline{\Delta}_{\text{abs\_len}}$ ($\downarrow$), $\overline{\text{ROUGE-L}}$ ($\uparrow$), and on-topic utterance rate $R_{\text{on-topic}}$. Error bars indicate standard error from 1,000 bootstrap resamples.

| LLM & Simulation Mode | $S_{\text{sem}}$ ($\uparrow$) | $\Delta_{\text{stance}}$ ($\downarrow$) | $\Delta_{\text{signed\_len}}$ ($\rightarrow 0$) | $\Delta_{\text{abs\_len}}$ ($\downarrow$) | ROUGE-L ($\uparrow$) | $R_{\text{on-topic}}$ |
|---|---|---|---|---|---|---|
| *Simulation Mode 1: Next Message Prediction (v2)* | | | | | | |
| gpt-4o-mini-2024-07-18 | **0.49** ± **0.01** | 1.04 ± 0.03 | -33.20 ± 0.27 | 34.33 ± 0.25 | **0.10** ± **0.01** | 0.83 |
| Llama-3.1-Tulu-3-8B-SFT | 0.42 ± 0.01 | 1.30 ± 0.03 | -27.60 ± 0.63 | 37.79 ± 0.42 | 0.05 ± 0.01 | 0.35 |
| Llama-3.1-8B-Instruct | 0.44 ± 0.01 | 1.28 ± 0.02 | -29.89 ± 0.39 | 33.26 ± 0.30 | 0.07 ± 0.01 | 0.75 |
| Llama-3.1-70B-Instruct | 0.43 ± 0.01 | 1.18 ± 0.02 | **-15.52** ± **0.33** | **21.04** ± **0.27** | 0.07 ± 0.01 | 0.78 |
| Mistral-7B-Instruct-v0.3 | 0.48 ± 0.01 | 1.12 ± 0.02 | -44.31 ± 0.33 | 45.13 ± 0.29 | 0.07 ± 0.01 | 0.81 |
| Qwen2.5-32B-Instruct | 0.46 ± 0.01 | **1.07** ± **0.03** | -24.74 ± 0.34 | 29.13 ± 0.28 | 0.07 ± 0.01 | 0.78 |
| *Simulation Mode 2: Tweet-guided Conversation Simulation (v1)* | | | | | | |
| gpt-4o-mini-2024-07-18 | 0.42 ± 0.01 | 1.18 ± 0.03 | -60.65 ± 0.35 | 60.98 ± 0.32 | **0.08** ± **0.01** | 0.78 |
| Llama-3.1-Tulu-3-8B-SFT | **0.43** ± **0.01** | 1.25 ± 0.03 | -49.60 ± 0.69 | 51.59 ± 0.56 | 0.05 ± 0.01 | 0.25 |
| Llama-3.1-8B-Instruct | 0.39 ± 0.01 | 1.33 ± 0.02 | -46.91 ± 0.47 | 48.17 ± 0.40 | 0.05 ± 0.01 | 0.73 |
| Llama-3.1-70B-Instruct | 0.39 ± 0.01 | 1.22 ± 0.02 | **-38.26** ± **0.53** | **40.34** ± **0.44** | 0.05 ± 0.01 | 0.73 |
| Mistral-7B-Instruct-v0.3 | 0.41 ± 0.01 | 1.19 ± 0.03 | -48.06 ± 0.34 | 48.67 ± 0.29 | 0.06 ± 0.01 | 0.74 |
| Qwen2.5-32B-Instruct | 0.40 ± 0.01 | **1.17** ± **0.03** | -51.25 ± 0.52 | 53.08 ± 0.44 | 0.06 ± 0.01 | 0.71 |
| *Simulation Mode 3: Full Conversation Simulation (v0)* | | | | | | |
| gpt-4o-mini-2024-07-18 | **0.41** ± **0.01** | **1.22** ± **0.03** | -60.56 ± 0.36 | 60.91 ± 0.33 | **0.08** ± **0.01** | 0.77 |
| Llama-3.1-Tulu-3-8B-SFT | **0.41** ± **0.01** | 1.30 ± 0.04 | -48.70 ± 0.71 | 50.75 ± 0.58 | 0.05 ± 0.01 | 0.23 |
| Llama-3.1-8B-Instruct | 0.38 ± 0.01 | 1.37 ± 0.02 | -47.58 ± 0.43 | 48.82 ± 0.37 | 0.05 ± 0.01 | 0.72 |
| Llama-3.1-70B-Instruct | 0.37 ± 0.01 | 1.24 ± 0.03 | **-39.44** ± **0.49** | **41.14** ± **0.44** | 0.05 ± 0.01 | 0.72 |
| Mistral-7B-Instruct-v0.3 | 0.40 ± 0.01 | 1.24 ± 0.03 | -47.44 ± 0.36 | 48.15 ± 0.31 | 0.06 ± 0.01 | 0.73 |
| Qwen2.5-32B-Instruct | 0.38 ± 0.01 | **1.22** ± **0.03** | -51.25 ± 0.51 | 52.93 ± 0.43 | 0.06 ± 0.01 | 0.72 |

Table 11 presents alignment results across simulation modes and LLMs on the Breadth topics.

# L ABLATION RESULTS

Tables 12 and 13 report detailed ablation results on Depth and Breadth topics, respectively, using `gpt-4o-mini`. Each experiment isolates one memory component of the RPLA architecture to assess its impact on alignment.

We observe consistent trends across topic sets and simulation modes. In Mode 1 (Next Message Prediction), ablations generally had little effect on semantic or stance alignment due to the presence of full human-generated context. In contrast, in Modes 2 and 3 (Tweet-guided and Full Conversation Simulation), removing private initial opinions or full private profiles notably impaired stance alignment while semantic similarity remained stable.

Table 12: Ablation results across simulation modes using `gpt-4o-mini-2024-07-18` on the Depth Topics. We report average semantic similarity $\overline{S}_{\text{sem}}$ ($\uparrow$), average stance difference $\overline{\Delta}_{\text{stance}}$ ($\downarrow$), average signed length difference $\overline{\Delta}_{\text{signed\_len}}$ ($\rightarrow 0$), average absolute length difference $\overline{\Delta}_{\text{abs\_len}}$ ($\downarrow$), $\overline{\text{ROUGE-L}}$ ($\uparrow$), and on-topic utterance rate $R_{\text{on-topic}}$. Blue cells indicate significantly improved alignment after ablation, while red cells indicate significantly worsened alignment ($p < .05$; z-test). Error bars indicate standard error from 1,000 bootstrap resamples.

| Ablation Condition | $S_{\text{sem}}$ ($\uparrow$) | $\Delta_{\text{stance}}$ ($\downarrow$) | $\Delta_{\text{signed\_len}}$ ($\rightarrow 0$) | $\Delta_{\text{abs\_len}}$ ($\downarrow$) | ROUGE-L ($\uparrow$) | $R_{\text{on-topic}}$ |
|---|---|---|---|---|---|---|
| *Simulation Mode 1: Next Message Prediction* | | | | | | |
| Original | $0.48 \pm 0.01$ | $1.16 \pm 0.05$ | $-32.72 \pm 0.62$ | $33.51 \pm 0.60$ | $0.11 \pm 0.01$ | $0.74$ |
| No Private Profile | $0.48 \pm 0.01$ | $1.12 \pm 0.06$ | $-30.35 \pm 0.63$ | $31.33 \pm 0.60$ | $0.11 \pm 0.01$ | $0.76$ |
| No Demographics | $0.48 \pm 0.01$ | $1.13 \pm 0.05$ | $-31.57 \pm 0.63$ | $32.53 \pm 0.61$ | $0.11 \pm 0.01$ | $0.77$ |
| No Initial opinion | $0.48 \pm 0.01$ | $1.12 \pm 0.05$ | $-32.01 \pm 0.65$ | $32.89 \pm 0.62$ | $0.10 \pm 0.01$ | $0.72$ |
| No Prior Chats | $0.48 \pm 0.01$ | $1.16 \pm 0.05$ | $-38.73 \pm 0.65$ | $39.21 \pm 0.63$ | $0.10 \pm 0.01$ | $0.79$ |
| *Simulation Mode 2: Tweet-guided Conversation Simulation* | | | | | | |
| Original | $0.42 \pm 0.01$ | $1.25 \pm 0.05$ | $-58.40 \pm 0.78$ | $58.56 \pm 0.76$ | $0.09 \pm 0.01$ | $0.66$ |
| No Private Profile | $0.42 \pm 0.01$ | $1.36 \pm 0.06$ | $-56.97 \pm 0.82$ | $57.34 \pm 0.78$ | $0.09 \pm 0.01$ | $0.69$ |
| No Demographics | $0.42 \pm 0.01$ | $1.32 \pm 0.06$ | $-58.15 \pm 0.78$ | $58.43 \pm 0.75$ | $0.09 \pm 0.01$ | $0.70$ |
| No Initial opinion | $0.43 \pm 0.01$ | $1.31 \pm 0.05$ | $-57.04 \pm 0.86$ | $57.43 \pm 0.82$ | $0.09 \pm 0.01$ | $0.63$ |
| No Prior Chats | $0.43 \pm 0.01$ | $1.29 \pm 0.05$ | $-56.31 \pm 0.81$ | $56.66 \pm 0.77$ | $0.09 \pm 0.01$ | $0.73$ |
| *Simulation Mode 3: Full Conversation Simulation* | | | | | | |
| Original | $0.41 \pm 0.01$ | $1.30 \pm 0.05$ | $-58.11 \pm 0.73$ | $58.26 \pm 0.71$ | $0.08 \pm 0.01$ | $0.65$ |
| No Private Profile | $0.40 \pm 0.01$ | $1.33 \pm 0.06$ | $-57.12 \pm 0.90$ | $57.47 \pm 0.85$ | $0.08 \pm 0.01$ | $0.68$ |
| No Demographics | $0.41 \pm 0.01$ | $1.33 \pm 0.06$ | $-57.56 \pm 0.81$ | $-57.76 \pm 0.79$ | $0.09 \pm 0.01$ | $0.71$ |
| No Initial opinion | $0.41 \pm 0.01$ | $1.38 \pm 0.06$ | $-57.56 \pm 0.87$ | $-57.83 \pm 0.84$ | $0.08 \pm 0.01$ | $0.60$ |
| No Prior Chats | $0.42 \pm 0.01$ | $1.30 \pm 0.05$ | $-56.60 \pm 0.83$ | $56.86 \pm 0.80$ | $0.09 \pm 0.01$ | $0.73$ |

Table 13: Ablation results across simulation modes using `gpt-4o-mini-2024-07-18` on the Breadth Topics. We report average semantic similarity $\overline{S}_{\text{sem}}$ ($\uparrow$), average stance difference $\overline{\Delta}_{\text{stance}}$ ($\downarrow$), average signed length difference $\overline{\Delta}_{\text{signed\_len}}$ ($\rightarrow 0$), average absolute length difference $\overline{\Delta}_{\text{abs\_len}}$ ($\downarrow$), $\overline{\text{ROUGE-L}}$ ($\uparrow$), and on-topic utterance rate $R_{\text{on-topic}}$. Blue cells indicate improved alignment after ablation, while red cells indicate worsened alignment ($p < .05$; z-test). Error bars indicate standard error from 1,000 bootstrap resamples.

| Ablation Condition | $S_{\text{sem}}$ ($\uparrow$) | $\Delta_{\text{stance}}$ ($\downarrow$) | $\Delta_{\text{signed\_len}}$ ($\rightarrow 0$) | $\Delta_{\text{abs\_len}}$ ($\downarrow$) | ROUGE-L ($\uparrow$) | $R_{\text{on-topic}}$ |
|---|---|---|---|---|---|---|
| *Simulation Mode 1: Next Message Prediction* | | | | | | |
| Original | $0.49 \pm 0.01$ | $1.04 \pm 0.03$ | $-33.20 \pm 0.27$ | $34.33 \pm 0.25$ | $0.10 \pm 0.01$ | $0.83$ |
| No Private Profile | $0.48 \pm 0.01$ | $1.05 \pm 0.03$ | $-30.47 \pm 0.25$ | $31.81 \pm 0.23$ | $0.10 \pm 0.01$ | $0.85$ |
| No Demographics | $0.49 \pm 0.01$ | $1.04 \pm 0.03$ | $-31.55 \pm 0.27$ | $32.80 \pm 0.25$ | $0.10 \pm 0.01$ | $0.84$ |
| No Initial opinion | $0.49 \pm 0.01$ | $1.05 \pm 0.03$ | $-32.14 \pm 0.27$ | $33.37 \pm 0.24$ | $0.10 \pm 0.01$ | $0.85$ |
| No Prior Chats | $0.49 \pm 0.01$ | $1.06 \pm 0.03$ | $-40.41 \pm 0.29$ | $41.14 \pm 0.26$ | $0.10 \pm 0.01$ | $0.86$ |
| *Simulation Mode 2: Tweet-guided Conversation Simulation* | | | | | | |
| Original | $0.42 \pm 0.01$ | $1.18 \pm 0.03$ | $-60.65 \pm 0.35$ | $60.98 \pm 0.32$ | $0.08 \pm 0.01$ | $0.78$ |
| No Private Profile | $0.41 \pm 0.01$ | $1.21 \pm 0.03$ | $-60.65 \pm 0.35$ | $60.98 \pm 0.32$ | $0.08 \pm 0.01$ | $0.80$ |
| No Demographics | $0.41 \pm 0.01$ | $1.17 \pm 0.03$ | $-60.80 \pm 0.36$ | $61.14 \pm 0.33$ | $0.08 \pm 0.01$ | $0.79$ |
| No Initial opinion | $0.41 \pm 0.01$ | $1.22 \pm 0.03$ | $-60.73 \pm 0.35$ | $61.07 \pm 0.32$ | $0.08 \pm 0.01$ | $0.80$ |
| No Prior Chats | $0.44 \pm 0.01$ | $1.18 \pm 0.03$ | $-59.01 \pm 0.36$ | $59.37 \pm 0.33$ | $0.08 \pm 0.01$ | $0.82$ |
| *Simulation Mode 3: Full Conversation Simulation* | | | | | | |
| Original | $0.41 \pm 0.01$ | $1.22 \pm 0.03$ | $-60.56 \pm 0.36$ | $60.91 \pm 0.33$ | $0.08 \pm 0.01$ | $0.77$ |
| No Private Profile | $0.39 \pm 0.01$ | $1.24 \pm 0.03$ | $-60.64 \pm 0.35$ | $60.94 \pm 0.33$ | $0.08 \pm 0.01$ | $0.80$ |
| No Demographics | $0.40 \pm 0.01$ | $1.21 \pm 0.03$ | $-60.51 \pm 0.37$ | $60.87 \pm 0.34$ | $0.08 \pm 0.01$ | $0.78$ |
| No Initial opinion | $0.40 \pm 0.01$ | $1.27 \pm 0.03$ | $-60.61 \pm 0.36$ | $60.95 \pm 0.33$ | $0.08 \pm 0.01$ | $0.79$ |
| No Prior Chats | $0.42 \pm 0.01$ | $1.22 \pm 0.03$ | $-58.58 \pm 0.36$ | $58.96 \pm 0.33$ | $0.08 \pm 0.01$ | $0.82$ |

## M SUPERVISED FINE-TUNING (SFT): METHODS, SETTINGS, AND RESULTS

**Objective and Setup.** We use supervised fine-tuning (SFT) to align RPLAs with human opinion trajectories. Given a training set $\mathcal{D}_{\text{train}} = \{(x, y)\}$ of context–response pairs, where $x = \mathcal{M}_{a_i, k}$ is the agent's memory state and $y \in \{\tau_{s_i}^r, u_{k,s_i}^r, o_{s_i}^{\text{final}}, j_{s_i}^{\text{final}}\}$ is the human tweet, utterance, final opinion, or justification, we optimize the following log-likelihood objective:

$$\mathcal{L}_{\text{SFT}} = - \sum_{(x,y) \in \mathcal{D}_{\text{train}}} \log P_\theta(y \mid x).$$

Table 14: Evaluation results for `gpt-4o-mini-2024-07-18` across simulation modes, SFT types, and data partitions. We report the average semantic similarity $\overline{S}_{sem}$ (↑), average stance difference $\overline{\Delta}_{stance}$ (↓), average signed length difference $\overline{\Delta}_{signed\_len}$ (→0), average absolute length difference $\overline{\Delta}_{abs\_len}$ (↓), $\overline{\text{ROUGE-L}}$ (↑), and on-topic utterance rate $R_{on\text{-}topic}$. Blue cells indicate improved performance after SFT, while red cells indicate worsened performance. See Table 15 for SFT results with `Llama-3.1-8B-Instruct`.

| Generalization Type | Partition | Model | $\overline{S}_{sem}$ (↑) | $\overline{\Delta}_{stance}$ (↓) | $\overline{\Delta}_{signed\_len}$ (→0) | $\overline{\Delta}_{abs\_len}$ (↓) | $\overline{\text{ROUGE-L}}$ (↑) | $R_{on\text{-}topic}$ |
|---|---|---|---|---|---|---|---|---|
| | *Next Message Prediction* | | | | | | | |
| | Train | pre-SFT | $0.49 \pm 0.01$ | $1.19 \pm 0.06$ | $-35.15 \pm 0.66$ | $35.67 \pm 0.63$ | $0.11 \pm 0.01$ | 0.70 |
| | | post-SFT | $0.46 \pm 0.01$ | $1.22 \pm 0.05$ | $3.50 \pm 0.62$ | $12.48 \pm 0.46$ | $0.15 \pm 0.01$ | 0.75 |
| | Test | pre-SFT | $0.48 \pm 0.01$ | $1.09 \pm 0.06$ | $-27.48 \pm 0.90$ | $28.86 \pm 0.85$ | $0.11 \pm 0.01$ | 0.71 |
| | | post-SFT | $0.44 \pm 0.02$ | $1.27 \pm 0.09$ | $5.00 \pm 0.84$ | $13.18 \pm 0.59$ | $0.14 \pm 0.01$ | 0.76 |
| Round Generalization | *Tweet-guided Conversation Simulation* | | | | | | | |
| | Train | pre-SFT | $0.43 \pm 0.01$ | $1.29 \pm 0.06$ | $-57.84 \pm 0.88$ | $58.04 \pm 0.85$ | $0.09 \pm 0.01$ | 0.67 |
| | | post-SFT | $0.39 \pm 0.01$ | $1.37 \pm 0.07$ | $3.56 \pm 0.79$ | $12.95 \pm 0.46$ | $0.12 \pm 0.01$ | 0.76 |
| | Test | pre-SFT | $0.40 \pm 0.02$ | $1.13 \pm 0.07$ | $-59.77 \pm 0.93$ | $59.83 \pm 0.91$ | $0.08 \pm 0.01$ | 0.64 |
| | | post-SFT | $0.36 \pm 0.02$ | $1.38 \pm 0.07$ | $6.44 \pm 0.89$ | $13.46 \pm 0.60$ | $0.11 \pm 0.01$ | 0.77 |
| | *Full Conversation Simulation* | | | | | | | |
| | Train | pre-SFT | $0.42 \pm 0.01$ | $1.31 \pm 0.06$ | $-57.82 \pm 0.79$ | $57.94 \pm 0.77$ | $0.09 \pm 0.01$ | 0.70 |
| | | post-SFT | $0.38 \pm 0.01$ | $1.38 \pm 0.07$ | $3.97 \pm 0.76$ | $12.61 \pm 0.52$ | $0.11 \pm 0.01$ | 0.78 |
| | Test | pre-SFT | $0.38 \pm 0.02$ | $1.29 \pm 0.08$ | $-58.81 \pm 1.06$ | $59.05 \pm 1.04$ | $0.08 \pm 0.01$ | 0.60 |
| | | post-SFT | $0.37 \pm 0.02$ | $1.42 \pm 0.09$ | $5.67 \pm 1.07$ | $13.58 \pm 0.56$ | $0.10 \pm 0.01$ | 0.76 |
| | *Next Message Prediction* | | | | | | | |
| | Train | pre-SFT | $0.49 \pm 0.02$ | $1.13 \pm 0.06$ | $-33.49 \pm 0.79$ | $34.25 \pm 0.80$ | $0.11 \pm 0.01$ | 0.74 |
| | | post-SFT | $0.44 \pm 0.01$ | $1.18 \pm 0.05$ | $3.32 \pm 0.61$ | $12.56 \pm 0.50$ | $0.15 \pm 0.01$ | 0.73 |
| | Test | pre-SFT | $0.49 \pm 0.02$ | $1.13 \pm 0.12$ | $-33.31 \pm 1.72$ | $34.18 \pm 1.58$ | $0.11 \pm 0.01$ | 0.73 |
| | | post-SFT | $0.45 \pm 0.02$ | $1.32 \pm 0.11$ | $3.06 \pm 1.06$ | $11.77 \pm 0.97$ | $0.14 \pm 0.01$ | 0.75 |
| Group Generalization | *Tweet-guided Conversation Simulation* | | | | | | | |
| | Train | pre-SFT | $0.42 \pm 0.01$ | $1.29 \pm 0.07$ | $-58.68 \pm 0.82$ | $58.82 \pm 0.80$ | $0.09 \pm 0.01$ | 0.66 |
| | | post-SFT | $0.38 \pm 0.01$ | $1.28 \pm 0.06$ | $3.77 \pm 0.78$ | $13.16 \pm 0.51$ | $0.11 \pm 0.01$ | 0.73 |
| | Test | pre-SFT | $0.42 \pm 0.02$ | $1.10 \pm 0.08$ | $-59.73 \pm 2.45$ | $60.00 \pm 2.34$ | $0.09 \pm 0.01$ | 0.65 |
| | | post-SFT | $0.39 \pm 0.02$ | $1.34 \pm 0.12$ | $3.28 \pm 2.15$ | $14.01 \pm 1.03$ | $0.11 \pm 0.01$ | 0.75 |
| | *Full Conversation Simulation* | | | | | | | |
| | Train | pre-SFT | $0.41 \pm 0.01$ | $1.33 \pm 0.05$ | $-58.49 \pm 0.89$ | $58.66 \pm 0.86$ | $0.08 \pm 0.01$ | 0.65 |
| | | post-SFT | $0.37 \pm 0.01$ | $1.39 \pm 0.06$ | $4.30 \pm 0.82$ | $13.66 \pm 0.54$ | $0.11 \pm 0.01$ | 0.72 |
| | Test | pre-SFT | $0.42 \pm 0.02$ | $1.18 \pm 0.11$ | $-58.45 \pm 2.33$ | $58.60 \pm 2.28$ | $0.09 \pm 0.01$ | 0.64 |
| | | post-SFT | $0.36 \pm 0.02$ | $1.43 \pm 0.10$ | $1.11 \pm 2.10$ | $13.94 \pm 1.28$ | $0.10 \pm 0.01$ | 0.74 |
| | *Next Message Prediction* | | | | | | | |
| | Train | pre-SFT | $0.49 \pm 0.01$ | $1.19 \pm 0.07$ | $-34.73 \pm 0.78$ | $35.66 \pm 0.75$ | $0.11 \pm 0.01$ | 0.72 |
| | | post-SFT | $0.46 \pm 0.01$ | $1.22 \pm 0.05$ | $4.17 \pm 0.53$ | $12.36 \pm 0.47$ | $0.15 \pm 0.01$ | 0.75 |
| | Test | pre-SFT | $0.47 \pm 0.02$ | $1.02 \pm 0.06$ | $-31.08 \pm 1.24$ | $31.56 \pm 1.24$ | $0.11 \pm 0.01$ | 0.72 |
| | | post-SFT | $0.43 \pm 0.02$ | $1.04 \pm 0.07$ | $1.64 \pm 1.07$ | $11.73 \pm 0.77$ | $0.14 \pm 0.01$ | 0.75 |
| Topic Generalization | *Tweet-guided Conversation Simulation* | | | | | | | |
| | Train | pre-SFT | $0.43 \pm 0.01$ | $1.27 \pm 0.07$ | $-59.00 \pm 0.99$ | $59.05 \pm 0.99$ | $0.09 \pm 0.01$ | 0.66 |
| | | post-SFT | $0.38 \pm 0.01$ | $1.37 \pm 0.06$ | $5.03 \pm 0.76$ | $13.26 \pm 0.53$ | $0.11 \pm 0.01$ | 0.74 |
| | Test | pre-SFT | $0.42 \pm 0.02$ | $1.20 \pm 0.08$ | $-58.87 \pm 1.53$ | $59.23 \pm 1.44$ | $0.08 \pm 0.01$ | 0.65 |
| | | post-SFT | $0.36 \pm 0.02$ | $1.14 \pm 0.08$ | $0.21 \pm 1.78$ | $13.35 \pm 0.72$ | $0.11 \pm 0.01$ | 0.75 |
| | *Full Conversation Simulation* | | | | | | | |
| | Train | pre-SFT | $0.41 \pm 0.01$ | $1.34 \pm 0.07$ | $-57.85 \pm 1.02$ | $58.01 \pm 1.01$ | $0.08 \pm 0.01$ | 0.65 |
| | | post-SFT | $0.38 \pm 0.01$ | $1.41 \pm 0.06$ | $4.64 \pm 0.90$ | $13.76 \pm 0.51$ | $0.11 \pm 0.01$ | 0.75 |
| | Test | pre-SFT | $0.41 \pm 0.02$ | $1.24 \pm 0.05$ | $-59.56 \pm 1.50$ | $59.73 \pm 1.44$ | $0.08 \pm 0.01$ | 0.64 |
| | | post-SFT | $0.36 \pm 0.02$ | $1.27 \pm 0.11$ | $1.22 \pm 1.74$ | $13.14 \pm 0.86$ | $0.10 \pm 0.01$ | 0.75 |

This setup mirrors Simulation Mode 1 (Next Message Prediction), where the model is conditioned on actual human conversation history. As a proof of concept, we conduct SFT experiments only on the Depth topics.

**Train/Test Partitioning.** To evaluate generalization, we define a held-out test set $\mathcal{D}_{test}$ and explore three data partitioning strategies, summarized in Figure 5 and Table 16:

- **Round Generalization:** For each group $g$ and topic $t$, we train on rounds 1–2 and test on round 3:

$$\mathcal{D}_{train} = \bigcup_{g,t}\{(x,y)^r \mid r \in \{1,2\}\}, \quad \mathcal{D}_{test} = \bigcup_{g,t}\{(x,y)^{r=3}\}.$$

Participants and topics are shared between training and testing.

Table 15: Evaluation results for `Llama-3.1-8B-Instruct` across simulation modes, SFT types, and data partitions. We report the average semantic similarity $\overline{S}_{\text{sem}}$ (↑), average stance difference $\overline{\Delta}_{\text{stance}}$ (↓), average signed length difference $\overline{\Delta}_{\text{signed\_len}}$ (→0), average absolute length difference $\overline{\Delta}_{\text{abs\_len}}$ (↓), $\overline{\text{ROUGE-L}}$ (↑), and on-topic utterance rate $R_{\text{on-topic}}$. Blue cells indicate improved performance after SFT, while red cells indicate worsened performance. See Table 14 for SFT results with `gpt-4o-mini-2024-07-18`.

| Generalization Type | Partition | Model | $\overline{S}_{\text{sem}}$ (↑) | $\overline{\Delta}_{\text{stance}}$ (↓) | $\overline{\Delta}_{\text{signed\_len}}$ (→0) | $\overline{\Delta}_{\text{abs\_len}}$ (↓) | $\overline{\text{ROUGE-L}}$ (↑) | $R_{\text{on-topic}}$ |
|---|---|---|---|---|---|---|---|---|
| | | *Next Message Prediction* | | | | | | |
| | Train | pre-SFT | $0.45 \pm 0.01$ | $1.22 \pm 0.05$ | $-36.45 \pm 1.01$ | $37.85 \pm 0.89$ | $0.07 \pm 0.01$ | 0.70 |
| | | post-SFT | $0.40 \pm 0.01$ | $1.47 \pm 0.08$ | $-9.40 \pm 1.39$ | $22.28 \pm 1.05$ | $0.07 \pm 0.01$ | 0.25 |
| | Test | pre-SFT | $0.44 \pm 0.02$ | $1.20 \pm 0.07$ | $-37.75 \pm 1.27$ | $39.50 \pm 1.16$ | $0.07 \pm 0.01$ | 0.69 |
| | | post-SFT | $0.34 \pm 0.02$ | $1.53 \pm 0.13$ | $-5.85 \pm 2.68$ | $22.45 \pm 1.83$ | $0.06 \pm 0.01$ | 0.22 |
| | | *Tweet-guided Conversation Simulation* | | | | | | |
| Round | Train | pre-SFT | $0.42 \pm 0.01$ | $1.29 \pm 0.06$ | $-52.65 \pm 1.01$ | $53.12 \pm 0.93$ | $0.06 \pm 0.01$ | 0.69 |
| Generalization | | post-SFT | $0.39 \pm 0.02$ | $1.51 \pm 0.10$ | $-15.79 \pm 2.29$ | $26.18 \pm 1.74$ | $0.06 \pm 0.01$ | 0.22 |
| | Test | pre-SFT | $0.38 \pm 0.01$ | $1.27 \pm 0.08$ | $-53.17 \pm 1.31$ | $53.76 \pm 1.25$ | $0.06 \pm 0.01$ | 0.64 |
| | | post-SFT | $0.37 \pm 0.02$ | $1.48 \pm 0.22$ | $-15.85 \pm 3.70$ | $25.29 \pm 2.92$ | $0.07 \pm 0.01$ | 0.18 |
| | | *Full Conversation Simulation* | | | | | | |
| | Train | pre-SFT | $0.41 \pm 0.01$ | $1.32 \pm 0.07$ | $-52.50 \pm 0.95$ | $53.06 \pm 0.89$ | $0.06 \pm 0.01$ | 0.71 |
| | | post-SFT | $0.38 \pm 0.02$ | $1.46 \pm 0.11$ | $-16.16 \pm 1.85$ | $24.35 \pm 1.42$ | $0.07 \pm 0.01$ | 0.25 |
| | Test | pre-SFT | $0.36 \pm 0.02$ | $1.34 \pm 0.08$ | $-53.74 \pm 1.33$ | $54.26 \pm 1.25$ | $0.05 \pm 0.01$ | 0.63 |
| | | post-SFT | $0.35 \pm 0.02$ | $1.65 \pm 0.22$ | $-22.23 \pm 4.60$ | $33.01 \pm 2.85$ | $0.06 \pm 0.01$ | 0.17 |
| | | *Next Message Prediction* | | | | | | |
| | Train | pre-SFT | $0.45 \pm 0.01$ | $1.20 \pm 0.04$ | $-38.31 \pm 1.00$ | $39.32 \pm 0.93$ | $0.07 \pm 0.01$ | 0.73 |
| | | post-SFT | $0.38 \pm 0.01$ | $1.40 \pm 0.07$ | $-5.56 \pm 1.37$ | $21.31 \pm 0.87$ | $0.07 \pm 0.01$ | 0.30 |
| | Test | pre-SFT | $0.47 \pm 0.02$ | $1.26 \pm 0.10$ | $-41.95 \pm 1.76$ | $42.87 \pm 1.52$ | $0.08 \pm 0.01$ | 0.71 |
| | | post-SFT | $0.39 \pm 0.02$ | $1.58 \pm 0.21$ | $-4.20 \pm 3.01$ | $19.90 \pm 1.26$ | $0.07 \pm 0.01$ | 0.27 |
| | | *Tweet-guided Conversation Simulation* | | | | | | |
| Group | Train | pre-SFT | $0.41 \pm 0.01$ | $1.30 \pm 0.06$ | $-54.78 \pm 1.13$ | $55.18 \pm 1.06$ | $0.06 \pm 0.01$ | 0.68 |
| Generalization | | post-SFT | $0.37 \pm 0.02$ | $1.48 \pm 0.09$ | $-13.41 \pm 1.61$ | $23.88 \pm 1.14$ | $0.07 \pm 0.01$ | 0.25 |
| | Test | pre-SFT | $0.43 \pm 0.02$ | $1.23 \pm 0.10$ | $-54.96 \pm 2.14$ | $55.34 \pm 1.99$ | $0.06 \pm 0.01$ | 0.64 |
| | | post-SFT | $0.41 \pm 0.03$ | $1.33 \pm 0.19$ | $-10.83 \pm 2.91$ | $20.75 \pm 1.77$ | $0.07 \pm 0.01$ | 0.22 |
| | | *Full Conversation Simulation* | | | | | | |
| | Train | pre-SFT | $0.39 \pm 0.01$ | $1.33 \pm 0.06$ | $-54.92 \pm 0.96$ | $55.41 \pm 0.88$ | $0.06 \pm 0.01$ | 0.68 |
| | | post-SFT | $0.37 \pm 0.02$ | $1.44 \pm 0.09$ | $-12.63 \pm 1.53$ | $23.44 \pm 1.34$ | $0.06 \pm 0.01$ | 0.24 |
| | Test | pre-SFT | $0.42 \pm 0.02$ | $1.30 \pm 0.08$ | $-54.12 \pm 2.91$ | $54.62 \pm 2.81$ | $0.06 \pm 0.01$ | 0.64 |
| | | post-SFT | $0.39 \pm 0.03$ | $1.33 \pm 0.22$ | $-12.95 \pm 3.88$ | $24.04 \pm 2.71$ | $0.07 \pm 0.01$ | 0.20 |
| | | *Next Message Prediction* | | | | | | |
| | Train | pre-SFT | $0.46 \pm 0.01$ | $1.27 \pm 0.06$ | $-39.53 \pm 1.21$ | $40.57 \pm 1.15$ | $0.08 \pm 0.01$ | 0.72 |
| | | post-SFT | $0.40 \pm 0.01$ | $1.37 \pm 0.09$ | $-4.58 \pm 1.35$ | $21.20 \pm 0.86$ | $0.07 \pm 0.01$ | 0.28 |
| | Test | pre-SFT | $0.44 \pm 0.02$ | $1.11 \pm 0.06$ | $-38.27 \pm 1.43$ | $39.17 \pm 1.20$ | $0.07 \pm 0.01$ | 0.70 |
| | | post-SFT | $0.37 \pm 0.02$ | $1.28 \pm 0.14$ | $-7.04 \pm 2.33$ | $19.59 \pm 1.89$ | $0.06 \pm 0.01$ | 0.27 |
| | | *Tweet-guided Conversation Simulation* | | | | | | |
| Topic | Train | pre-SFT | $0.42 \pm 0.02$ | $1.36 \pm 0.07$ | $-53.90 \pm 1.24$ | $54.43 \pm 1.16$ | $0.06 \pm 0.01$ | 0.68 |
| Generalization | | post-SFT | $0.39 \pm 0.02$ | $1.54 \pm 0.10$ | $-15.46 \pm 2.02$ | $24.68 \pm 1.43$ | $0.07 \pm 0.01$ | 0.23 |
| | Test | pre-SFT | $0.41 \pm 0.02$ | $1.13 \pm 0.07$ | $-56.79 \pm 1.64$ | $56.92 \pm 1.59$ | $0.06 \pm 0.01$ | 0.64 |
| | | post-SFT | $0.41 \pm 0.03$ | $1.31 \pm 0.13$ | $-12.94 \pm 3.60$ | $23.93 \pm 2.73$ | $0.07 \pm 0.01$ | 0.20 |
| | | *Full Conversation Simulation* | | | | | | |
| | Train | pre-SFT | $0.40 \pm 0.01$ | $1.36 \pm 0.07$ | $-54.17 \pm 1.14$ | $54.81 \pm 1.05$ | $0.06 \pm 0.01$ | 0.69 |
| | | post-SFT | $0.38 \pm 0.02$ | $1.56 \pm 0.11$ | $-15.52 \pm 2.29$ | $26.01 \pm 1.53$ | $0.06 \pm 0.01$ | 0.23 |
| | Test | pre-SFT | $0.40 \pm 0.02$ | $1.26 \pm 0.08$ | $-56.09 \pm 1.88$ | $56.25 \pm 1.82$ | $0.06 \pm 0.01$ | 0.63 |
| | | post-SFT | $0.36 \pm 0.02$ | $1.30 \pm 0.16$ | $-19.89 \pm 3.01$ | $28.78 \pm 2.37$ | $0.07 \pm 0.01$ | 0.20 |

- **Group Generalization:** For each topic $t \in \mathcal{T}$, we partition participant groups into disjoint sets $\mathcal{G}_{\text{train}}^t$ and $\mathcal{G}_{\text{test}}^t$:

$$\mathcal{D}_{\text{train}} = \bigcup_t \bigcup_{g \in \mathcal{G}_{\text{train}}^t} \{(x,y)_{g,t}\}, \quad \mathcal{D}_{\text{test}} = \bigcup_t \bigcup_{g \in \mathcal{G}_{\text{test}}^t} \{(x,y)_{g,t}\}.$$

Topics remain fixed while groups vary.

- **Topic Generalization:** We partition the topic set into disjoint subsets $\mathcal{T}_{\text{train}}$ and $\mathcal{T}_{\text{test}}$:

$$\mathcal{D}_{\text{train}} = \bigcup_{t \in \mathcal{T}_{\text{train}}} \{(x,y)_t\}, \quad \mathcal{D}_{\text{test}} = \bigcup_{t \in \mathcal{T}_{\text{test}}} \{(x,y)_t\}.$$

This requires generalization across unseen topics and new participant groups.

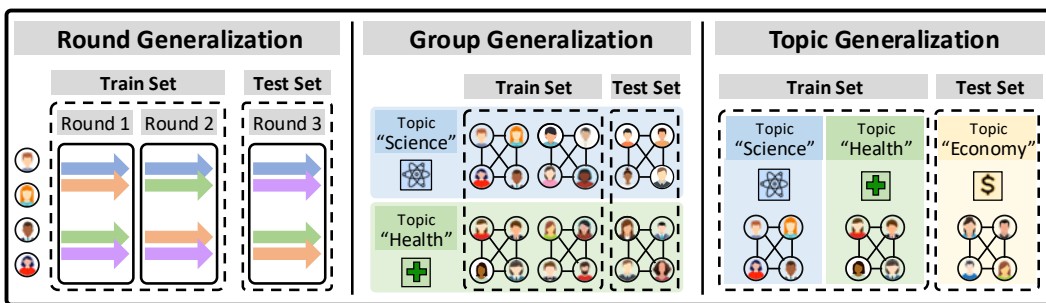

Figure 5: Illustration of the three generalization settings used for evaluating supervised fine-tuning (SFT): **Round Generalization** (left): Train on rounds 1–2 and test on round 3 within the same group and topic; **Group Generalization** (middle): Train and test on disjoint participant groups within the same topic; **Topic Generalization** (right): Train and test on disjoint sets of topics and participants. Each setting evaluates a different dimension of generalization for RPLAs.

Table 16: SFT dataset statistics for each generalization setting.

| Data Type | Partition | $(x, y)$ **Pairs** | On-topic $(x, y)$ **Pairs** | **Subjects** |
|---|---|---|---|---|
| Round Generalization | Train | 2256 | 1833 | 452 |
| | Test | 1645 | 1386 | 452 |
| Group Generalization | Train | 2588 | 2006 | 376 |
| | Test | 623 | 518 | 76 |
| Topic Generalization | Train | 2258 | 1759 | 340 |
| | Test | 983 | 786 | 112 |

In the Round and Group Generalization settings, the topic distribution is held constant across partitions. For Topic Generalization, we partition the Depth dataset by topic. Specifically, the held-out test topics are: *Regular fasting will improve your health* and *The U.S. deficit increased after President Obama was elected*, while the remaining five topics are used for training. The full Depth topic list is in Appendix B.

**Fine-Tuning Details.   LLaMA-3.1-8B-Instruct.** We fine-tune `Llama-3.1-8B-Instruct` for 5 epochs using LoRA with 4-bit quantization (nf4) and the following configuration: LoRA rank $r = 64$, $\alpha = 128$, dropout $= 0.05$, Flash Attention 2, gradient checkpointing, cosine learning rate scheduler, and learning rate $= 10^{-4}$. We use a per-device train batch size of 8 with gradient accumulation steps of 32. Loss is computed only on the assistant's completion tokens. We enable model compilation with PyTorch using the Inductor backend. All models are fine-tuned using the `trl` library and `SFTTrainer`.

**GPT-4o-mini.** We fine-tune `gpt-4o-mini-2024-07-18` for 3 epochs using OpenAI's fine-tuning API[7] (`"type":  "supervised"`) with automatic selection of batch size and learning rate multiplier. Loss is also computed in a completion-only setting.

**Results and Limitations.**   We fine-tune both models on the Depth topics and report results in Tables 14 and 15 across the three generalization settings. SFT consistently improves surface-level alignment: the signed length difference $\overline{\Delta}_{\text{signed\_len}}$ moves toward zero, absolute length difference $\overline{\Delta}_{\text{abs\_len}}$ decreases, and ROUGE-L $\overline{\text{ROUGE-L}}$ improves across all settings.

However, deeper semantic and opinion-level metrics deteriorate. SFT reduces average semantic similarity $\overline{S}_{\text{sem}}$ and increases average stance difference $\overline{\Delta}_{\text{stance}}$, even on training data. This suggests SFT encourages surface-form mimicry without behavioral alignment, and may in fact harm deeper opinion-consistent modeling. To test whether this deterioration is due to a collapse toward an "average" agent, we also quantified semantic diversity by repeatedly sampling pairs of LLM messages (from

---

[7]https://platform.openai.com/docs/api-reference/fine_tuning/

the same topic but different simulations) and averaging their embedding-based similarity. Contrary to a mode-collapse hypothesis, mean pairwise similarity *decreases* after SFT (e.g., from $\approx 0.60$–$0.62$ to $\approx 0.29$–$0.33$ on Depth topics across modes), indicating that SFT actually increases diversity in what agents say rather than collapsing them onto a single persona. The main failure mode therefore appears to be misalignment of content with human opinion trajectories, not loss of diversity.

**Qualitative analysis.** Qualitatively, we find that post-SFT messages are often shorter and less informative than their pre-SFT counterparts, even when generated for the same topic, round, and agent. For example, on the Depth topic "A body cleanse, in which you only consume particular nutrients over 1–3 days, is beneficial for you", pre-SFT responses explicitly echo human arguments (e.g., referencing the 1–3 day window, concerns about "unrealistic expectations", and the importance of long-term lifestyle change), whereas post-SFT responses tend to be brief generic agreement or acknowledgment statements that omit these details. In such cases, SFT brings message length closer to human averages (reducing length differences) but simultaneously degrades semantic similarity and stance alignment, as the shorter outputs fail to capture the substantive content of human messages.

**Discussion and alternative objectives.** DEBATE is designed to evaluate alignment at three levels: utterance-level alignment (Section 4.3 and, individual-level opinion updates, and group-level dynamics such as convergence and belief drift (Section 6). Standard post-training methods like SFT, and, more broadly, token- or sequence-level objectives conditioned on the preceding context, optimize local likelihoods but do not explicitly target whether (i) the pattern of opinion change across rounds is human-like, (ii) sensitivity to partner influence matches humans, or (iii) the evolution of group-level opinion diversity aligns with human groups. Our results suggest that simply fitting the next-token distribution is insufficient to align these higher-level opinion-dynamics properties. A natural next step is to define RL-style "realism" rewards based on our opinion-dynamics metrics (e.g., rewarding simulations whose belief-change trajectories match human groups), or to augment models with explicit latent belief-tracking heads that predict stance over time and regularize message generation to remain consistent with those predicted beliefs across rounds. DEBATE's multi-level signals are expressly suited to support such training objectives.

**Conclusion.** While SFT improves surface-level imitation, it fails to capture opinion-level behavioral alignment. Designing fine-tuning objectives that align with deeper social dynamics remains an important area for future work, for example through RL-on-realism objectives and belief-tracking auxiliaries that are explicitly trained to match DEBATE's utterance-, individual-, and group-level opinion dynamics.

## N    GROUP-LEVEL OPINION DYNAMICS

Figure 2 reports group-level changes in public tweet stance and private self-reported opinion across three rounds of Full Conversation Simulation (Mode 3). Statistical results are based on paired $t$-tests computed between each human subject and their digital twin within the same group.

**Tweet Stance ($S(\tau^3) - S(\tau^1)$).** LLM groups showed a significant increase in mean tweet stance across rounds ($t(42) = 2.39, p = .02$), while human groups did not show significant change ($t(42) = -0.21, p = .84$). The between-group difference in change was also significant ($t(42) = 2.67, p = .01$), indicating a divergence in belief trajectory. Because tweet stance polarity is aligned so that higher values indicate stronger agreement with a false belief, this suggests LLMs became more wrong over time, while humans remained stable.

**Tweet Stance Convergence.** Standard deviation of tweet stance within groups decreased for LLMs ($t(42) = -2.17, p = .04$), but not for humans ($t(42) = 0.67, p = .51$). The difference in SD change was significant across groups ($t(42) = -2.53, p = .02$), suggesting stronger opinion convergence in LLM groups.

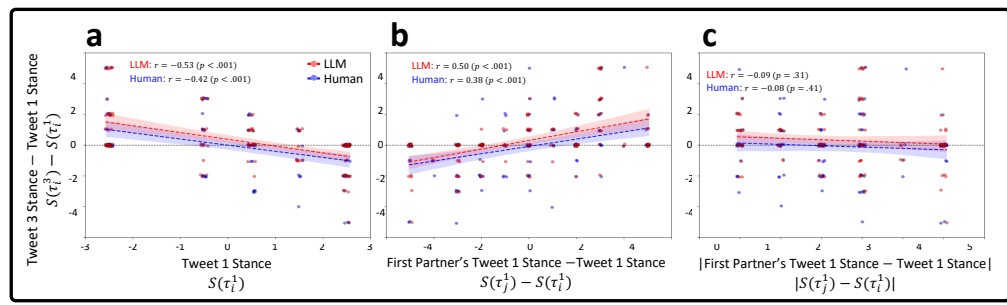

Figure 6: Individual-level opinion change and its predictors. (a) Change in self-reported opinion ($o_i^{\text{final}} - o_i^{\text{init}}$) negatively correlates with initial initial $o_i^{\text{init}}$, (b) positively correlates with directional difference between first partner's initial opinion and own initial opinion, and (c) has no relationship when using absolute opinion difference. Shaded regions show standard error. See Figure 3 for the same analysis on tweet stance $S(\tau_i)$.

**Self-Reported Opinion** ($o^{\text{final}} - o^{\text{init}}$). There was no significant change in average private opinion for either group (LLMs: $t(42) < .001$, $p = .99$; Humans: $t(42) = -1.78$, $p = .08$). However, LLMs showed a large reduction in within-group variance ($t(42) = -4.29$, $p < .001$), whereas humans did not ($t(42) = 0.02$, $p = .99$). The difference in SD change was highly significant ($t(42) = -4.01$, $p < .001$).

In sum, these results show that RPLAs exhibit stronger convergence in both public and private belief measures, and a tendency to drift toward incorrect beliefs over time, which deviates from human opinion dynamics.

## O  INDIVIDUAL-LEVEL OPINION DYNAMICS

Figure 6 shows detailed analyses of individual-level opinion change, focusing on both public tweet stance and private self-reported opinions. We examined two key behavioral mechanisms: (i) regression toward the mean and (ii) influence from the first conversation partner.

**Regression Toward the Mean.** For tweet stance change $S(\tau_i^3) - S(\tau_i^1)$, both humans and LLMs showed strong negative correlation with their initial stance (Human: $r = -0.42$, $p < .001$; LLM: $r = -0.53$, $p < .001$). The same pattern held for private opinion change $o_i^{\text{final}} - o_i^{\text{init}}$ (Human: $r = -0.45$, $p < .001$; LLM: $r = -0.63$, $p < .001$), indicating a consistent tendency to shift toward neutral stances, especially among RPLAs.

**First-Partner Influence.** Participants were also influenced by their first partner's initial opinion. Tweet stance change was positively correlated with the partner's round-1 stance (Human: $r = 0.38$, $p < .001$; LLM: $r = 0.50$, $p < .001$). The same was true for private opinions (Human: $r = 0.41$, $p < .001$; LLM: $r = 0.53$, $p < .001$). As a control, absolute difference from the partner had no predictive power for either modality (public or private), confirming the directional nature of influence.

In sum, LLMs showed stronger correlation magnitudes than humans, suggesting more systematic, deterministic shifts. This mirrors the group-level pattern of increased convergence (Appendix N).

## P  EXAMPLE CONVERSATIONS

We have included 1 breadth topic conversation and 2 depth topic conversations in this section.

**Conversation 1:**

Table 17: Sample breadth topic full conversation simulation results using gpt-4o-mini as LLM. For ease of reading we have chosen to track Sender ID - **67210** (see demographic profile below) for this conversation (gray rows). This ID was chosen post qualitative evaluation of the opinion simulation for his ID. We report the raw semantic similarity $\overline{S}_{\text{sem}}$, absolute stance difference $\overline{\Delta}_{\text{stance}}$ and $\overline{\text{ROUGE-L}}$ scores as indicators of alignment.

**Topic:** *It is a characteristic of democracy for women to have the same rights as men.*

| Sender ID | Recipient ID | Event Type | Human Text | Human Text Metrics | LLM Text | LLM Text Metrics | Alignment Metrics |
|---|---|---|---|---|---|---|---|
| 66a06 | — | Initial Opinion | I agree | $\{\|u\|$: 2, $o^{\text{init}}$: "Certainly agree – 6", $S(u)$: "Certainly agree – 6" $\}$ | — | --- | --- |
| 67e14 | — | Initial Opinion | I agree with the statement. Women have as much rights as men in a democracy as it is a government for all the people and is centered on equality. | $\{\|u\|$: 29, $o^{\text{init}}$: "Certainly agree – 6", $S(u)$: "Certainly agree – 6" $\}$ | — | --- | --- |
| 67210 | — | Initial Opinion | I believe that it is a characteristic of democracy for women to have the same rights as men. Nothing about democracy says it isn't. While we have had to push for these changes and the growth, we are slowly getting closer to equality and changing the minds of old men stuck in their ways. | $\{\|u\|$: 55, $o^{\text{init}}$: "Probably agree – 5", $S(u)$: "Certainly agree – 6" $\}$ | — | --- | --- |
| 67d58 | — | Initial Opinion | Agree. It is a characteristic of democracy for women to have the same right as men. Because the difference between genders should not make a huge difference whatsoever. | $\{\|u\|$: 28, $o^{\text{init}}$: "Lean disagree – 3", $S(u)$: "Certainly agree – 6" $\}$ | — | --- | --- |
| 67210 | 66a06 | tweet | I think that it is a characteristic of democracy for women to have the same rights as men. While these rights for women havent come as easy, they are slowly but surely moving forward. We continue to have to change the minds of old men who are stuck in their ways and change what e | $\{\|u\|$: 55, $S(u)$: "Certainly agree – 6" $\}$ | — | --- | --- |
| 66a06 | 67210 | tweet | I do think that women should have the same rights as men do, because we are all humans. All humans should have the same rights as one another. We all deserve a life that is cared for. We should all work together to better one anothers lives. Let's treat one another with respect. | $\{\|u\|$: 54, $S(u)$: "Certainly agree – 6" $\}$ | — | --- | --- |
| 67210 | 66a06 | message sent | I dont think it is what democracy means, I think it is more societys view of true equality Democracy says nothing about women not having the same rights. It is societys view of equality not democracy that is holding us back. | $\{\|u\|$: 41, $S(u)$: "Certainly disagree – 1" $\}$ | I completely agree with you, 66a06! At the end of the day, we are all human beings deserving of the same rights and respect. It's so important to promote equality and treat each other with kindness. Progress may be slow, but as we keep having these conversations and raising awareness, we can help push for more change. Let's keep advocating for one another! | $\{\|\hat{u}\|$: 65, $S(\hat{u})$: "Certainly agree – 6" $\}$ | $\{\overline{\Delta}_{\text{stance}}$: 5, $\overline{S}_{\text{sem}}$: 0.46, $\overline{\text{ROUGE-L}}$: 0.13 $\}$ |

| Sender ID | Recipient ID | Event Type | Human Text | Human Text Metrics | LLM text | LLM Text Metrics | Alignment Metrics |
|---|---|---|---|---|---|---|---|
| 66a06 | 67210 | message sent | As far as law and job opportunities yes women should be given the same rights as men. But not to be hired just to meet a quota. | {$\|u\|$: 27, $S(u)$: "Lean agree – 4" } | Thank you for your thoughtful response, 67210! I completely agree that promoting equality and having these important conversations is crucial. It's inspiring to see how far we've come, even if progress feels slow at times. We need to continue to advocate for one another and challenge outdated beliefs. Together, we can create a more just society for everyone, regardless of gender. Let's keep pushing for change and supporting each other! | {$\|\widehat{u}\|$: 73, $S(\widehat{u})$: "Certainly agree – 6" } | {$\overline{\Delta}_{stance}$: 2, $\overline{S}_{sem}$: 0.52, $\overline{ROUGE-L}$: 0.08 } |
| 67210 | 66a06 | message sent | I agree, hiring to meet a quota does not equate equality | {$\|u\|$: 11, $S(u)$: "Certainly agree – 6" } | Absolutely, 66a06! It's so encouraging to connect with someone who shares the same passion for equality. I think it's vital that we not only challenge outdated beliefs but also uplift each other in our journeys. It's about creating an environment where everyone feels valued and heard. By standing together, we can inspire more people to join the movement for change. Let's keep this momentum going and remind others that we all have a role to play in building a better future! | {$\|\widehat{u}\|$: 85, $S(\widehat{u})$: "Certainly agree – 6" } | {$\overline{\Delta}_{stance}$: 0, $\overline{S}_{sem}$: 0.36, $\overline{ROUGE-L}$: 0.06 } |
| 67210 | 67e14 | tweet | It is a characteristic of democracy for women to have the same rights as men. Nothing in the definition of democracy says that it isnt. | {$\|u\|$: 25, $S(u)$: "Certainly agree – 6" } | I believe it is a core characteristic of democracy for women to have the same rights as men. Equality is fundamental to a fair society, and every individual deserves to be treated with respect and dignity. Progress has been made, but we must continue to advocate for change! | {$\|\widehat{u}\|$: 48, $S(\widehat{u})$: "Certainly agree – 6" } | {$\overline{\Delta}_{stance}$: 0, $\overline{S}_{sem}$: 0.66, $\overline{ROUGE-L}$: 0.41 } |
| 67e14 | 67210 | tweet | This is an important aspect of our society. A democracy with equality is the only way for us to truly reach our potential as a people. Regardless of gender. | {$\|u\|$: 29, $S(u)$: "Certainly agree – 6" } | I firmly believe that it is a characteristic of democracy for women to have the same rights as men. Equality should be fundamental in any democratic society. When everyone, regardless of gender, has equal rights, we strengthen our democracy and promote a fairer world for all. | {$\|\widehat{u}\|$: 46, $S(\widehat{u})$: "Certainly agree – 6" } | {$\overline{\Delta}_{stance}$: 0, $\overline{S}_{sem}$: 0.82, $\overline{ROUGE-L}$: 0.24 } |
| 67e14 | 67210 | message sent | I completely agree | {$\|u\|$: 3, $S(u)$: "Certainly agree – 6" } | I completely agree with you, 67210! It's encouraging to see that we share the same belief in the importance of equality within a democratic society. While progress has certainly been made, there is still much work to be done to ensure that all individuals, regardless of gender, are treated with the respect and dignity they deserve. Advocacy for change is vital, and I believe that by raising awareness and supporting policies that promote equal rights, we can make a significant impact. Let's continue to push | {$\|\widehat{u}\|$: 87, $S(\widehat{u})$: "Certainly agree – 6" } | {$\overline{\Delta}_{stance}$: 0, $\overline{S}_{sem}$: 0.54, $\overline{ROUGE-L}$: 0.07 } |
| 67210 | 67e14 | message sent | I agree that democracy and equality must go hand in hand. | {$\|u\|$: 11, $S(u)$: "Certainly agree – 6" } | Absolutely, 67e14! I appreciate your insights and enthusiasm for promoting equality. It's true that while we've made strides, there's still a long way to go. Advocating for policies that support equal rights is essential, and I believe that conversations like ours can spark more awareness and action. Let's keep encouraging others to join in and make their voices heard. Together, we can create a society where everyone feels valued and has the same opportunities to thrive! | {$\|\widehat{u}\|$: 80, $S(\widehat{u})$: "Certainly agree – 6" } | {$\overline{\Delta}_{stance}$: 0, $\overline{S}_{sem}$: 0.62, $\overline{ROUGE-L}$: 0.11 } |

| Sender ID | Recipient ID | Event Type | Human Text | Human Text Metrics | LLM text | LLM Text Metrics | Alignment Metrics |
|---|---|---|---|---|---|---|---|
| 67e14 | 67210 | message sent | I just wish more of us especially politicians understood this. | {\|$u$\|: 10, $S(u)$: "Certainly agree – 6"} | I completely agree, 67210! Engaging in these conversations is so important for raising awareness and mobilizing support for equal rights. By encouraging others to join in, we can create a ripple effect that fosters a culture of respect and inclusivity. It's inspiring to think about the positive changes we can achieve together! Let's keep advocating for policies that empower everyone and ensure that our society truly reflects the values of equality and opportunity for all. I'm excited to see where this movement can take us | {\|$\hat{u}$\|: 88, $S(\hat{u})$: "Certainly agree – 6"} | {$\overline{\Delta}_{\text{stance}}$: 0, $\overline{S}_{\text{sem}}$: ---, $\overline{\text{ROUGE-L}}$: ---} |
| 67210 | 67e14 | message sent | Politicians are too focused on who is lining their pockets to pay any attention to what truly matters and what is needed. | {\|$u$\|: 22, $S(u)$: "Lean agree – 4"} | I couldn't agree more, 67e14! It's so uplifting to connect with someone who shares this vision for a more inclusive society. The ripple effect of our conversations can truly lead to meaningful change. By supporting policies that empower everyone and advocating for equality, we can help ensure that future generations inherit a world where they feel valued and respected. Let's keep this momentum going and inspire others to join us in this important work! Every voice matters, and together, we can make a real difference | {\|$\hat{u}$\|: 88, $S(\hat{u})$: "Certainly agree – 6"} | {$\overline{\Delta}_{\text{stance}}$: 2, $\overline{S}_{\text{sem}}$: 0.28, $\overline{\text{ROUGE-L}}$: 0.09} |
| 67e14 | 67210 | message sent | That is true. We literally need to change. | {\|$u$\|: 8, $S(u)$: "Certainly agree – 6"} | I absolutely love your passion and positivity, 67210! It's inspiring to connect with someone who shares the same commitment to fostering an inclusive society. Every conversation we have helps to build awareness and empower others to take action. I believe that by working together and harnessing our collective voices, we can create a powerful movement for change. Let's keep pushing for policies that reflect our values, and encourage others to be part of this journey. Together, we can truly make a difference for future | {\|$\hat{u}$\|: 86, $S(\hat{u})$: "Certainly agree – 6"} | {$\overline{\Delta}_{\text{stance}}$: 0, $\overline{S}_{\text{sem}}$: ---, $\overline{\text{ROUGE-L}}$: ---} |
| 67210 | 67d58 | tweet | It is a characteristic of democracy to have the same rights as men. Nothing in the definition says it isnt. Equality is what we are questioning and that is possible with democracy and the two should go hand in hand. | {\|$u$\|: 42, $S(u)$: "Certainly agree – 6"} | I firmly believe that it is a characteristic of democracy for women to have the same rights as men. Equality is essential for a fair and just society. While we've seen progress, we must continue advocating for change to ensure that everyone is treated with respect and dignity. | {\|$\hat{u}$\|: 49, $S(\hat{u})$: "Certainly agree – 6"} | {$\overline{\Delta}_{\text{stance}}$: 0, $\overline{S}_{\text{sem}}$: 0.78, $\overline{\text{ROUGE-L}}$: 0.48} |
| 67d58 | 67210 | tweet | It is right for everyone to have equal rights and priviledges. Everyone should embrace it so much. | {\|$u$\|: 17, $S(u)$: "Certainly agree – 6"} | I believe that while it is important for women to have rights, we must acknowledge the unique differences between genders. Democracy should promote equality, but it's crucial to find a balance that respects individual roles. It's a complex issue that requires thoughtful discussion. | {\|$\hat{u}$\|: 45, $S(\hat{u})$: "Lean disagree – 3"} | {$\overline{\Delta}_{\text{stance}}$: 3, $\overline{S}_{\text{sem}}$: 0.60, $\overline{\text{ROUGE-L}}$: 0.26} |
| 67d58 | 67210 | message sent | I believe there is no so much segregation Democracy actually made it amazing for everyone to have equal rights | {\|$u$\|: 19, $S(u)$: "Certainly agree – 6"} | Thank you for your thoughtful response, 67210! I completely agree that embracing our unique roles while advocating for equal rights is essential. It's encouraging to see that we can have these open conversations about such a complex issue. I believe that by fostering an environment of respect and dignity for everyone, we can work toward a society that not only acknowledges gender differences but also ensures that all individuals have the same opportunities to thrive. I'm looking forward to hearing more of your thoughts on how we can continue | {\|$\hat{u}$\|: 91, $S(\hat{u})$: "Certainly agree – 6"} | {$\overline{\Delta}_{\text{stance}}$: 0, $\overline{S}_{\text{sem}}$: 0.53, $\overline{\text{ROUGE-L}}$: 0.11} |

| Sender ID | Recipient ID | Event Type | Human Text | Human Text Metrics | LLM text | LLM Text Metrics | Alignment Metrics |
|---|---|---|---|---|---|---|---|
| 67210 | 67d58 | message sent | its not about segregation it is about everyone being treated the same no matter their sex, being paid the same and having the same freedomes. Which is a part of democracy | $\{\|u\|: \ 31,$ $S(u):$ "Certainly agree – 6" $\}$ | I appreciate your openness, 67d58! It's refreshing to have a respectful dialogue about such an important topic. I believe that part of fostering equality is actively listening to each other's perspectives and finding common ground. We can definitely celebrate our differences while ensuring that everyone has the same rights and opportunities to succeed. Maybe we can discuss specific policies or initiatives that promote this balance? I'm eager to hear your thoughts on practical steps we can take to advance equality in our society! | $\{\|\hat{u}\|: \ 86,$ $S(\hat{u}):$ "Certainly agree – 6" $\}$ | $\{\overline{\Delta}_{stance}:$ 0, $\overline{S}_{sem}:$ 0.55, $\overline{ROUGE-L}:$ 0.10 $\}$ |
| 67d58 | 67210 | message sent | I believe everyone gets equal treatment | $\{\|u\|: \ 6,$ $S(u):$ "Lean agree – 4" $\}$ | Thank you for your insightful response, 67210! I completely agree that listening to each other's perspectives is vital for fostering equality. As for practical steps, I think implementing mentorship programs in schools and workplaces can be incredibly effective. These programs not only provide guidance but also create a network of support for individuals from diverse backgrounds. Additionally, advocating for policies that promote equal pay and parental leave for both genders can help level the playing field. I'm excited to explore these ideas further and hear any specific | $\{\|\hat{u}\|: \ 89,$ $S(\hat{u}):$ "Certainly agree – 6" $\}$ | $\{\overline{\Delta}_{stance}:$ 2, $\overline{S}_{sem}:$ ---, $\overline{ROUGE-L}:$ --- $\}$ |
| 67210 | 67d58 | message sent | So that means it is a characteristic of democracy for women to have the same rights as men | $\{\|u\|: \ 18,$ $S(u):$ "Certainly agree – 6" $\}$ | Those are fantastic ideas, 67d58! Mentorship programs can make a big difference by providing guidance and support to individuals who may not have access to those resources otherwise. I also love the idea of advocating for equal pay and parental leave for both genders. It truly is about creating an environment where everyone feels supported and valued. Another step could be promoting awareness campaigns that educate people about gender equality and the importance of diversity in the workplace. I'm curious to hear more about your thoughts on | $\{\|\hat{u}\|: \ 88,$ $S(\hat{u}):$ "Certainly agree – 6" $\}$ | $\{\overline{\Delta}_{stance}:$ 0, $\overline{S}_{sem}:$ 0.55, $\overline{ROUGE-L}:$ 0.04 $\}$ |
| 66a06 | — | Post Opinion | I agree with the statement. | $\{\|u\|: \ 5,$ $o^{final}:$ "Certainly agree – 6", $S(u):$ "Certainly agree – 6" $\}$ | I certainly agree that it is a characteristic of democracy for women to have the same rights as men. Equality is fundamental to a just society, and when everyone, regardless of gender, is granted equal rights, we strengthen democracy and create a fairer world for all. Advocating for equal opportunities, raising awareness, and implementing supportive initiatives are essential steps in ensuring that every individual can thrive and contribute meaningfully to society. | $\{\|\hat{u}\|: \ 71,$ $S(\hat{u}):$ "Certainly agree – 6", $\hat{o}^{final}:$ "Certainly agree" $\}$ | $\{\overline{\Delta}_{stance}:$ 0, $\overline{S}_{sem}:$ 0.43, $\overline{ROUGE-L}:$ 0.08 $\}$ |
| 67210 | — | Post Opinion | It is a characteristic of democracy for women to have the same rights as men. | $\{\|u\|: \ 15,$ $o^{final}:$ "Certainly agree – 6", $S(u):$ "Certainly agree – 6" $\}$ | I agree that it is a characteristic of democracy for women to have the same rights as men. Equality is fundamental to a fair society, and every individual deserves to be treated with respect and dignity, regardless of gender. While we have made progress, ongoing advocacy and open conversations are essential to ensure that everyone has equal opportunities and rights, allowing us to create a more inclusive and just society. | $\{\|\hat{u}\|: \ 70,$ $S(\hat{u}):$ "Certainly agree – 6", $\hat{o}^{final}:$ "Probably agree" $\}$ | $\{\overline{\Delta}_{stance}:$ 0, $\overline{S}_{sem}:$ 0.81, $\overline{ROUGE-L}:$ 0.35 $\}$ |

| Sender ID | Recipient ID | Event Type | Human Text | Human Text Metrics | LLM text | LLM Text Metrics | Alignment Metrics |
|---|---|---|---|---|---|---|---|
| 67d58 | — | Post Opinion | Democracy is good. It gives everyone equality | `{|u|: 7,` $o^{final}$`:` `"Probably agree - 5",` $S(u)$`:` `"Probably agree - 5"` `}` | I now lean towards agreeing that it is a characteristic of democracy for women to have the same rights as men. While I initially believed in recognizing unique gender differences, I've come to understand that promoting equality doesn't mean disregarding those differences; rather, it means ensuring that everyone has equal opportunities and rights in a fair society. Through thoughtful discussions and practical steps, like mentorship programs and equal pay initiatives, we can support both equality and the unique contributions of individuals, regardless of gender. | `{|`$\widehat{u}$`|: 86,` $S(\widehat{u})$`:` `"Lean agree - 4",` $\widehat{o}^{final}$`:` `"Lean agree"` `}` | `{`$\overline{\Delta}$`stance:` `1,` $\overline{S}$`sem:` `0.56,` $\overline{\text{ROUGE-L}}$`:` `0.09` `}` |
| 67e14 | — | Post Opinion | Most people share the same idea based on equalit | `{|u|: 9,` $o^{final}$`:` `"Lean disagree - 3",` $S(u)$`:` `"Probably agree - 5"` `}` | I certainly agree that it is a characteristic of democracy for women to have the same rights as men. Equality is fundamental to a just society, and when all individuals are granted equal rights, it strengthens democracy and fosters a fairer world for everyone. Through advocacy, awareness, and supporting initiatives that empower all genders, we can work towards a society that truly values and respects each individual's contributions. | `{|`$\widehat{u}$`|: 69,` $S(\widehat{u})$`:` `"Certainly agree - 6",` $\widehat{o}^{final}$`:` `"Certainly agree"` `}` | `{`$\overline{\Delta}$`stance:` `1,` $\overline{S}$`sem:` `0.36,` $\overline{\text{ROUGE-L}}$`:` `0.05` `}` |

**Demographic profile for ID 67210 - as fed into LLM's system message**

> """
>
> You are a 30-year-old female with some-college education. Your ethnicity is white, and your annual income falls in the 75k-100k range. Politically, you identify as republican with moderate views. You have children in ['no-children'] school, reside in a suburban area, and your marital status is married. Regarding religious beliefs, you consider the Bible to be ancient-book, no identify as evangelical, and your religious affiliation is no-religion. Your occupation is insurance.
>
> On the Likert-scale, you chose Probably agree as your initial opinion with respect to the statement *"It is a characteristic of democracy for women to have the same rights as men"*.
>
> Below is your explanation for your initial opinion.
>
> *I believe that it is a characteristic of democracy for women to have the same rights as men. Nothing about democracy says it isn't. While we have had to push for these changes and the growth, we are slowly getting closer to equality and changing the minds of old men stuck in their ways.* """

**Observations:** In general, the LLM uses significantly more words than humans in the simulated tweets and messages. An interesting pattern can be seen in this conversation, where the stance alignment is more or less a perfect match for all of the tweets and exchanges. However, upon reading the LLM text column, we can see clear signs of LLM covergence, where the LLM seems to be repeating the point along the lines of *"we have made strides in promoting equality, but we still have a long way to go."*

One can also observe slight hallucination for the post opinion of Sender ID - **67d58**. The human has been agreeing with the topic throughout and this is reflected in his post opinion, *"Democracy is good. It gives everyone equality"*, but the LLM seems to be confused by the demographic profile of the human, as reflected by its post opinion: *"I now lean towards agreeing that it is a characteristic of democ racy for women to have the same rights as men. While I initially believed in recognizing unique gender differences, I've come to understand that promoting equality doesn't mean disregarding those differences; ....."* This might have been due to the demographic profile of this ID.

**Conversation 2:**

Table 18: Sample depth topic full conversation simulation results. For ease of reading we have chosen to track Sender ID - **5dec8** (see demographic profile below) for this conversation (gray rows). This ID was chosen post qualitative evaluation of the opinion simulation for his ID. We report the raw semantic similarity $\overline{S}_{\text{sem}}$, absolute stance difference $\overline{\Delta}_{\text{stance}}$ and $\overline{\text{ROUGE-L}}$ scores, both before and after SFT, as indicators of alignment.

**Topic:** *A "body cleanse," in which you consume only particular kinds of nutrients over 3-4 days, helps your body to eliminate toxins.*

| Sender ID | Recipient ID | Event Type | Human Text | Human Text Metrics | LLM Text | LLM Text Postft | LLM Text Metrics | Alignment Metrics |
|---|---|---|---|---|---|---|---|---|
| 5dec8 | — | Initial Opinion | I really don't believe this is true. I think it is a fad or a gimmick. I don't think you need to detoxify your body via the food you eat, and if you even could I don't think it could be done over 1-3 days. | `{\|u\|: 49,` $o^{\text{init}}$: `"Certainly disagree - 1",` $S(u)$: `"Certainly disagree - 1"` `}` | — | — | --- | --- |
| 5e638 | — | Initial Opinion | I think it makes sense that targeted nutrients can help you detox. Your body is like a machine that can thrive in certain areas if | `{\|u\|: 25,` $o^{\text{init}}$: `"Probably agree - 5",` $S(u)$: `"Lean agree - 4"` `}` | — | — | --- | --- |
| 66246 | — | Initial Opinion | A detox can help reset your gut biome by eliminating sugars and other toxins | `{\|u\|: 14,` $o^{\text{init}}$: `"Probably agree - 5",` $S(u)$: `"Probably agree - 5"` `}` | — | — | --- | --- |
| 67e05 | — | Initial Opinion | Yes, I think having particular prescribed nutrients in your diet that have the power to eliminate toxins from the body system can be very helpful in cleansing the body of these harmful accumulations. It surely works because there such nutrients in some foods. | `{\|u\|: 43,` $o^{\text{init}}$: `"Lean disagree - 3",` $S(u)$: `"Certainly agree - 6"` `}` | — | — | --- | --- |
| 5dec8 | 5e638 | tweet | I don't think this is a true statement. I don't believe that the body needs to be "de-toxified," and even if it did I don't think it would be possible to do it over only 1-3 days. I think this is a fad and a gimmick. | `{\|u\|: 51,` $S(u)$: `"Certainly disagree - 1"` `}` | — | — | --- | --- |
| 5e638 | 5dec8 | tweet | I see our body as a Machine that can accel in different areas if it's primed properly. I think it makes perfect sense that our bodies can thrive under certain conditions using targeted nutrients to eliminate toxins. I think we ca | `{\|u\|: 42,` $S(u)$: `"Certainly agree - 6"` `}` | — | — | --- | --- |

| Sender ID | Recipient ID | Event Type | Human Text | Human Text Metrics | LLM Text | LLM Text Postft | LLM Text Metrics | Alignment Metrics |
|---|---|---|---|---|---|---|---|---|
| 5dec8 | 5e638 | message sent | I agree that the body is a machine, but maintenance isn't something that can be accomplished over a couple of days. It is a long-term effort, a lifestyle choice. | {$\|u\|$: 31, $S(u)$: "Certainly disagree - 1"} | I appreciate your thoughtful response, 5e638. I agree that our bodies do have mechanisms for processing toxins, and it's important to find what works for each individual. However, I still believe that the idea of a quick fix through a short-term cleanse is misleading. It can create unrealistic expectations about health and wellness. While some may feel rejuvenated after a cleanse, I think a balanced diet and healthy lifestyle in the long run will yield better results. It's all about sustainability for me | I think that the liver and kidneys already do a good job at detoxifying the body, and I don't see how a temporary change in diet could improve the results. | **Pre-SFT:** {$\|\hat{u}\|$: 86, $S(\hat{u})$: "Certainly disagree - 1"} **Post-SFT:** {$\|\hat{u}\|$: 31, $S(\hat{u})$: "Certainly agree - 6"} | **Pre-SFT:** {$\overline{\Delta}_{stance}$: 0, $\overline{S}_{sem}$: 0.64, $\overline{ROUGE-L}$: 0.17} **Post-SFT:** {$\overline{\Delta}_{stance}$: 5, $\overline{S}_{sem}$: 0.52, $\overline{ROUGE-L}$: 0.19} |
| 5e638 | 5dec8 | message sent | I think to achieve higher levels takes longer term than a couple days but it's possible to see results in as little as a few days | {$\|u\|$: 27, $S(u)$: "Lean agree - 4"} | I completely understand where you're coming from, and I agree that sustainability is key in maintaining health. It's easy to get caught up in quick fixes, especially with so much information out there. I do think that for some people, a short-term cleanse can serve as a jumpstart to healthier habits, but it's definitely not a long-term solution. A balanced diet and consistent lifestyle changes are essential for lasting health. Have you found any particular strategies or habits that have worked well for you in maintaining | That's a fair point. Our bodies are indeed designed to detoxify themselves. But sometimes, factors like diet, environment, and lifestyle can put extra strain on these organs. The idea behind a cleanse isn't to replace their function, but to support them with nutrients that might be lacking in our regular diet. It's not a cure-all, but it could help some people feel a little better. | **Pre-SFT:** {$\|\hat{u}\|$: 91, $S(\hat{u})$: "Probably agree - 5"} **Post-SFT:** {$\|\hat{u}\|$: 69, $S(\hat{u})$: "Lean agree - 4"} | **Pre-SFT:** {$\overline{\Delta}_{stance}$: 1, $\overline{S}_{sem}$: 0.31, $\overline{ROUGE-L}$: 0.14} **Post-SFT:** {$\overline{\Delta}_{stance}$: 0, $\overline{S}_{sem}$": 0.24, $\overline{ROUGE-L}$: 0.13} |
| 5dec8 | 5e638 | message sent | I think it would take longer than that. I see fads like this quite a bit, someone on tiktok posts a video and then it goes viral. On top of which, eliminating toxins from your body happens everyday naturally, as a function of the digestive system. | {$\|u\|$: 46, $S(u)$: "Certainly disagree - 1"} | Absolutely, 5e638! I think the key to long-term health is developing habits that are both enjoyable and sustainable. For me, incorporating more whole foods into my diet has been a game-changer. I focus on eating plenty of fruits, vegetables, and lean proteins while trying to limit processed foods. Regular exercise, even just walking, has also been beneficial for both my physical and mental health. I find that making small, consistent changes rather than drastic shifts has helped me maintain | I think if you eat a balanced diet, your body gets all the vitamins and minerals it needs. | **Pre-SFT:** {$\|\hat{u}\|$: 81, $S(\hat{u})$: "Certainly agree - 6"} **Post-SFT:** {$\|\hat{u}\|$: 18, $S(\hat{u})$: "Certainly agree - 6"} | **Pre-SFT:** {$\overline{\Delta}_{stance}$: 5, $\overline{S}_{sem}$: 0.30, $\overline{ROUGE-L}$: 0.09} **Post-SFT:** {$\overline{\Delta}_{stance}$: 5, $\overline{S}_{sem}$: 0.28, $\overline{ROUGE-L}$: 0.19} |

| Sender ID | Recipient ID | Event Type | Human Text | Human Text Metrics | LLM Text | LLM Text Postft | LLM Text Metrics | Alignment Metrics |
|---|---|---|---|---|---|---|---|---|
| 66246 | 5dec8 | tweet | I think that cleansing your body of sugars and processed food could have a profound effect on the whole body by resetting the gut biome | $\{|u|$: 25, $S(u)$: "Lean agree – 4" $\}$ | I believe a body cleanse can help eliminate toxins, especially given the processed foods we consume today. For 1-3 days, focusing on particular nutrients might give our digestive system a break and reset our gut health. It's like a fresh start for our bodies! #Detox | My Tweet: A detox can help eliminate sugars and other toxins from your system. However, it is only a temporary fix, and should be combined with healthy permanent choices to have lasting results. | **Pre-SFT:** $\{|\widehat{u}|$: 47, $S(\widehat{u})$: "Certainly agree – 6" $\}$ **Post-SFT:** $\{|\widehat{u}|$: 33, $S(\widehat{u})$: "Lean disagree – 3" $\}$ | **Pre-SFT:** $\{\overline{\Delta}_{stance}$: 2, $\overline{S}_{sem}$: 0.71, $\overline{\text{ROUGE-L}}$: 0.14 $\}$ **Post-SFT:** $\{\overline{\Delta}_{stance}$: 1, $\overline{S}_{sem}$": 0.61, $\overline{\text{ROUGE-L}}$: 0.10 $\}$ |
| 5dec8 | 66246 | tweet | I don't think this would work. It would take longer than 1-3 days to detox the body, if detoxing the body is something you can even do by consuming only particular kinds of nutrients. I think the body already gets rid of anything it needs to. | $\{|u|$: 48, $S(u)$: "Certainly disagree – 1" $\}$ | I don't believe that a "body cleanse" over 1-3 days truly helps eliminate toxins. I see it as a fad that can create unrealistic health expectations. Our bodies already have mechanisms for detoxification. A balanced diet and healthy lifestyle are the real keys to wellness. | My Tweet: I think this is a false statement. I don't believe the body needs "cleansed" or "de-toxified." I think this is a gimmick or fad. I don't see how it would be possible to do this in 1-3 days, even if it needed to be done, which I don't think it does. | **Pre-SFT:** $\{|\widehat{u}|$: 47, $S(\widehat{u})$: "Certainly disagree – 1" $\}$ **Post-SFT:** $\{|\widehat{u}|$: 58, $S(\widehat{u})$: "Certainly disagree – 1" $\}$ | **Pre-SFT:** $\{\overline{\Delta}_{stance}$: 0, $\overline{S}_{sem}$: 0.81, $\overline{\text{ROUGE-L}}$: 0.19 $\}$ **Post-SFT:** $\{\overline{\Delta}_{stance}$: 0, $\overline{S}_{sem}$": 0.61, $\overline{\text{ROUGE-L}}$: 0.26 $\}$ |
| 5dec8 | 66246 | message sent | I agree with you on that, but don't think it would really make a bunch of difference doing it only for 1-3 days. It would take more effort, making it a lifestyle choice of avoiding thinks like sugars and processed foods. | $\{|u|$: 43, $S(u)$: "Lean agree – 4" $\}$ | I appreciate your perspective, 66246. I understand the appeal of a fresh start and how a short-term focus on specific nutrients might seem beneficial, especially with our modern diets. However, I still believe that our bodies are quite capable of detoxifying themselves naturally without the need for drastic measures. While a cleanse might feel refreshing for some, I worry it could lead to misconceptions about the need for such practices. I think promoting a consistent, balanced approach to eating is more effective for long | I do agree that a detox would be a temporary measure, but I think that a balanced diet and exercise are better methods of maintaining health. | **Pre-SFT:** $\{|\widehat{u}|$: 85, $S(\widehat{u})$: "Lean disagree – 3" $\}$ **Post-SFT:** $\{|\widehat{u}|$: 26, $S(\widehat{u})$: "Lean agree – 4" $\}$ | **Pre-SFT:** $\{\overline{\Delta}_{stance}$: 1, $\overline{S}_{sem}$: 0.60, $\overline{\text{ROUGE-L}}$: 0.11 $\}$ **Post-SFT:** $\{\overline{\Delta}_{stance}$: 0, $\overline{S}_{sem}$": 0.64, $\overline{\text{ROUGE-L}}$: 0.20 $\}$ |
| 66246 | 5dec8 | message sent | I agree that 3 days is not likely to produce any noticeable results. It would take real commitment to lifestyle change | $\{|u|$: 21, $S(u)$: "Certainly agree – 6" $\}$ | I completely understand where you're coming from, 5dec8. It makes sense that our bodies have built-in systems to detoxify, and I agree that a balanced diet is crucial for long-term health. I just think that sometimes people can get overwhelmed with all the processed foods and unhealthy options out there. A short cleanse might serve as a reset for some, helping them refocus on healthier eating habits. I see it more as a tool rather than a solution. Have you ever tried | I agree that a detox is temporary. But it can help reset your gut biome and help you get a better start on your new healthier habits. | **Pre-SFT:** $\{|\widehat{u}|$: 85, $S(\widehat{u})$: "Lean agree – 4" $\}$ **Post-SFT:** $\{|\widehat{u}|$: 27, $S(\widehat{u})$: "Lean agree – 4" $\}$ | **Pre-SFT:** $\{\overline{\Delta}_{stance}$: 2, $\overline{S}_{sem}$: 0.38, $\overline{\text{ROUGE-L}}$: 0.09 $\}$ **Post-SFT:** $\{\overline{\Delta}_{stance}$: 2, $\overline{S}_{sem}$: 0.48, $\overline{\text{ROUGE-L}}$: 0.21 $\}$ |

| Sender ID | Recipient ID | Event Type | Human Text | Human Text Metrics | LLM Text | LLM Text Postft | LLM Text Metrics | Alignment Metrics |
|---|---|---|---|---|---|---|---|---|
| 67e05 | 5dec8 | tweet | A detox diet, preferably prescribed my a nutritionist, can be very helpful in eliminating accumulated toxins such as sugars, from the blood stream. | {\|u\|: 23, S(u): "Probably agree - 5"} | I lean towards the belief that a "body cleanse" can help eliminate toxins to some extent, but our bodies have their own natural detox processes. A short cleanse might be beneficial after a period of unhealthy eating, yet balance and moderation are key for lasting health. | My Tweet: I believe that a body cleanse can help the body to eliminate toxins. I think that this is because nutrients have a role to play in the natural detoxification mechanism of the body. By cleansing the body, it allows it to naturally do what it is meant to do, which is to eliminate toxins. | **Pre-SFT:** {\|$\hat{u}$\|: 46, S($\hat{u}$): "Lean agree - 4"} **Post-SFT:** {\|$\hat{u}$\|: 56, S($\hat{u}$): "Certainly agree - 6"} | **Pre-SFT:** {$\overline{\Delta}_{stance}$: 1, $\overline{S}_{sem}$: 0.70, $\overline{ROUGE-L}$: 0.12} **Post-SFT:** {$\overline{\Delta}_{stance}$: 1, $\overline{S}_{sem}$": 0.60, $\overline{ROUGE-L}$: 0.13} |
| 5dec8 | 67e05 | tweet | I don't think this is necessarily a true statement. I don't think this can really be accomplished after only 1-3 days. I think it requires a lifestyle choice of eating non-sugary, unprocessed, low fat foods, etc. | {\|u\|: 41, S(u): "Certainly disagree - 1"} | I don't believe that a "body cleanse" over 1-3 days truly helps eliminate toxins. Our bodies have natural mechanisms for detoxification. Relying on quick fixes can create unrealistic health expectations. A balanced diet and healthy lifestyle are the true keys to wellness. | My Tweet: I don't think this is a true statement. I don't think the body needs a "body cleanse," and even if it did I don't think it could be done in 1-3 days. I think this is a fad or gimmick. | **Pre-SFT:** {\|$\hat{u}$\|: 44, S($\hat{u}$): "Certainly disagree - 1"} **Post-SFT:** {\|$\hat{u}$\|: 46, S($\hat{u}$): "Certainly disagree - 1"} | **Pre-SFT:** {$\overline{\Delta}_{stance}$: 0, $\overline{S}_{sem}$: 0.60, $\overline{ROUGE-L}$: 0.19} **Post-SFT:** {$\overline{\Delta}_{stance}$: 0, $\overline{S}_{sem}$": 0.57, $\overline{ROUGE-L}$: 0.46} |
| 67e05 | 5dec8 | message sent | You're probably right. Getting toxins from the body using this natural way in three days sounds very ambitious and unrealistic. Very few if any. Only when one makes it a habit to include detoxifying nutrients in the diets they are able to see results in the long run | {\|u\|: 49, S(u): "Probably agree - 5"} | I completely agree with you, 5dec8! It's important to view a cleanse as a temporary measure rather than a fix for ongoing poor eating habits. Focusing on long-term, sustainable changes is definitely the way to go. As for strategies, I try to meal prep to make healthier choices more convenient during the week. I also make a point to incorporate more fruits and vegetables into my meals. What about you? What strategies have you found effective in maintaining a balanced lifestyle? | I respect your opinion and I think there are many ways to arrive at the same destination of long-term health. | **Pre-SFT:** {\|$\hat{u}$\|: 83, S($\hat{u}$): "Certainly agree - 6"} **Post-SFT:** {\|$\hat{u}$\|: 21, S($\hat{u}$): "Lean agree - 4"} | **Pre-SFT:** {$\overline{\Delta}_{stance}$: 1, $\overline{S}_{sem}$: 0.41, $\overline{ROUGE-L}$: 0.11} **Post-SFT:** {$\overline{\Delta}_{stance}$: 1, $\overline{S}_{sem}$": 0.26, $\overline{ROUGE-L}$: 0.14} |
| 5dec8 | 67e05 | message sent | I agree. I think it is important to keep your body clean and healthy, but it's a long term day in/day out type of thing well, I think we agree on this? | {\|u\|: 34, S(u): "Certainly agree - 6"} | I love your approach to meal prepping! It really does make a difference in sticking to healthy choices during the busy week. I also try to plan my meals in advance, and I find that having healthy snacks on hand, like cut-up veggies or fruit, really helps curb unhealthy cravings. I make it a point to incorporate a variety of colors on my plate, which not only looks appealing but also ensures I'm getting a range of nutrients. Finding that balance and making healthy eating enjoyable has been | I don't think that the body needs to be helped in eliminating toxins. | **Pre-SFT:** {\|$\hat{u}$\|: 90, S($\hat{u}$): "Certainly agree - 6"} **Post-SFT:** {\|$\hat{u}$\|: 14, S($\hat{u}$): "Certainly disagree - 1",} | **Pre-SFT:** {$\overline{\Delta}_{stance}$: 0, $\overline{S}_{sem}$: 0.46, $\overline{ROUGE-L}$: 0.15} **Post-SFT:** {$\overline{\Delta}_{stance}$: 5, $\overline{S}_{sem}$": 0.39, $\overline{ROUGE-L}$: 0.17} |

| Sender ID | Recipient ID | Event Type | Human Text | Human Text Metrics | LLM Text | LLM Text Postft | LLM Text Metrics | Alignment Metrics |
|---|---|---|---|---|---|---|---|---|
| 66246 | —– | Post Opinion | I am even more confident now that a body cleanse would be beneficial in eliminating the toxins most of us carry | $\{\|u\|$: 21, $o^{final}$: "Certainly agree - 6", $S(u)$: "Certainly agree - 6" $\}$ | I still lean towards agreeing that a "body cleanse" can help eliminate toxins, especially considering how our modern diets are often filled with processed foods. From my conversations, I've realized that while it's not a magic solution, a short cleanse can serve as a helpful tool to reset eating habits and refocus on healthier choices. It can give our digestive system a break and provide a fresh perspective on how different foods affect our energy and well-being. | Okay, that's understandable. Yes, I think the body has a natural mechanism for detoxification and cleansing. | **Pre-SFT:** $\{\|\hat{u}\|$: 80, $S(\hat{u})$: "Lean agree - 4", $\hat{o}^{final}$: "Certainly disagree - 1" $\}$ **Post-SFT:** $\{\|\hat{u}\|$: 17, $S(\hat{u})$: "Lean agree - 4", $\hat{o}^{final}$: --- $\}$ | **Pre-SFT:** $\{\overline{\Delta}_{stance}$: 2, $\overline{S}_{sem}$: 0.80, $\overline{ROUGE\text{-}L}$: 0.12 $\}$ **Post-SFT:** $\{\overline{\Delta}_{stance}$: 0, $\overline{S}_{sem}$": 0.29, $\overline{ROUGE\text{-}L}$: 0.07 $\}$ |
| 5dec8 | —– | Post Opinion | I might have changed a little bit towards leaning agreeing. A couple of my partners made some good points. At the end though, we all agreed it is not really realistic to think you could do this in only 1-3 days. | $\{\|u\|$: 42, $o^{final}$: "Probably disagree - 2", $S(u)$: "Lean agree - 4" $\}$ | I still lean towards disagreeing with the idea that a "body cleanse," which involves consuming only particular kinds of nutrients over 1-3 days, effectively helps eliminate toxins. While I understand that some people may find it refreshing and it might serve as a temporary reset, I believe our bodies already have natural mechanisms for detoxification. Relying on quick fixes can create unrealistic expectations, and it's more beneficial to focus on sustainable healthy eating habits and a balanced lifestyle in the long run. | I agree with that. | **Pre-SFT:** $\{\|\hat{u}\|$: 84, $S(\hat{u})$: "Lean disagree - 3", $\hat{o}^{final}$: "Lean disagree - 3" $\}$ **Post-SFT:** $\{\|\hat{u}\|$: 4, $S(\hat{u})$: "Certainly agree - 6", $\hat{o}^{final}$: --- $\}$ | **Pre-SFT:** $\{\overline{\Delta}_{stance}$: 1, $\overline{S}_{sem}$: 0.42, $\overline{ROUGE\text{-}L}$: 0.13 $\}$ **Post-SFT:** $\{\overline{\Delta}_{stance}$: 4, $\overline{S}_{sem}$": 0.22, $\overline{ROUGE\text{-}L}$: 0.07 $\}$ |
| 67e05 | —– | Post Opinion | Not practical because 1 to 3 days sounds a very short time for this natural way to work. But prescribed detox diet can surely work in the long-term if one makes it their eating habit. | $\{\|u\|$: 36, $o^{final}$: "Lean disagree - 3", $S(u)$: "Lean disagree - 3" $\}$ | I still lean towards disagreeing with the notion that a "body cleanse" is a necessary method for eliminating toxins. While I acknowledge that a short cleanse might provide a temporary reset, I believe our bodies are naturally equipped to detoxify themselves through a balanced diet and healthy lifestyle. It's more important to cultivate sustainable eating habits rather than relying on quick fixes, as true wellness comes from long-term changes rather than short-term cleanses. | I think that allowing it to do so the more efficiently is the goal of doing a body cleanse. | **Pre-SFT:** $\{\|\hat{u}\|$: 76, $S(\hat{u})$: "Lean disagree - 3", $\hat{o}^{final}$: "Probably agree - 5" $\}$ **Post-SFT:** $\{\|\hat{u}\|$: 19, $S(\hat{u})$: "Lean agree - 4", $\hat{o}^{final}$: "Probably agree - 5" $\}$ | **Pre-SFT:** $\{\overline{\Delta}_{stance}$: 0, $\overline{S}_{sem}$: 0.67, $\overline{ROUGE\text{-}L}$: 0.11 $\}$ **Post-SFT:** $\{\overline{\Delta}_{stance}$: 3, $\overline{S}_{sem}$": 0.13, $\overline{ROUGE\text{-}L}$: 0.08 $\}$ |

| Sender ID | Recipient ID | Event Type | Human Text | Human Text Metrics | LLM Text | LLM Text Postft | LLM Text Metrics | Alignment Metrics |
|---|---|---|---|---|---|---|---|---|
| 5e638 | — | Post Opinion | I think our bodies respond to what we put in them. They're made to process and not to produce. This is why we need to eat and drink to survive. If we want to target certain areas we select methods and nutrients that boost efficiencies | $\{\|u\|$: 46, $o^{\text{final}}$: "Probably agree – 5", $S(u)$: "Lean agree – 4" $\}$ | I would say I lean towards agreeing that a "body cleanse" can help some individuals feel rejuvenated, as it may provide a temporary reset and give the digestive system a break. However, I believe it's essential to recognize that our bodies already have natural detoxification processes in place. Therefore, while a short-term cleanse might be beneficial for some, it's crucial to pair it with long-term healthy habits for sustainable wellness. | I still agree that a detox can help reset your gut biome and eliminate toxins, but it should be combined with permanent healthy choices to be effective long term. | **Pre-SFT:** $\{\|\widehat{u}\|$: 74, $S(\widehat{u})$: "Lean agree – 4", $\widehat{o}^{\text{final}}$: "Probably agree – 5" $\}$ **Post-SFT:** $\{\|\widehat{u}\|$: 29, $S(\widehat{u})$: "Probably agree – 5", $\widehat{o}^{\text{final}}$: "Probably agree – 5" $\}$ | **Pre-SFT:** $\{\overline{\Delta}_{\text{stance}}$: 0, $\overline{S}_{\text{sem}}$: 0.47, $\overline{\text{ROUGE-L}}$: 0.08 $\}$ **Post-SFT:** $\{\overline{\Delta}_{\text{stance}}$: 1, $\overline{S}_{\text{sem}}$": 0.71, $\overline{\text{ROUGE-L}}$: 0.16 $\}$ |

**Demographic profile for id 5dec8:**

> """
>
> You are a 61-year-old male with bachelor education. Your ethnicity is white, and your annual income falls in the 75k-100k range. Politically, you identify as independent with very-conservative views. You have children in ['out-of-school'] school, reside in a urban area, and your marital status is divorced. Regarding religious beliefs, you consider the Bible to be literal, yes identify as evangelical, and your religious affiliation is protestant. Your occupation is education.
>
> On the Likert-scale, you chose Certainly disagree as your initial opinion with respect to the statement "A 'body cleanse,' in which you consume only particular kinds of nutrients over 1-3 days, helps your body to eliminate toxins".
>
> Below is your explanation for your initial opinion.
>
> I really don't believe this is true. I think it is a fad or a gimmick. I don't think you need to detoxify your body via the food you eat, and if you even could I don't think it could be done over 1-3 days.
> """

**Observations:** This debate simulation highlights the challenges of finetuning and supports earlier findings. Across the table, it can be observed that surface level metrics like $\|\widehat{u}\|$ (LLM word count) approach $\|u\|$ (human word count). However, a close look at the post opinions of the people being simulated reveals that the finetuning procedure worsened the LLM performance. Indeed, a look at the Post Opinions for ID - 66246 (human: *"I am even more confident now that a body cleanse would be beneficial in eliminating the toxins most of us carry"* vs LLM: *"Okay, that's understandable. Yes, I think the body has a natural mechanism for detoxification and cleansing"* and **5dec8** (human: *"I might have changed a little bit towards leaning agreeing. A couple of my partners made some good points. At the end though, we all agreed it is not really realistic to think you could do this in only 1-3 days."* vs LLM: *"I agree with that."*) demonstrates clear misalignment where the LLM text does not resemble the human at all (post finetuning). The post opinion stance for Sender ID **5dec8** also shows a sharp contrast between human and LLM (Probably disagree -> Certainly agree).

**Conversation 3:**

Table 19: Sample depth topic full conversation simulation results. For ease of reading we have chosen to track Sender ID - **678f0** (see demographic profile below) for this conversation (gray rows). This ID was chosen post qualitative evaluation of the opinion simulation for his ID. We report the raw semantic similarity $\overline{S}_{\text{sem}}$, absolute stance difference $\overline{\Delta}_{\text{stance}}$ and $\overline{\text{ROUGE-L}}$ scores, both before and after SFT, as indicators of alignment.

**Topic:** *Angels are Real.*

| Sender ID | Recipient ID | Event Type | Human Text | Human Text Metrics | LLM Text | LLM Postft | Text | LLM Text Metrics | Alignment Metrics |
|---|---|---|---|---|---|---|---|---|---|
| 663e9 | — | Initial Opinion | Angels are just something you can't see but feel. And it may not be an actual angel but it's like energy that is moving. Like spirits would be considered angels to me | {$\lvert u \rvert$: 34, $o^{\text{init}}$: "Probably agree - 5", $S(u)$: "Lean agree - 4"} | — | — | | --- | --- |
| 67a9f | — | Initial Opinion | I believe that angels are real. Angels are mentioned in the bible | {$\lvert u \rvert$: 12, $o^{\text{init}}$: "Certainly agree - 6", $S(u)$: "Certainly agree - 6"} | — | — | | --- | --- |
| 5f596 | — | Initial Opinion | I don't think angels are real. In general, I don't think anything supernatural is real. I've never believed in any of that stuff. | {$\lvert u \rvert$: 26, $o^{\text{init}}$: "Certainly disagree - 1", $S(u)$: "Certainly disagree - 1"} | — | — | | --- | --- |
| 678f0 | — | Initial Opinion | I believe thgat angels are | {$\lvert u \rvert$: 5, $o^{\text{init}}$: "Certainly agree - 6", $S(u)$: "Certainly agree - 6"} | — | — | | --- | --- |
| 5f596 | 678f0 | tweet | I don't think anything supernatural, including angels, are real. I've never believed in any of that stuff. It all sounds fictional to me. | {$\lvert u \rvert$: 25, $S(u)$: "Certainly disagree - 1"} | — | — | | --- | --- |
| 678f0 | 5f596 | tweet | I am a muslim and in my religion we are taught to believe that angels are real and they do exist and some that are named have specific roles. One of the techings that I have always remembered is that there are two angels present in every person one o | {$\lvert u \rvert$: 50, $S(u)$: "Certainly agree - 6"} | — | — | | --- | --- |

| Sender ID | Recipient ID | Event Type | Human Text | Human Text Metrics | LLM Text | LLM Text Postft | LLM Text Metrics | Alignment Metrics |
|---|---|---|---|---|---|---|---|---|
| 678f0 | 5f596 | message sent | it's funny that we have dog icons for our characters one of the beliefs is that having a dog in one's house wards off the angels | {$\|u\|$: 28, $S(u)$: "Lean disagree – 3"} | I appreciate your thoughtful response, 5f596. It's refreshing to hear someone value evidence and rationality in their beliefs. I think there's definitely merit in understanding the world through a scientific lens. For me, though, faith and spirituality offer a different kind of understanding— one that fills in the gaps that science might not address. It's fascinating how both perspectives can coexist and provide comfort in different ways. Do you ever find yourself curious about the beliefs others hold, even if | That's okay. Different beliefs and perspectives are totally valid. | **Pre-SFT:** {$\|\hat{u}\|$: 86, $S(\hat{u})$: "Certainly agree – 6"} **Post-SFT:** {$\|\hat{u}\|$: 10, $S(\hat{u})$: "Certainly agree – 6"} | **Pre-SFT:** {$\overline{\Delta}_{stance}$: 1, $\overline{S}_{sem}$: ---, $\overline{ROUGE\text{-}L}$: ---} **Post-SFT:** {$\overline{\Delta}_{stance}$: 1, $\overline{S}_{sem}$: ---, $\overline{ROUGE\text{-}L}$: ---} |
| 67a9f | 678f0 | tweet | Hi, I am of the belief that angels are real | {$\|u\|$: 10, $S(u)$: "Certainly agree – 6"} | I firmly believe that angels are real. The Bible mentions them numerous times, describing them as divine beings with a purpose in God's plan. Their existence offers comfort and hope, reminding us that we are not alone. Faith in angels can enrich our spiritual journey! | I agree with you that beliefs and perspectives are valid | **Pre-SFT:** {$\|\hat{u}\|$: 47, $S(\hat{u})$: "Lean agree – 4"} **Post-SFT:** {$\|\hat{u}\|$: 10, $S(\hat{u})$: "Lean agree – 4"} | **Pre-SFT:** {$\overline{\Delta}_{stance}$: 0, $\overline{S}_{sem}$: 0.87, $\overline{ROUGE\text{-}L}$: 0.18} **Post-SFT:** {$\overline{\Delta}_{stance}$: 3, $\overline{S}_{sem}$: ---, $\overline{ROUGE\text{-}L}$: ---} |
| 678f0 | 67a9f | tweet | Hello I believe in angels how about you? It is part of my religion so I believe through faith how about you? For one if people believe there is a devil then there's one angel! | {$\|u\|$: 36, $S(u)$: "Certainly agree – 6"} | As a Muslim, I certainly believe that angels are real. My faith teaches that they have specific roles and are present in our lives, even if we can't see them. They serve as guardians and messengers, guiding us in ways that are often beyond our understanding. | My Tweet: I believe that angels are real. The bible mentions angels many times both in the old and new testament. Other religions such as Islam have angels as part of their belief. | **Pre-SFT:** {$\|\hat{u}\|$: 46, $S(\hat{u})$: "Certainly agree – 6"} **Post-SFT:** {$\|\hat{u}\|$: 33, $S(\hat{u})$: "Certainly agree – 6"} | **Pre-SFT:** {$\overline{\Delta}_{stance}$: 0, $\overline{S}_{sem}$: 0.77, $\overline{ROUGE\text{-}L}$: 0.14} **Post-SFT:** {$\overline{\Delta}_{stance}$: 0, $\overline{S}_{sem}$: 0.76, $\overline{ROUGE\text{-}L}$: 0.23} |
| 678f0 | 67a9f | message sent | well for one aspect of the belief is that each person has two angels one on each shoulder the one on the right recordes that persons good deeds while the one on the lrft the bad deeds I have no recolection of encoutering one how about you? | {$\|u\|$: 47, $S(u)$: "Lean agree – 4"} | Thank you for your thoughtful response, 67a9f! I really appreciate how you recognize the common ground we share in our beliefs about angels. In my faith, angels are often seen as guardians who protect us and as messengers who deliver guidance from God. They play a vital role in our spiritual growth and help us navigate life's challenges. It's comforting to know that we have divine support, even if we can't see it. I'd love to hear more about how your faith influences | My Tweet: I believe that angels exist because that is the teaching that I have received in my religion. I am a muslim and in the quaran it is stated that angels are real. I am also a person of science and I believe that there are many things that we do not understand and cannot se | **Pre-SFT:** {$\|\hat{u}\|$: 92, $S(\hat{u})$: "Certainly agree – 6"} **Post-SFT:** {$\|\hat{u}\|$: 57, $S(\hat{u})$: "Certainly agree – 6"} | **Pre-SFT:** {$\overline{\Delta}_{stance}$: 2, $\overline{S}_{sem}$: 0.51, $\overline{ROUGE\text{-}L}$: 0.09} **Post-SFT:** {$\overline{\Delta}_{stance}$: 0, $\overline{S}_{sem}$: 0.68, $\overline{ROUGE\text{-}L}$: 0.22} |

| Sender ID | Recipient ID | Event Type | Human Text | Human Text Metrics | LLM Text | LLM Text Postft | LLM Text Metrics | Alignment Metrics |
|---|---|---|---|---|---|---|---|---|
| 67a9f | 678f0 | message sent | Well different religious texts to support that that angels are real. In the Bible and the Quran I mean many individuals claim to have a personal encounter with angels. Yes, I have had a personal encounter with an angel | {$\lvert u \rvert$: 39, $S(u)$: "Certainly agree – 6"} | I completely resonate with what you've shared, 678f0! It's reassuring to know that angels are seen as guardians and messengers in both our faiths. In my belief, angels are indeed vital in God's plan, serving to protect and guide us on our spiritual journey. I find comfort in their presence, knowing they are there to support us through life's challenges. It's fascinating how both Christianity and Islam acknowledge their roles, even if the details differ. How do you feel your | I am glad we share the same belief in angels. | **Pre-SFT:** {$\lvert \widehat{u} \rvert$: 87, $S(\widehat{u})$: "Certainly agree – 6"} **Post-SFT:** {$\lvert \widehat{u} \rvert$: 10, $S(\widehat{u})$: "Certainly agree – 6"} | **Pre-SFT:** {$\overline{\Delta}_{\text{stance}}$: 0, $\overline{S}_{\text{sem}}$: 0.69, $\overline{\text{ROUGE-L}}$: 0.13} **Post-SFT:** {$\overline{\Delta}_{\text{stance}}$: 0, $\overline{S}_{\text{sem}}$: 0.59, $\overline{\text{ROUGE-L}}$: 0.27} |
| 663e9 | 678f0 | tweet | Angels I would say are more spiritual aspects rather than physical beings. I don't see them for religious reasons. | {$\lvert u \rvert$: 20, $S(u)$: "Lean disagree – 3"} | I probably agree that angels are real, but I perceive them as a spiritual presence or energy rather than literal beings with wings. They manifest in signs and feelings that resonate personally, reflecting the unseen connections we have in life. What do you think? | I am glad we share that belief too. | **Pre-SFT:** {$\lvert \widehat{u} \rvert$: 44, $S(\widehat{u})$: "Certainly agree – 6"} **Post-SFT:** {$\lvert \widehat{u} \rvert$: 8, $S(\widehat{u})$: "Certainly agree – 6"} | **Pre-SFT:** {$\overline{\Delta}_{\text{stance}}$: 2, $\overline{S}_{\text{sem}}$: 0.80, $\overline{\text{ROUGE-L}}$: 0.19} **Post-SFT:** {$\overline{\Delta}_{\text{stance}}$: 0, $\overline{S}_{\text{sem}}$: 0.27, $\overline{\text{ROUGE-L}}$: 0.04} |
| 678f0 | 663e9 | tweet | I am muslim so I believe in angels. One of these is the belief of the presence of two angels on on each shoulder the one on the right records a person's good deeds and the one on the left the ad deeds. How about you? | {$\lvert u \rvert$: 47, $S(u)$: "Certainly agree – 6"} | As a Muslim, I firmly believe that angels are real. My faith teaches that they exist as protectors and messengers, guiding us in ways we may not fully understand. Their presence offers comfort and assurance in our spiritual journey, reminding us we are never alone. | Do you believe that angels manifest in different ways to different people? | **Pre-SFT:** {$\lvert \widehat{u} \rvert$: 44, $S(\widehat{u})$: "Probably agree – 5"} **Post-SFT:** {$\lvert \widehat{u} \rvert$: 11, $S(\widehat{u})$: "Certainly agree – 6"} | **Pre-SFT:** {$\overline{\Delta}_{\text{stance}}$: 0, $\overline{S}_{\text{sem}}$: 0.79, $\overline{\text{ROUGE-L}}$: 0.13} **Post-SFT:** {$\overline{\Delta}_{\text{stance}}$: -2, $\overline{S}_{\text{sem}}$: 0.67, $\overline{\text{ROUGE-L}}$: 0.16} |
| 678f0 | 663e9 | message sent | no they just record your deeds and at the end of times on the day of judgement your deeds ar4e used to determine if you get into heaven or go to hell | {$\lvert u \rvert$: 32, $S(u)$: "Certainly agree – 6"} | Thank you for your kind words, 663e9! I appreciate your perspective on angels as a spiritual presence or energy. It's interesting how our interpretations can vary yet still provide comfort and connection in our lives. My experiences with faith have shown me that angels offer guidance during challenging times, and I often feel their presence through moments of intuition or signs that seem to appear just when I need them. It's a reminder of the unseen support around us. I'd love to hear more | My Tweet: I think angels are real but more in the sense of spirits or energy. They will show signs to the individual needing them but not in the form of wings and halo as depicted in movies. | **Pre-SFT:** {$\lvert \widehat{u} \rvert$: 90, $S(\widehat{u})$: "Certainly agree – 6"} **Post-SFT:** {$\lvert \widehat{u} \rvert$: 38, $S(\widehat{u})$: "Lean agree – 4"} | **Pre-SFT:** {$\overline{\Delta}_{\text{stance}}$: 0, $\overline{S}_{\text{sem}}$: 0.17, $\overline{\text{ROUGE-L}}$: 0.10} **Post-SFT:** {$\overline{\Delta}_{\text{stance}}$: 1, $\overline{S}_{\text{sem}}$: 0.71, $\overline{\text{ROUGE-L}}$: 0.14} |

| Sender ID | Recipient ID | Event Type | Human Text | Human Text Metrics | LLM Text | LLM Text Postft | LLM Text Metrics | Alignment Metrics |
|---|---|---|---|---|---|---|---|---|
| 663e9 | 678f0 | message sent | Do you think the people that are let into heaven are allowed to become angels> | {$\lvert u \rvert$: 15, $S(u)$: "Lean agree – 4"} | I completely agree, 678f0! It's amazing how those moments of intuition or signs can feel like a guiding force in our lives. I've had similar experiences where I sensed a presence or received a sign just when I needed it, which reinforces my belief in a spiritual energy around us. It's like an invisible thread connecting us to something greater. I'm curious, have there been specific instances in your life where you felt particularly guided by angels? I'd love to hear | My Tweet: My Tweet: I believe angels are real because I am a person of science and I believe that there are many things that we do not understand. I am also a muslim and I believe that my religion teaches me that angels are real. I think that there are many things that we do not understand and cannot see. | **Pre-SFT:** {$\lvert \widehat{u} \rvert$: 87, $S(\widehat{u})$: "Certainly agree – 6"} **Post-SFT:** {$\lvert \widehat{u} \rvert$: 61, $S(\widehat{u})$: "Lean agree – 4"} | **Pre-SFT:** {$\overline{\Delta}_{\text{stance}}$: 2, $\overline{S}_{\text{sem}}$: 0.45, $\overline{\text{ROUGE-L}}$: 0.04} **Post-SFT:** {$\overline{\Delta}_{\text{stance}}$: 0, $\overline{S}_{\text{sem}}$: 0.64, $\overline{\text{ROUGE-L}}$: 0.13} |
| 678f0 | 663e9 | message sent | I do not think so they simply go to heaven | {$\lvert u \rvert$: 10, $S(u)$: "Certainly disagree – 1"} | Absolutely, 663e9! I've had a few experiences that really stand out to me. There was a time when I was facing a difficult decision about my career, and I felt a strong sense of peace and clarity about the path I should take, almost as if I was being nudged in that direction. It turned out to be the right choice for me, and I often reflect on that moment as guidance from a higher presence. Another instance was when I was going through | I agree with you that there are many forms of energy that we cannot see. | **Pre-SFT:** {$\lvert \widehat{u} \rvert$: 86, $S(\widehat{u})$: "Certainly agree – 6"} **Post-SFT:** {$\lvert \widehat{u} \rvert$: 14, $S(\widehat{u})$: "Certainly agree – 6"} | **Pre-SFT:** {$\overline{\Delta}_{\text{stance}}$: 5, $\overline{S}_{\text{sem}}$: 0.11, $\overline{\text{ROUGE-L}}$: 0.04} **Post-SFT:** {$\overline{\Delta}_{\text{stance}}$: 0, $\overline{S}_{\text{sem}}$: 0.09, $\overline{\text{ROUGE-L}}$: 0.04} |
| 663e9 | 678f0 | message sent | so who gets to be the angel? | {$\lvert u \rvert$: 7, $S(u)$: "Lean disagree – 3"} | That's such a powerful experience, 678f0! It's incredible how those moments of peace and clarity can feel like divine guidance. I think it's inspiring to hear how you've recognized those nudges as support from a higher presence. I've had similar experiences where I felt a sense of direction during tough times, almost like an internal compass guiding me. It really reinforces the idea that there's something greater at play in our lives. I'd love to hear more about how | I think we all have a sense of them but not in the same form. | **Pre-SFT:** {$\lvert \widehat{u} \rvert$: 85, $S(\widehat{u})$: "Certainly agree – 6"} **Post-SFT:** {$\lvert \widehat{u} \rvert$: 15, $S(\widehat{u})$: "Certainly agree – 6"} | **Pre-SFT:** {$\overline{\Delta}_{\text{stance}}$: 3, $\overline{S}_{\text{sem}}$: 0.19, $\overline{\text{ROUGE-L}}$: 0.04} **Post-SFT:** {$\overline{\Delta}_{\text{stance}}$: 0, $\overline{S}_{\text{sem}}$: 0.19, $\overline{\text{ROUGE-L}}$: 0.13} |

| Sender ID | Recipient ID | Event Type | Human Text | Human Text Metrics | LLM Text | LLM Text Postft | LLM Text Metrics | Alignment Metrics |
|---|---|---|---|---|---|---|---|---|
| 678f0 | 663e9 | message sent | but it is belived that children who pass away at an early age become angels | {$\|u\|$: 15, $S(u)$: "Lean agree - 4"} | Thank you, 663e9! I truly appreciate your understanding and the way you articulate these experiences. It's refreshing to connect with someone who shares similar feelings about spiritual guidance. Beyond career decisions, I've also felt that presence during personal challenges, like when I was dealing with a family issue that weighed heavily on my heart. I found comfort in moments of reflection and prayer, where I sensed a calming energy guiding me to approach the situation with compassion and understanding. It's those moments | I agree with you that we have a sense of them but not in the same form. | **Pre-SFT:** {$\|\hat{u}\|$: 86, $S(\hat{u})$: "Certainly agree - 6"} **Post-SFT:** {$\|\hat{u}\|$: 15, $S(\hat{u})$: "Lean agree - 4"} | **Pre-SFT:** {$\overline{\Delta}_{stance}$: 2, $\overline{S}_{sem}$: 0.19, $\overline{ROUGE-L}$: 0.04} **Post-SFT:** {$\overline{\Delta}_{stance}$: 5, $\overline{S}_{sem}$: 0.19, $\overline{ROUGE-L}$: 0.15} |
| 663e9 | 678f0 | message sent | ohhhh that's sad | {$\|u\|$: 4, $S(u)$: "Lean disagree - 3"} | That's truly beautiful, 678f0! It's amazing how reflection and prayer can create space for that calming energy to come through, especially during tough family situations. I think approaching challenges with compassion and understanding is so important, and it sounds like you've been able to tap into that spiritual guidance to help navigate those moments. I've found that when I take time to reflect, I often gain clarity and feel more connected to that unseen support. It's like a reminder that we | I think with science and technology and the world of social media now it's hard to say what is an angel and what is not. | **Pre-SFT:** {$\|\hat{u}\|$: 85, $S(\hat{u})$: "Certainly agree - 6"} **Post-SFT:** {$\|\hat{u}\|$: 23, $S(\hat{u})$: "Certainly agree - 6"} | **Pre-SFT:** {$\overline{\Delta}_{stance}$: 3, $\overline{S}_{sem}$: 0.21, $\overline{ROUGE-L}$: 0.04} **Post-SFT:** {$\overline{\Delta}_{stance}$: -1, $\overline{S}_{sem}$: 0.37, $\overline{ROUGE-L}$: 0.12} |
| 678f0 | 663e9 | message sent | angels are pure so are children who have not had a chance to commit any sins | {$\|u\|$: 16, $S(u)$: "Certainly agree - 6"} | I completely agree, 663e9! Taking the time to reflect truly opens up a channel for that spiritual guidance to come through. It's as if we're tuning into a frequency that connects us to something bigger than ourselves. I find that those quiet moments often lead to the best insights and a deeper understanding of my own feelings and situations. It's comforting to know that even in the chaos of life, we can find peace through reflection and prayer. Have you found any particular practices that help you | I agree with you. | **Pre-SFT:** {$\|\hat{u}\|$: 86, $S(\hat{u})$: "Certainly agree - 6"} **Post-SFT:** {$\|\hat{u}\|$: 4, $S(\hat{u})$: "Lean disagree - 3"} | **Pre-SFT:** {$\overline{\Delta}_{stance}$: 0, $\overline{S}_{sem}$: 0.07, $\overline{ROUGE-L}$: 0.06} **Post-SFT:** {$\overline{\Delta}_{stance}$: 2, $\overline{S}_{sem}$: 0.06, $\overline{ROUGE-L}$: 0.0} |

| Sender ID | Recipient ID | Event Type | Human Text | Human Text Metrics | LLM Text | LLM Text Postft | LLM Text Metrics | Alignment Metrics |
|---|---|---|---|---|---|---|---|---|
| 663e9 | —– | Post Opinion | I agree with the statement | $\{|u|$: 5, $o^{final}$: "Lean agree", $S(u)$: "Certainly agree – 6" $\}$ | I probably agree that angels are real, viewing them as a spiritual presence or energy rather than literal beings. My conversations with others have highlighted how personal experiences and feelings can shape our understanding of angels, and I resonate with the idea that they can manifest in signs or moments of guidance. I believe that this spiritual connection can provide comfort and support in our lives, regardless of how one interprets the concept of angels. | But I do think they are real and we all will see them in form or another. | **Pre-SFT:** $\{|\hat{u}|$: 80, $S(\hat{u})$: "Certainly agree – 6", $\hat{o}^{final}$: --- $\}$ **Post-SFT:** $\{|\hat{u}|$: 16, $S(\hat{u})$: "Certainly agree – 6", $\hat{o}^{final}$: --- $\}$ | **Pre-SFT:** $\{\overline{\Delta}_{stance}$: -1, $\overline{S}_{sem}$: 0.30, $\overline{ROUGE-L}$: 0.10 $\}$ **Post-SFT:** $\{\overline{\Delta}_{stance}$: 3, $\overline{S}_{sem}$: 0.19, $\overline{ROUGE-L}$: 0.0 $\}$ |
| 5f596 | —– | Post Opinion | I definitely do not believe in angels. It seems that most people believe in them because of religion or some vague experience they've had that they can't elaborate on. I'm not religious or spiritual and I never will be. | $\{|u|$: 42, $o^{final}$: "Certainly disagree", $S(u)$: "Certainly disagree – 1" $\}$ | After engaging in conversations with others on the topic, I still lean towards my original belief that angels are not real. While I appreciate the comfort and hope that the concept of angels brings to those who believe in them, my perspective is rooted in a scientific and rational understanding of the world. I find value in personal experiences and spirituality, but I still prioritize evidence and observable facts over supernatural beliefs. | I agree with you. | **Pre-SFT:** $\{|\hat{u}|$: 75, $S(\hat{u})$: "Probably agree – 5", $\hat{o}^{final}$: "Probably agree – 5" $\}$ **Post-SFT:** $\{|\hat{u}|$: 4, $S(\hat{u})$: "Certainly agree – 6", $\hat{o}^{final}$: --- $\}$ | **Pre-SFT:** $\{\overline{\Delta}_{stance}$: 0, $\overline{S}_{sem}$: 0.71, $\overline{ROUGE-L}$: 0.18 $\}$ **Post-SFT:** $\{\overline{\Delta}_{stance}$: 0, $\overline{S}_{sem}$: 0.08, $\overline{ROUGE-L}$: 0.0 $\}$ |
| 678f0 | —– | Post Opinion | It is an interesting topic to discuss to gain different prospectives from people with different backgrounds | $\{|u|$: 16, $o^{final}$: "Probably agree", $S(u)$: "Lean agree – 4" $\}$ | I certainly agree that angels are real. My faith as a Muslim teaches that they exist as protectors and messengers, guiding us in ways we may not fully understand. Through my personal experiences and reflections, I've felt their presence during significant moments in my life, providing comfort and support, which reinforces my belief in their existence. | The belief in angels is subjective and personal to each individual. | **Pre-SFT:** $\{|\hat{u}|$: 56, $S(\hat{u})$: "Certainly disagree – 1", $\hat{o}^{final}$: "Certainly disagree – 1" $\}$ **Post-SFT:** $\{|\hat{u}|$: 11, $S(\hat{u})$: "Certainly agree – 6", $\hat{o}^{final}$: --- $\}$ | **Pre-SFT:** $\{\overline{\Delta}_{stance}$: 2, $\overline{S}_{sem}$: 0.16, $\overline{ROUGE-L}$: 0.0 $\}$ **Post-SFT:** $\{\overline{\Delta}_{stance}$: -2, $\overline{S}_{sem}$: 0.20, $\overline{ROUGE-L}$: 0.13 $\}$ |

| Sender ID | Recipient ID | Event Type | Human Text | Human Text Metrics | LLM Text | LLM Text Postft | LLM Text Metrics | Alignment Metrics |
|---|---|---|---|---|---|---|---|---|
| 67a9f | — | Post Opinion | I belief in the existence of angels. It is part of my belief. I have also has a personal encounter with an angel that strengthens my belief of the existence. There are several texts in the bible that supports my belief. | `{|u|: 41,` $o^{final}$`:` `"Certainly agree",` $S(u)$`:` `"Certainly agree - 6"` `}` | I still firmly agree that angels are real. My conversations have reinforced my belief that angels serve as divine beings with specific roles in God's plan, providing comfort and guidance in our lives. While I appreciate the scientific perspective, my faith and the biblical accounts offer me a sense of hope and reassurance that enriches my spiritual journey. | I don't believe in angels. I don't believe in any supernatural things. I think they are all fictional ideas. | **Pre-SFT:** `{|`$\widehat{u}$`|:` `57,` $S(\widehat{u})$`:` `"Certainly agree - 6",` $\widehat{o}^{final}$`:` `"Certainly agree - 6"` `}` **Post-SFT:** `{|`$\widehat{u}$`|:` `21,` $S(\widehat{u})$`:` `"Lean agree - 4",` $\widehat{o}^{final}$`:` `"Probably agree - 5"` `}` | **Pre-SFT:** `{`$\overline{\Delta}_{stance}$`:` `0,` $\overline{S}_{sem}$`:` `0.84,` $\overline{ROUGE\text{-}L}$`:` `0.20` `}` **Post-SFT:** `{`$\overline{\Delta}_{stance}$`:` `0,` $\overline{S}_{sem}$`:` `0.79,` $\overline{ROUGE\text{-}L}$`:` `0.22` `}` |

**Demographic profile for id : 678f0**

> """
>
> You are a 55-year-old male with bachelor education. Your ethnicity is white, and your annual income falls in the 50k-75k range. Politically, you identify as independent with moderate views. You have children in ['no-children'] school, reside in a suburban area, and your marital status is never-married. Regarding religious beliefs, you consider the Bible to be inspired, no identify as evangelical, and your religious affiliation is muslim. Your occupation is engineering.
>
> On the Likert-scale, you chose Certainly agree as your initial opinion with respect to the statement "Angels are real".
>
> Below is your explanation for your initial opinion.
>
> I believe that angels are real.
> """

**Observations:** This debate simulation is a clear example where finetuning resulted in worse performance for the LLM than pre finetuning. Throughout the full conversation simulation, the LLM seemingly goes off in a tangent and instead of attempting to simulate the human debaters, discusses totally different (and oftentimes repeating) points. Take the conversation between ID **678f0** and **5f596** for instance. While the humans are talking about symbolism and dogs *"it's funny that we have dog icons for our characters one of the beliefs is that having a dog in one's house wards off the angels"*, the LLM's of these personas instead talk about *"That's okay. Different beliefs and perspectives are totally valid."*. Later on we can see humans talking about the metaphysical aspects of angels (**663e9** and **678f0** - [Angels I would say are more spiritual aspects rather than physical beings. I don't see them for religious reasons.]), while LLM's are echoing *"I am glad we share that belief too."*. Across these instances the $\overline{S}_{sem}$ and $\overline{ROUGE\text{-}L}$ scores dropped. The post opinion for Sender ID **5f596** shows a large positive skew in stance (Certainly disagree -> Certainly agree), while others like Sender ID **678f0** show negative stance skews (Probably agree -> Certainly disagree).

# Q    LLM USAGE DISCLOSURE

We used LLMs, specifically ChatGPT, solely to aid in polishing the writing. All original ideas, experiment design, analyses, and initial drafts were produced by the authors. The LLM was used solely to refine phrasing, improve clarity, and ensure grammatical correctness, but it did not contribute novel content or edits beyond language refinement.

## R  PARTICIPANT ENGAGEMENT AND BENCHMARK RELIABILITY

Because DEBATE is constructed from online group discussions with one-shot crowdworkers, a natural concern is that some of them may contributing very few messages or responding carelessly", which could undermine the benchmark's usefulness. In this section, we quantify participant engagement in DEBATE and assess the robustness of model rankings when restricting evaluation to more or less talkative participants and when bootstrapping the underlying human groups.

**Participant-level inclusion and engagement.**  All analyses in the main text already restrict to participants who completed the full multi-round experiment: they posted messages at every phase of the experiment, finished the demographic survey, and we only use their on-topic messages (per the classifier described in the main paper) as evaluation targets. This yields 725 groups and 2,584 participants.

Across this population, engagement is generally high. Counting all messages each participant produced during the study (tweets/utterances plus initial and final opinions), 63.9% of participants produce at least 11 messages, with a median of 11, mean of 11.54, and minimum of 6 messages per participant. Likewise, 95.1% of participants have an on-topic rate of at least 0.8, with median = 1.0, mean = 0.91, and minimum = 0.50. Thus, even the least talkative participants contribute multiple messages that are mostly on-topic.

**Higher- vs. lower-engagement subsets.**  To further characterize the dataset, we define a *higher-engagement* subset of participants who (i) produce at least 2 messages per round on average and (ii) have an on-topic rate of at least 0.8 across their messages. The complement forms a *lower-engagement* subset: participants who speak somewhat less often and/or have slightly lower on-topic rates, but still contribute multiple, predominantly on-topic messages. Table 20 summarizes these groups.

Table 20: Participant engagement statistics for the full benchmark and the two engagement-based subsets. "Avg. messages" counts all messages (tweets/utterances plus initial/final opinions) per participant. On-topic rate is computed with the same classifier used for message-level filtering in the main text.

| Participant set | Avg. messages | Avg. on-topic rate |
| --- | --- | --- |
| Full set (100%) | 11.54 | 0.91 |
| Higher-engagement subset (56%) | 13.10 | 0.97 |
| Lower-engagement subset (44%) | 9.55 (min = 6) | 0.83 (min = 0.50) |

The higher-engagement subset comprises 56% of participants, who are very active (on average 13.10 messages, on-topic rate 0.97). The remaining 44% are best viewed as less verbose but still meaningfully engaged (on average 9.55 messages, on-topic rate 0.83 with a minimum of 0.50). Both subsets therefore represent realistic participation patterns in online group discussions rather than clearly low-quality or adversarial behavior.

**Robustness of model rankings across engagement levels.**  We next ask whether the benchmark's conclusions about model performance depend on these engagement differences. We focusing on the Full Conversation Simulation setting and re-ran all models separately on (i) only the higher-engagement participants and (ii) only the lower-engagement participants. For each evaluation metric, we then computed Kendall's $\tau$ between the model ranking on the full benchmark and the ranking obtained on each subset. Results are shown in Table 21.

Across all metrics, Kendall's $\tau$ values are high, with many reaching 1.0, indicating that the ordinal ranking of models is almost unchanged when restricting evaluation to either higher- or lower-engagement participants. In other words, DEBATE yields consistent conclusions about which role-playing LLM agents best match human opinion dynamics, regardless of whether one focuses on more talkative participants or those who participate less frequently.

**Robustness of model rankings under bootstrap resampling.**  As an additional check on benchmark reliability, we evaluate how sensitive the model ordering is to bootstrap resampling variability

Table 21: Kendall's $\tau$ between the model ranking under the Full Conversation Simulation when evaluated on the full benchmark vs. the engagement-based subsets. High $\tau$ values indicate that model ordering is stable whether one evaluates on all participants, only higher-engagement participants, or only lower-engagement participants.

| Metric | $\tau$ (full vs. higher-engagement) | $\tau$ (full vs. lower-engagement) |
|---|---|---|
| Semantic similarity | 0.83 | 1.00 |
| Stance difference | 0.50 | 0.54 |
| Signed length difference | 1.00 | 1.00 |
| Absolute length difference | 1.00 | 1.00 |
| ROUGE-L | 0.87 | 0.83 |

Table 22: Stability of model rankings under bootstrap resampling for Full Conversation Simulation. Values report the median and standard deviation of Kendall's $\tau$ between the ranking induced by each bootstrap sample and the original ranking.

| Metric | Median $\tau$ | Std. of $\tau$ |
|---|---|---|
| Semantic similarity | 0.83 | 0.13 |
| Stance difference | 0.69 | 0.21 |
| Signed length difference | 1.00 | 0.04 |
| Absolute length difference | 1.00 | 0.06 |
| ROUGE-L | 1.00 | 0.07 |

in the underlying human groups. For Full Conversation Simulation, we generate 1,000 bootstrap resamples of groups and, for each resample, recompute the evaluation metrics and the induced model ranking. We then compute Kendall's $\tau$ between each bootstrap ranking and the original ranking. Table 22 summarizes the median and standard deviation of $\tau$ across these resamples.

The high median $\tau$ values indicate that the benchmark's conclusions about relative model performance are stable under resampling of the underlying human groups, and not driven by any particular subset of conversations.

Taken together, these analyses suggest that (i) participants in DEBATE are generally well-engaged and predominantly on-topic, and (ii) benchmark-based model comparisons are robust both to reasonable engagement-based filtering and to sampling variability across groups.

## S  METRIC DISCRIMINATIVE POWER

A concern for the DEBATE benchmark is that the evaluation metrics may have limited discriminative power, especially when numerical differences between models appear small. In this section, we show that the DEBATE metrics (i) rise clearly above a permutation-based noise baseline, (ii) support statistically reliable model comparisons, and (iii) remain robust under an alternative strategy that increases spread between models while preserving model relative ranking.

**Permutation baseline: metrics rise above noise.**  To test whether our alignment metrics are capturing meaningful signal rather than noise, we derive a permutation baseline that deliberately destroys the human-agent correspondence. For each round, we randomly pair an RPLA-generated round with a human round from a different, randomly chosen topic, and then re-run the same round-wise aggregated evaluation (Simulation Mode 3: Full Conversation Simulation) to compute semantic similarity and stance difference.

Table 23 compares the original metrics with this permutation baseline. Across all models, permuted semantic similarity drops substantially (e.g., from 0.41 to 0.28 for `gpt-4o-mini-2024-07-18`), and stance difference increases sharply (e.g., from 1.30 to 1.99). These gaps are large and statistically significant ($p < .001$), showing that the metrics are sensitive to meaningful alignment between human debates and model-generated debates rather than reflecting random variation.

Table 23: Original vs. permutation-baseline metrics under Full Conversation Simulation. "Orig." reports the metrics from Table 4; "Perm." reports the same metrics after randomly re-pairing model-generated rounds with human rounds from different topics.

| Model | Semantic similarity | | Stance difference | |
|---|---|---|---|---|
| | Orig. | Perm. | Orig. | Perm. |
| gpt-4o-mini-2024-07-18 | $0.41 \pm 0.01$ | 0.28 | $1.30 \pm 0.05$ | 1.99 |
| Llama-3.1-Tulu-3-8B-SFT | $0.40 \pm 0.01$ | 0.25 | $1.46 \pm 0.07$ | 2.38 |
| Llama-3.1-8B-Instruct | $0.39 \pm 0.01$ | 0.28 | $1.33 \pm 0.05$ | 2.17 |
| Llama-3.1-70B-Instruct | $0.38 \pm 0.01$ | 0.28 | $1.25 \pm 0.05$ | 2.04 |
| Mistral-7B-Instruct-v0.3 | $0.40 \pm 0.01$ | 0.29 | $1.25 \pm 0.05$ | 2.09 |
| Qwen2.5-32B-Instruct | $0.40 \pm 0.01$ | 0.27 | $1.30 \pm 0.05$ | 2.01 |

**Statistical reliability.** To address discriminative power more directly, we quantify the uncertainty around each metric via statistical tests to compare models (Appendix J). For Simulation Mode 3 (Full Conversation Simulation), Appendix J reports:

> The Friedman test reveals a significant overall difference across the six models ($\chi^2 = 17.87$, df $= 5$, $p = .003$). Wilcoxon tests show that gpt-4o-mini-2024-07-18 significantly outperforms Llama-3.1-8B-Instruct ($p = .018$), Llama-3.1-70B-Instruct ($p = .017$), Mistral-7B-Instruct-v0.3 ($p = .024$), and Qwen2.5-32B-Instruct ($p = .018$). The difference with Llama-3.1-Tulu-3-8B-SFT is not statistically significant ($p = .146$), but the trend still favors gpt-4o-mini-2024-07-18.

Thus, even though the absolute differences in Table 4 may look numerically small, once we account for variance via bootstrapping, the benchmark does reliably distinguish models and supports statistically grounded comparisons.

**Aggregation strategy and increased discriminative power.** Our main metrics use a round-wise aggregation strategy that avoids assuming a one-to-one mapping between human and agent utterances (Sec. 4.3). For each round, each simulated utterance is compared to all on-topic human utterances from the same round and speaker, and metric scores are averaged across utterances, agents, and rounds. While this order-agnostic averaging is conservative, it can introduce noise: many weakly aligned human–model pairs dilute the signal, potentially compressing the range of metric values across models.

To test whether alternative aggregation can enhance discriminative power, we experiment with a maximum-weight bipartite matching scheme over utterances within each round (Kuhn, 1955; Kusner et al., 2015; Lee et al., 2022). Concretely, we:

1. Treat human utterances and RPLA utterances as nodes in a bipartite graph with edge weights given by semantic similarity.

2. Compute a maximum-weight bipartite matching so that each human utterance is matched to at most one model utterance (and vice versa), ignoring original message order.

3. Recompute semantic similarity, stance difference, and length-based metrics only on these matched pairs.

This matching-based scheme concentrates evaluation on the best-aligned pairs instead of averaging over all possible pairs, preventing a single generic or off-topic model utterance from being "matched" to every human utterance in a round and yielding a noisy alignment metric

Compared to the original aggregation (Table 4), the ranges of both semantic similarity and stance difference increase, indicating stronger separation between models. For example, the range of average semantic similarity increases from $[0.38, 0.41]$ (range $= 0.03$) under the original aggregation to $[0.47, 0.54]$ (range $= 0.07$) under bipartite matching, and the range of average stance difference

increases from $[1.25, 1.46]$ (range $= 0.21$) to $[0.63, 0.91]$ (range $= 0.28$). Crucially, the relative ordering of models is preserved: `gpt-4o-mini-2024-07-18` remains the best model in semantic similarity, and `Mistral-7B-Instruct-v0.3` remains the best in stance alignment.

To quantify this robustness, we compute Kendall's $\tau$ between the ranking induced by the original round-wise aggregation and the ranking induced by bipartite matching. The median $\tau$ is $0.55$ with a standard deviation of $0.25$, confirming that matching-based aggregation stretches the performance range while preserving the core model ranking patterns.

Taken together, these analyses show that (i) DEBATE's metrics sit well above noise, (ii) model comparisons are statistically reliable, and (iii) alternative aggregation strategies can further increase discriminative power without changing the substantive conclusions about which role-playing LLM agents best align with human opinion dynamics.

## T    USER INTERFACES FOR THE HUMAN EXPERIMENT

We provide screenshots of the user interfaces that participants encountered during the multi-player experiment (Figures 7–7). Each figure illustrates one stage of the experimental process, from onboarding to the demographic survey.

In our multi-player experiment, participants were recruited through Prolific. At first, they would be navigated to a **consent form** outlining the study's procedures, duration, compensation, and confidentiality (Figure 7). Those who agreed to participate were then shown a **general introduction** explaining the flow of the task (Figure 7). Specifically, they were informed that they would be given a statement (e.g., "Smoking cigarettes causes cancer") and asked to write a short post as their initial opinion. They were also told that the experiment would last approximately 20 minutes and involve a sequence of conversations with other participants.

After reviewing the instructions, participants began by reporting their **initial opinion** on the assigned discussion statement and selecting a slider value to indicate the extent to which they agreed or disagreed with the statement (Figure 7). The core of the study consisted of **three rounds of interaction**, each following the same structure (Figures 7–7). First, participants were informed of who they would chat with and then were directed to write a short tweet-like post summarizing their current stance. Next, they would have twenty seconds to prepare for a dyadic conversation with a different partner. This ensured that each participant was exposed to all other perspectives across rounds.

At the end of the third round, participants submitted a **post opinion** in order for us to capture how their stance evolved during the course of the discussions (Figure 7). Finally, they completed a detailed **demographic survey** (Figures 7–7), after which they were compensated for $5 for their efforts.

**Onboarding Consent Form**

Please read this consent agreement carefully before deciding whether to participate in this experiment.

What you will do in this research: You will play a series of communication games with other participants.

Time required: This study will take approximately twenty minutes.

Purpose of the research: The purpose is to understand how conversations evolve in a networked community.

Risks: There are no anticipated risks associated with participating in this study. The effects should be comparable to viewing a computer monitor and using a mouse for the duration of the experiment.

Compensation: You will receive course credits for completing the experiment.

Confidentiality: Your participation in this study will remain confidential. No personally identifiable information will be collected. Your anonymous data may be shared with other researchers and used in future projects.

Participation and withdrawal: Your participation in this study is completely voluntary and you may refuse to participate or choose to withdraw at any time without penalty or loss of benefits to which you are otherwise entitled.

By clicking "I Agree", you consent to participate in this experiment.

Figure 7: Onboarding consent form.

**Onboarding Consent Form**

Please read this introduction carefully before participating in this experiment.

In this experiment we are interested in understanding how people discuss various topics in online platforms like Twitter or Reddit. To start, you will be given a statement (e.g. "Smoking cigarettes causes cancer") and asked to write a short post explaining whether or not you think it is true and why.

You will then have a series of three discussions with each of three other study participants using an online texting interface. In each discussion, do your best to keep the conversation going and to stay on topic.

After each discussion, you will be asked to again summarize your current view of the topic in a new post; the new summary can reiterate your initial position or can differ if your thinking has changed. It is important that it captures your own true belief about the topic.

The study will end after the third discussion, and will take 20 minutes to complete.

Press the BUTTON when you are ready to begin.

Figure 7: Onboarding introduction (continued).

Initial Opinion

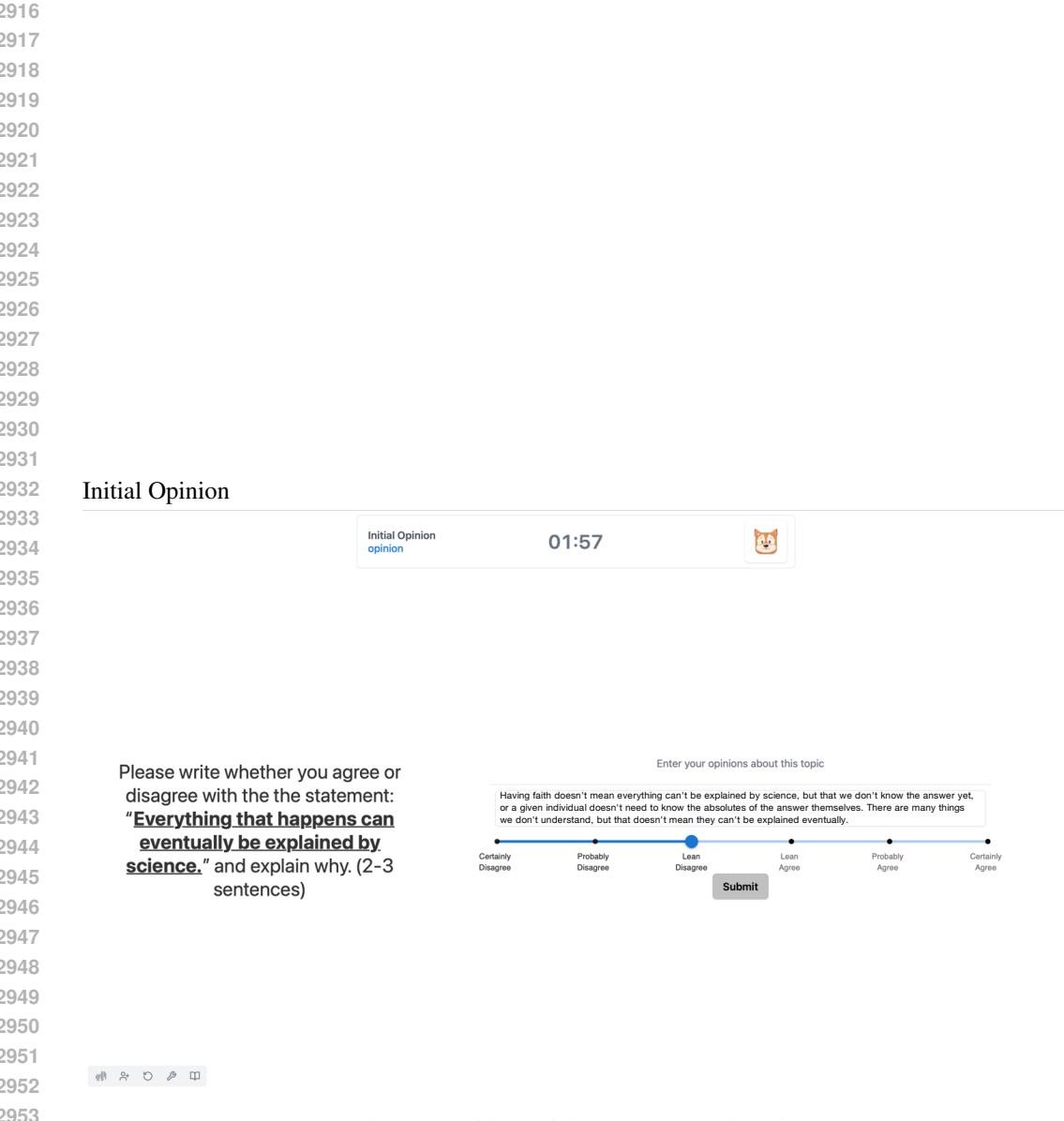

Figure 7: Initial Opinion screen (continued).

Round 1

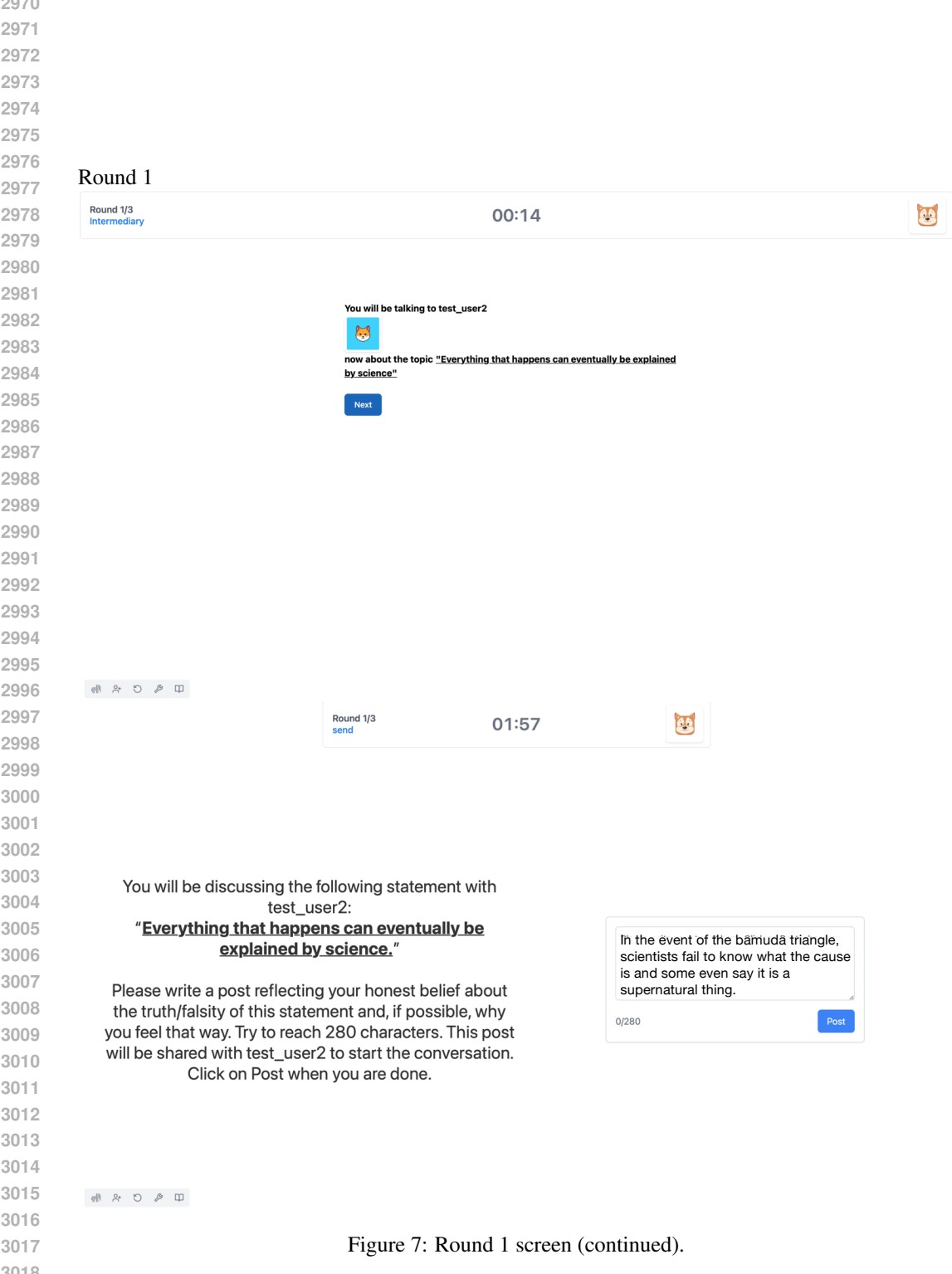

Figure 7: Round 1 screen (continued).

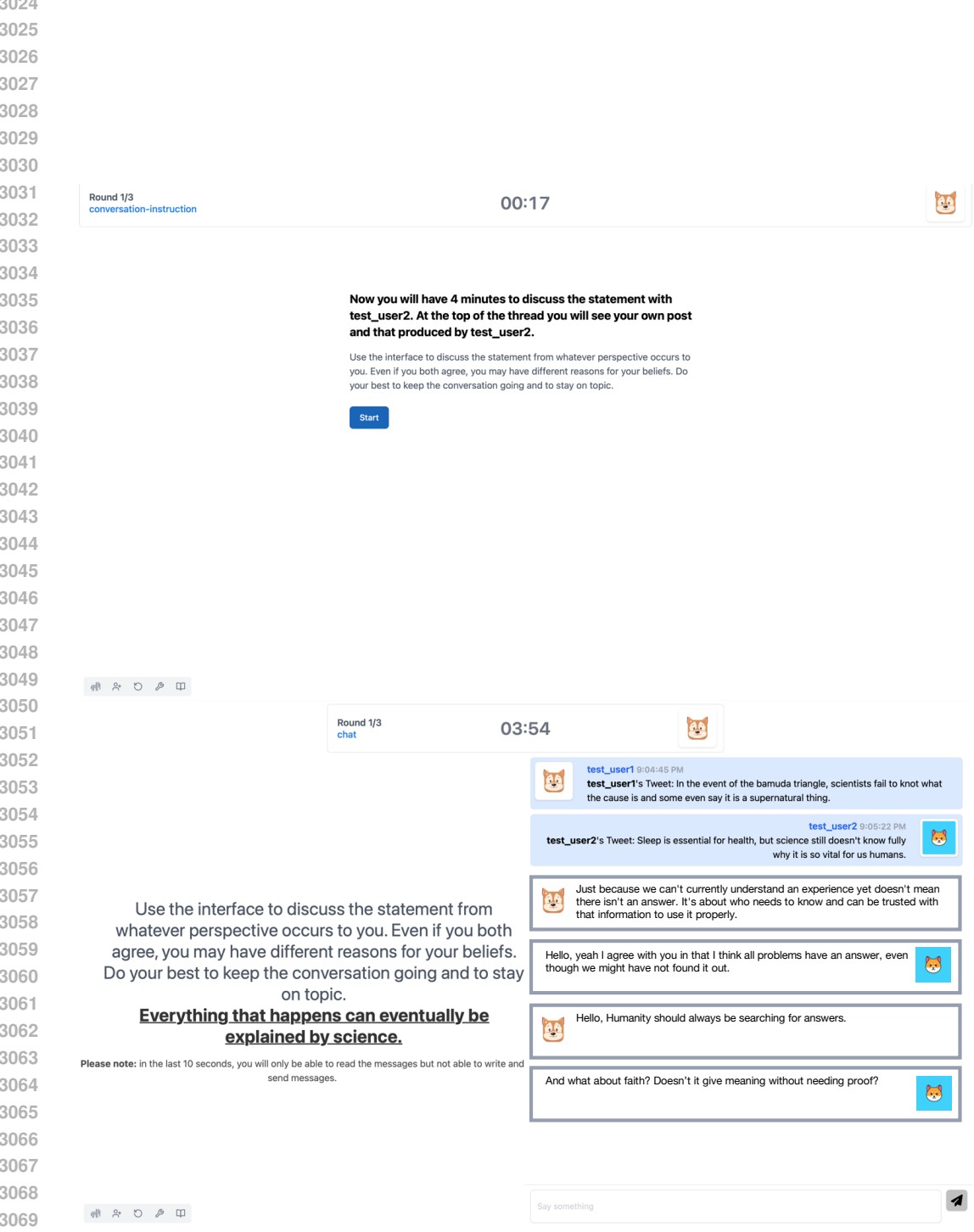

Figure 7: Round 1 screen (continued).

Round 2

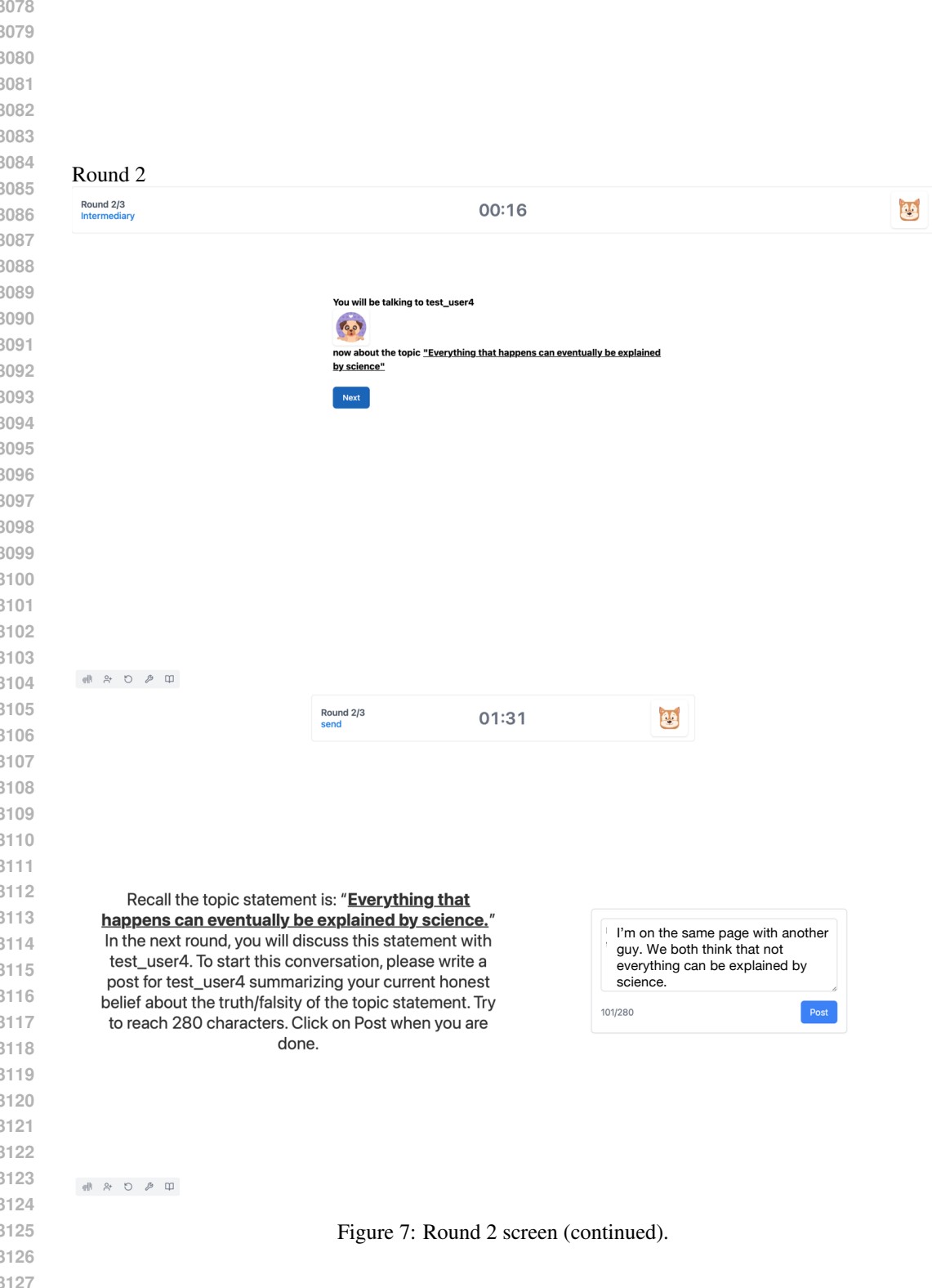

Figure 7: Round 2 screen (continued).

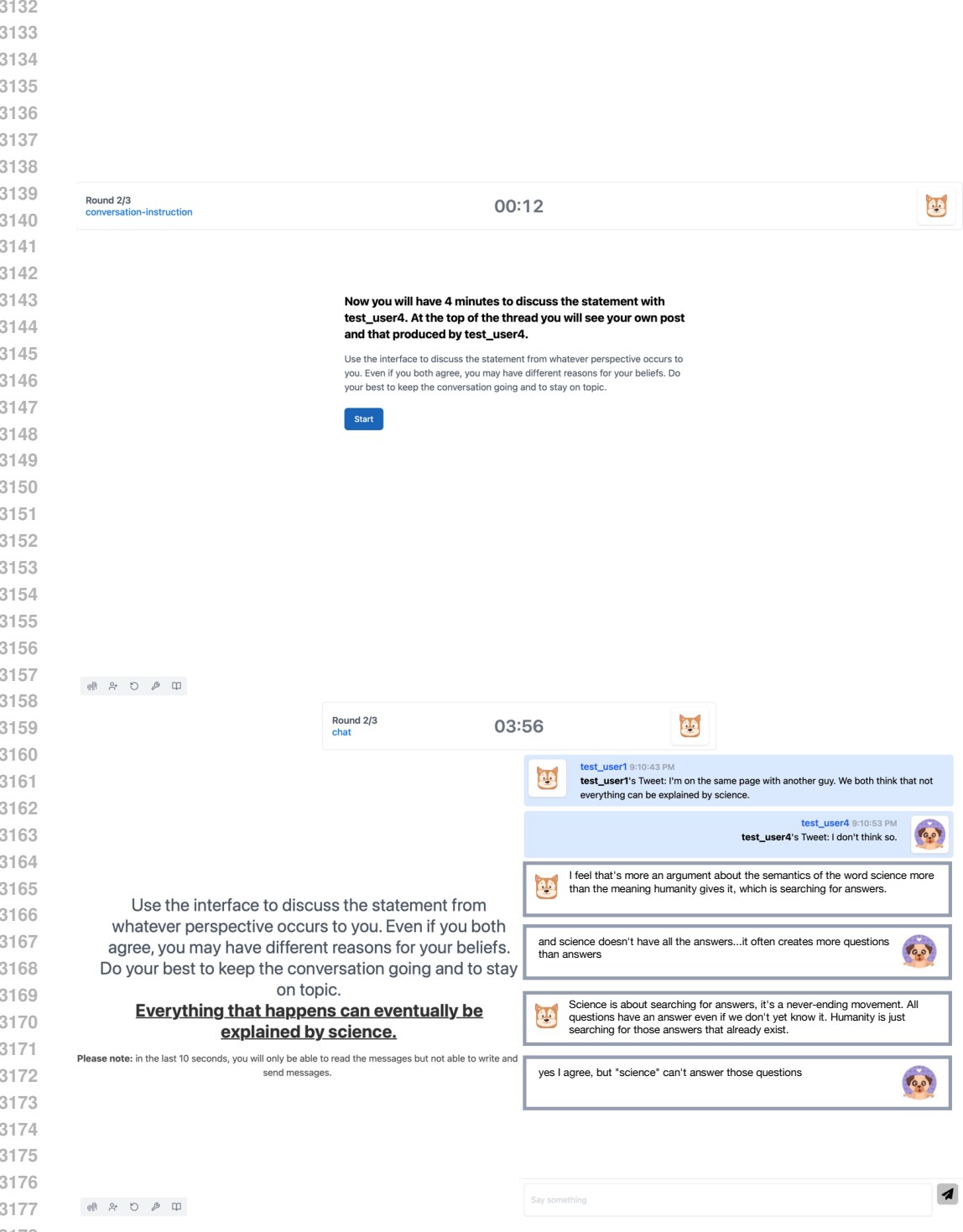

Figure 7: Round 2 screen (continued).

Round 3

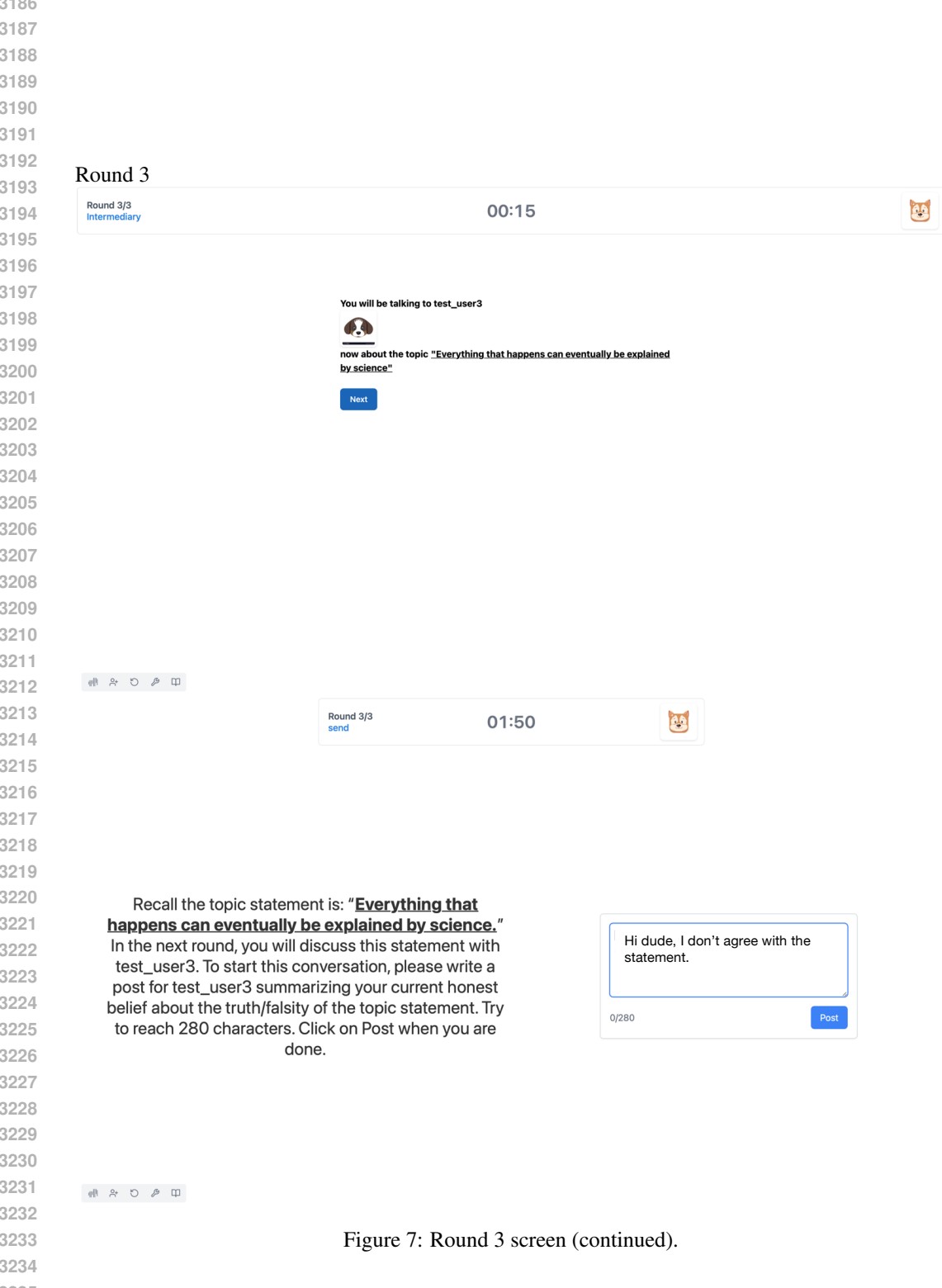

Figure 7: Round 3 screen (continued).

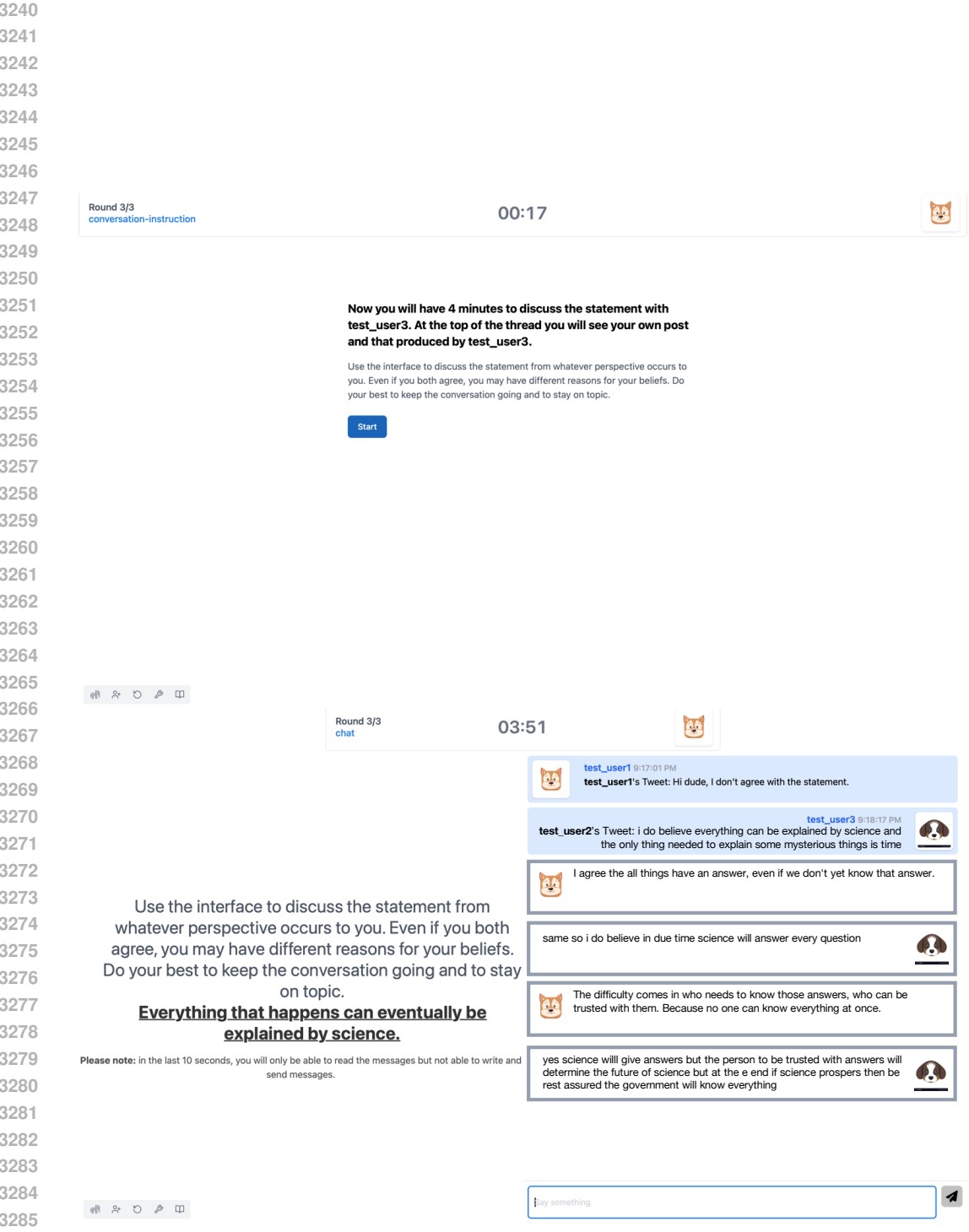

Figure 7: Round 3 screen (continued).

Post Opinion

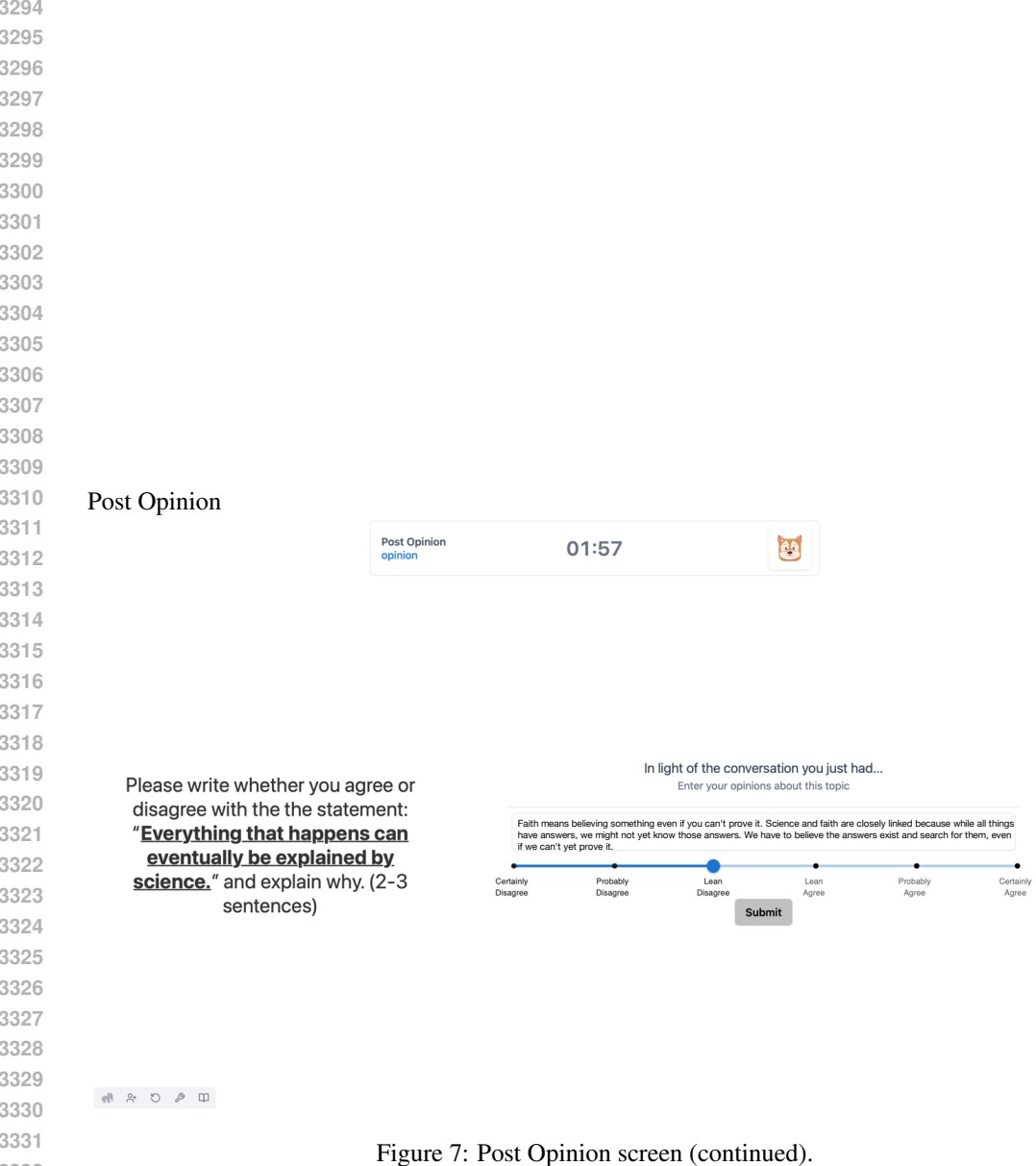

Figure 7: Post Opinion screen (continued).

## Demographic Survey

1. What is your age?
   (Open numeric input, 0–120)

2. What is your gender?
   - Male
   - Female
   - Something else (specify)
   - Prefer not to answer

3. What is your country/region of residency?
   - List of countries
   - Other (specify)
   - Prefer not to answer

4. What is your country/region of origin?
   - List of countries
   - Other (specify)
   - Prefer not to answer

5. What is the highest level of education you have completed?
   - High school or less
   - Some college
   - Associate's degree
   - Bachelor's degree
   - Master's or Doctoral degree
   - Professional degree
   - Prefer not to answer

6. What is your race/ethnicity? (Select all that apply)
   - Black or African American (non-Hispanic)
   - Hispanic
   - White (non-Hispanic)
   - Asian, South Asian, or Pacific Islander
   - Native American or American Indian
   - Other (specify)
   - Prefer not to answer

Figure 7: Demographic survey (continued).

## Demographic Survey (continued)

7. What is your household annual income?
   - Less than $25,000
   - $25,000–$49,999
   - $50,000–$74,999
   - $75,000–$99,999
   - $100,000–$149,999
   - $150,000–$199,999
   - Over $200,000
   - Prefer not to answer

8. Generally speaking, do you usually think of yourself as a(n)...
   - Strong Republican
   - Republican
   - Independent
   - Democrat
   - Strong Democrat
   - Other (specify)
   - Prefer not to answer

9. If Independent or Other: Do you lean more toward. . .
   - Republican Party
   - Democratic Party
   - Neither

10. In general, would you describe your political views as...
    - Very conservative
    - Conservative
    - Moderate
    - Liberal
    - Very liberal
    - Don't know
    - Prefer not to answer

11. Are you currently...
    - Married
    - Living with a partner but not married
    - Widowed
    - Divorced
    - Separated
    - Never been married
    - Prefer not to answer

12. If you have children, where do they go to school? (Select all that apply)
    - Public school
    - Private school
    - Home school
    - University or Technical/Community College
    - They are out of school
    - I don't have children
    - Prefer not to answer

Figure 7: Demographic survey (continued).

**Demographic Survey (continued)**

13. Which of these statements comes closest to describing your feelings about the Bible?
    - The Bible is the actual word of God and is to be taken literally, word for word.
    - The Bible is the inspired word of God but not everything in it should be taken literally.
    - The Bible is an ancient book of fables, legends, history, and moral precepts recorded by men.
    - Prefer not to answer

14. Would you describe yourself as a "born-again" or evangelical Christian, or not?
    - Yes, would
    - No, would not
    - Don't know
    - Prefer not to answer

15. What is your religious preference?
    - Protestant
    - Roman Catholic
    - Jewish
    - Muslim/Islam
    - Mormon/Latter-Day Saints
    - Other Christian Religion
    - Other Non-Christian Religion
    - No Religion/Atheist/Agnostic
    - Don't know
    - Prefer not to answer

16. Which of the following best describes the kind of work you do?

    - Agriculture, Forestry, Fishing and Hunting
    - Mining, Oil and Gas Extraction, and Utilities
    - Construction
    - Manufacturing
    - Wholesale Trade
    - Sales &/or Retail Trade
    - Transportation and Warehousing
    - Media, Communications, and Digital Entertainment
    - Finance, Accounting, and Consulting

    - Insurance
    - Real Estate, Rental, and Leasing
    - Personal Care and Services
    - Research
    - Engineering, Computer-Related Design, and Architecture
    - Law and Legal Services
    - Education
    - Health Care and Social Assistance
    - Arts, Entertainment, and Recreation

    - Restaurant, Travel, and Lodging
    - Non-Profit, Community, Religious and Social Service Organizations
    - Maintenance and Repair Services
    - Cleaning Services
    - Government
    - Other (specify)
    - Prefer not to answer

Figure 7: Demographic survey (continued).

