# OpenReview forum: "DEBATE: A Large-Scale Benchmark for Evaluating Opinion Dynamics in Role-Playing LLM Agents"
_ICLR.cc/2026/Conference — Submitted to ICLR 2026_

### Official Review · Reviewer_K4Xk · 2025-10-20

**Soundness:** 3
**Presentation:** 2
**Contribution:** 2
**Rating:** 4
**Confidence:** 4

**Summary:**

This paper introduces DEBATE, a large-scale dataset of multi-player, multi-round discussions across controversial debate topics which can be used to evaluate the effectiveness of LLM agents at simulating opinion dynamics. They design ways to measure alignment between agent and human opinion evolution, then use this for experiments assessing the realism of different base models and agent setups. They find that LLM agents differ from humans in key ways, including stronger opinion convergence and positive belief shift, which cannot be fixed simply by improved context or fine-tuning on human utterances.

**Strengths:**

- The dataset does seem to be both novel and relevant for the very timely problem of evaluating LLM agent-based social simulations. Having both internal stances and external utterances, as well as the diverse range of conversation topics and metadata, makes this a promising dataset.
- The analysis RQs are interesting and naturally emerge from the dataset construction - in its current form, I don’t think you need to frame the paper as mainly a dataset paper, as the analyses are also pretty interesting. The stance homogenization and positive belief shift findings do seem noteworthy and the right kinds of things to target for improvement in future RPLA work.
- Careful metric design, with human validation whenever necessary.

**Weaknesses:**

- My main concern is that there is not much quality validation of the human conversations in the dataset for a benchmark paper. It appears that all on-topic utterances that were a part of completed interactions were used for evaluation. More quality validation of the resulting dataset might be useful - I’m concerned that, as crowdworkers were completing a single-episode task with no incentive for honestly reporting preferences, there might be significant numbers of low-quality interactions that would preferably be filtered out.
- Similarly, there is not much analysis of what kinds of interactions are present in the dataset, as well as any observed benefits to conversation diversity/quality that stem from the new benchmark attributes.
- While the metrics are well-motivated, they seem to have very limited discriminative power (in Table 1, the semantic similarity/stance difference metrics show very little variance between different models, despite the diversity of the models evaluated), which may limit the dataset’s utility as a benchmark.

**Questions:**

- Why are LLMs off-topic so often? Even 78%, the highest on-topic rate in Table 1, seems lower than I would expect from recent LLMs.
- I’d advise you put any tables that are referenced in the main paper into the main paper, rather than the appendix (e.g. Tables 13-14), with your extra page for camera-ready. The current layout is a bit difficult to follow.

---

> ### Author Response · Authors · 2025-11-24
> **Response to weakness 1 - Part 1**
>
> > weakness 1: "My main concern is that there is not much quality validation of the human conversations in the dataset for a benchmark paper. It appears that all on-topic utterances that were a part of completed interactions were used for evaluation. More quality validation of the resulting dataset might be useful - I’m concerned that, as crowdworkers were completing a single-episode task with no incentive for honestly reporting preferences, there might be significant numbers of low-quality interactions that would preferably be filtered out."
>
> Thank you for raising this concern about data quality. We agree that, for a benchmark paper, it is important to show that results are not driven by low-engagement crowdworkers. In the current version, all analyses use (i) only groups that completed the full multi-round experiment (725 groups; 2,584 participants) and (ii) only utterances that pass an on-topic filter relative to the assigned topic. You are correct that this effectively means “all on-topic utterances from completed interactions.” To test whether additional cleaning at the individual level would change our conclusions, we ran a new robustness analysis.
>
> Concretely, for each participant we computed (a) their average number of messages per round and (b) their individual on-topic rate (using the same on-topic classification as described in the main paper). We then defined a selective “high-engagement” subset of participants who (1) produced at least 2 messages per round on average and (2) had an on-topic rate of at least 80% across their messages. This allows us to rerun the analysis separately for workers who are highly engaging (56% of workers) and those who interact less (44%) but still engage meaningfully (producing a minimum of 6 messages).
>
> We re-ran all three simulation modes using only this higher-engagement subset. Across models and simulation modes, as shown in the result table below, the evaluation metrics are very similar to those in Table 4 of the paper: the best semantically aligned model is the same, and the differences in semantic similarity, stance, and ROUGE-L are very small and do not alter any of our substantive claims (see new table below). In other words, excluding lower-engagement participants at the individual level does not reveal a hidden pool of low-quality conversations that drives our main results; instead, it supports that our original message-level on-topic filtering is already sufficient for the benchmark’s conclusions. In the camera-ready version, we will (i) report this participant-level quality analysis, and (ii) clearly document the definition of the high-engagement subset so that downstream users can choose between the full cleaned dataset and a stricter subset if desired.
>
> **Simulation Mode 1: Next Message Prediction**
>
> ---
> | Model | semantic_similarity | likert_diff | n_word_diff | n_word_abs_diff | rouge_L |
> |---------------------------|---------------------|------------------|--------------------|--------------------|----------------|
> | gpt-4o-mini-2024-07-18 | 0.47 ± 0.01 | 1.12 ± 0.05 | -32.34 ± 0.62 | 33.22 ± 0.60 | 0.10 ± 0.01 |
> | Llama-3.1-Tulu-3-8B-SFT | 0.43 ± 0.01 | 1.14 ± 0.06 | -41.23 ± 1.29 | 45.31 ± 0.93 | 0.06 ± 0.01 |
> | Llama-3.1-8B-Instruct | 0.44 ± 0.01 | 1.18 ± 0.04 | -37.52 ± 0.87 | 38.94 ± 0.76 | 0.07 ± 0.01 |
> | Llama-3.1-70B-Instruct | 0.44 ± 0.01 | 1.09 ± 0.05 | -25.69 ± 1.05 | 28.49 ± 0.85 | 0.08 ± 0.01 |
> | Mistral-7B-Instruct-v0.3 | 0.46 ± 0.01 | 1.11 ± 0.05 | -47.06 ± 0.71 | 47.68 ± 0.67 | 0.07 ± 0.01 |
> | Qwen2.5-32B-Instruct | 0.45 ± 0.01 | 1.08 ± 0.05 | -21.90 ± 0.82 | 26.67 ± 0.64 | 0.08 ± 0.01 |
> ---
>
> **Simulation Mode 2: Tweet-guided Conversation Simulation**
>
> ---
> | Model | semantic_similarity | likert_diff | n_word_diff | n_word_abs_diff | rouge_L |
> |---------------------------|---------------------|------------------|--------------------|--------------------|----------------|
> | gpt-4o-mini-2024-07-18 | 0.41 ± 0.01 | 1.19 ± 0.05 | -59.08 ± 0.78 | 59.23 ± 0.76 | 0.08 ± 0.01 |
> | Llama-3.1-Tulu-3-8B-SFT | 0.40 ± 0.01 | 1.35 ± 0.07 | -53.91 ± 0.88 | 54.70 ± 0.82 | 0.05 ± 0.01 |
> | Llama-3.1-8B-Instruct | 0.39 ± 0.01 | 1.21 ± 0.05 | -53.67 ± 0.93 | 54.20 ± 0.86 | 0.06 ± 0.01 |
> | Llama-3.1-70B-Instruct | 0.39 ± 0.01 | 1.13 ± 0.05 | -51.75 ± 1.24 | 52.28 ± 1.17 | 0.06 ± 0.01 |
> | Mistral-7B-Instruct-v0.3 | 0.40 ± 0.01 | 1.18 ± 0.06 | -47.27 ± 0.76 | 47.80 ± 0.70 | 0.06 ± 0.01 |
> | Qwen2.5-32B-Instruct | 0.40 ± 0.01 | 1.26 ± 0.06 | -48.93 ± 1.30 | 50.85 ± 1.10 | 0.07 ± 0.01 |
> ---

---

> > ### Author Response · Authors · 2025-11-24
> > **Response to weakness 1 - Part 2**
> >
> > [continued Response to weakness 1 - Part 1]
> >
> > **Simulation Mode 3: Full Conversation Simulation**
> >
> > ---
> > | Model | semantic_similarity | likert_diff | n_word_diff | n_word_abs_diff | rouge_L |
> > |---------------------------|---------------------|------------------|--------------------|--------------------|----------------|
> > | gpt-4o-mini-2024-07-18 | 0.40 ± 0.01 | 1.26 ± 0.05 | -58.71 ± 0.73 | 58.90 ± 0.71 | 0.08 ± 0.01 |
> > | Llama-3.1-Tulu-3-8B-SFT | 0.39 ± 0.01 | 1.42 ± 0.07 | -55.21 ± 0.92 | 56.32 ± 0.82 | 0.05 ± 0.01 |
> > | Llama-3.1-8B-Instruct | 0.38 ± 0.01 | 1.25 ± 0.05 | -53.52 ± 0.91 | 54.04 ± 0.85 | 0.06 ± 0.01 |
> > | Llama-3.1-70B-Instruct | 0.37 ± 0.01 | 1.20 ± 0.05 | -51.46 ± 1.11 | 52.18 ± 1.01 | 0.06 ± 0.01 |
> > | Mistral-7B-Instruct-v0.3 | 0.38 ± 0.01 | 1.25 ± 0.05 | -47.17 ± 0.80 | 47.79 ± 0.73 | 0.06 ± 0.01 |
> > | Qwen2.5-32B-Instruct | 0.38 ± 0.01 | 1.29 ± 0.05 | -50.54 ± 1.11 | 52.07 ± 0.95 | 0.06 ± 0.01 |
> > ---

---

> ### Author Response · Authors · 2025-11-24
> **Response to weakness 2**
>
> > weakness 2: 'Similarly, there is not much analysis of what kinds of interactions are present in the dataset, as well as any observed benefits to conversation diversity/quality that stem from the new benchmark attributes."
>
> Thank you for raising this point. We address it in two ways: (i) by quantifying how the DEBATE-specific benchmark attributes (private beliefs, demographics, personalized history) affect alignment, and (ii) by providing a more concrete characterization of interaction types in the dataset.
>
> **(i) Benchmark attributes and their effect on conversation quality/alignment.**
>
>  Beyond the “No Initial Opinion” ablation (Tables 12–13, Appendix L) discussed in our response to Reviewer 2, we have now run an additional ablation where we remove all distinct per-agent context from the RPLA memory: demographics, private initial opinion, initial tweet, and personalized history. In this condition, each agent is prompted only to “role-play a person” with no individual
> grounding. On Depth topics under Simulation Mode 3 (Full Conversation Simulation) with gpt-4o-mini-2024-07-18, both semantic similarity and stance alignment significantly worsen (p < .01):
>
> ---
> | Condition | average semantic similarity | average stance difference |
> |-------------------------------------------------------|-----------------------------|----------------------------|
> | Pre-ablation (Table 4) | 0.41 ± 0.01 | 1.22 ± 0.03 |
> | Remove all benchmark attributes from each RPLA agent | 0.38 | 1.53 |
> ---
>
> Together with the “No Initial Opinion” rows in Tables 13–14 (where stance difference increases from 1.30→1.38 on Depth topics and 1.22→1.27 on Breadth topics, both p < .01), these ablations show that the DEBATE-specific attributes (e.g., private self-reported beliefs and individualized memory) do not just sit in the prompt but materially improve alignment between human and RPLA trajectories. We will move a compact summary of these ablations into the main text.
>
> **(ii) What kinds of interactions are present.**
>
>  In the opinion-dynamics literature, both classic ABM work [1,2,3] and recent LLM-based opinion dynamics simulations [4,5] typically model interactions as 1-dimensional stance updates and evaluate dynamics in terms of aggregate quantities like mean opinion and opinion diversity (e.g., DeGroot-type averaging [2], bounded-confidence models [3], and recent LLM-based simulations [4,5]). Our group-level analyses in Sec. 6 follow this tradition by focusing on changes in average stance and within-group variance, as in prior work [1–5].
>
> That said, we agree it is useful to make the interaction types in DEBATE more concrete, especially because classic opinion-dynamics models typically reduce interaction to 1-dimensional stance updates and do not distinguish between different conversational acts [1–3].
>
> As a first step beyond this tradition, we manually coded 100 randomly sampled human utterances for coarse interaction type. Of these, 86% were on-topic contributions to the assigned issue (arguments, reasons, or evaluations), while the remaining 14% consisted of short digressions (6%), greetings or conversation-management moves (5%), and clarification questions (3%, e.g., “what do you mean?”). We will add a brief summary of this breakdown to the paper, and emphasize that more fine-grained interaction-type analyses (e.g., argument structure, agreement/disagreement moves) are a natural future direction for follow-up work using DEBATE. We will also note that DEBATE, as the first benchmark of its kind, creates an opportunity for the community to extend opinion-dynamics work beyond purely 1-dimensional stance-update models, toward richer analyses of conversational behavior.
>
> **References**
>
> [1] Flache, A., Mäs, M., Feliciani, T., Chattoe-Brown, E., Deffuant, G., Huet, S., & Lorenz, J. (2017). Models of social influence: Towards the next frontiers. The journal of artificial societies and social simulation, 20(4), 2.
>
> [2] DeGroot, M. H. (1974). Reaching a consensus. Journal of the American Statistical association, 69(345), 118-121.
>
> [3] Hegselmann, R. (2015). Opinion dynamics and bounded confidence: models, analysis and simulation. The Journal of Artificial Societies and Social Simulation.
>
> [4] Chuang, Y. S., Goyal, A., Harlalka, N., Suresh, S., Hawkins, R., Yang, S., ... & Rogers, T. (2024). Simulating opinion dynamics with networks of llm-based agents. In Findings of the association for computational linguistics: NAACL 2024 (pp. 3326-3346).
>
> [5] Taubenfeld, A., Dover, Y., Reichart, R., & Goldstein, A. (2024). Systematic Biases in LLM Simulations of Debates. In Proceedings of the 2024 Conference on Empirical Methods in Natural Language Processing (pp. 251-267).

---

> ### Author Response · Authors · 2025-11-24
> **Response to Weakness 3 Part 1**
>
> > Weakness 3: "While the metrics are well-motivated, they seem to have very limited discriminative power (in Table 1, the semantic similarity/stance difference metrics show very little variance between different models, despite the diversity of the models evaluated), which may limit the dataset’s utility as a benchmark."
>
> Thank you for raising this point about discriminative power. We agree that a useful benchmark should (i) rise clearly above noise, (ii) support statistically reliable model comparisons, and (iii) ideally offer further separation through improved discriminative power
> To address this, we (1) compare our metrics to a permutation baseline, (2) report bootstrapped uncertainty and non-parametric tests, and (3) explore a bipartite-matching aggregation that increases discriminative power while preserving the main trends.
>
> **(1) Permutation baseline: metrics are well above noise**
>
> To verify that our alignment metrics are not merely reflecting noise, we derived a permutation baseline that deliberately destroys the human–agent correspondence: for each round, we randomly pair an RPLA-generated round with a human round from a different, randomly chosen topic, and then re-run the same round-wise aggregated evaluation used in the paper to compute semantic similarity and stance difference.
>
> ---
> | model_name | Avg Semantic Similarity (original) | Avg Stance Difference (original) | Avg Semantic Similarity (permutation) | Avg Stance Difference (permutation) |
> |----------------------------|------------------------------------|----------------------------------|----------------------------------------|--------------------------------------|
> | gpt-4o-mini-2024-07-18 | 0.41 ± 0.01 | 1.30 ± 0.05 | 0.28 | 1.99 |
> | Llama-3.1-Tulu-3-8B-SFT | 0.40 ± 0.01 | 1.46 ± 0.07 | 0.25 | 2.38 |
> | Llama-3.1-8B-Instruct | 0.39 ± 0.01 | 1.33 ± 0.05 | 0.28 | 2.17 |
> | Llama-3.1-70B-Instruct | 0.38 ± 0.01 | 1.25 ± 0.05 | 0.28 | 2.04 |
> | Mistral-7B-Instruct-v0.3 | 0.40 ± 0.01 | 1.25 ± 0.05 | 0.29 | 2.09 |
> | Qwen2.5-32B-Instruct | 0.40 ± 0.01 | 1.30 ± 0.05 | 0.27 | 2.01 |
> ---
>
> The table above reports both the original metrics (Table 4, Simulation Mode 3) and the permutation baseline. Across all models, the permuted semantic similarity drops substantially (e.g., from 0.41 to 0.28 for gpt-4o-mini-2024-07-18), and the stance difference increases sharply (e.g., from 1.30 to 1.99). These gaps are large and statistically significant (ps < .001), showing that the metrics are not just noise but are sensitive to the meaningful alignment between human debates and model-generated debates. We will add this permutation-baseline table and description to the revision.
>
> **(2) Bootstrapping and statistical tests show reliable model differences**
>
> To address discriminative power more directly, we have quantified the uncertainty around each metric via bootstrapping and use non-parametric tests to compare models (Appendix J). For Simulation Mode 3 (full conversation), Appendix J.1 reports:
> > The Friedman test reveals a significant overall difference across the six models (χ² = 17.87, df = 5, p = .003). Wilcoxon tests show that gpt-4o-mini-2024-07-18 significantly outperforms Llama-3.1-8B-Instruct (p = .018), Llama-3.1-70B-Instruct (p = .017), Mistral-7B-Instruct-v0.3 (p = .024), and Qwen2.5-32B-Instruct (p = .018). The difference with Llama-3.1-Tulu-3-8B-SFT is not statistically significant (p = .146), but the trend still favors gpt-4o-mini.
>
> Thus, even though the absolute differences in Table 1 may look numerically small, once we account for variance via bootstrapping, the benchmark does reliably distinguish models and supports statistically grounded comparisons rather than being limited to “within-noise” variation. We will clarify this in the main text by explicitly pointing to the confidence intervals / standard errors and to Appendix J.

---

> ### Author Response · Authors · 2025-11-24
> **Response to Weakness 3 Part 2**
>
> **(3) Alternative aggregation: increasing discriminative power without changing conclusions**
>
> We also agree that “better than noise” is not the only goal; researchers may want stronger separation between models. One important source of noise stems from our current round-wise aggregation strategy, which we chose to avoid assuming a one-to-one mapping between human and agent utterances, as described in Sec 4.3:
>
> > “Because there is no one-to-one mapping between simulated and human utterances, we adopted a round-wise aggregated evaluation: each simulated utterance bub_ubu​ is compared to all on-topic human utterances uuu from the same round and speaker. We average metric scores across utterances, agents, and rounds…”
>
> While this approach ensures the metrics are order-agnostic, averaging over all possible pairs can be noisy. A few outlier pairs can pull the average toward or away from the true alignment signal, and the alignment metrics are effectively “noised” by many weak human–model pairings.
>
> To test how much this affects discriminative power, we experimented with an alternative, more direct aggregation method: “maximum-weight bipartite matching” [1, 2, 3]. For each round, we:
>
> - Treat the human utterances and RPLA utterances as nodes in a bipartite graph with edge weights given by semantic similarity.
> - Compute a maximum-weight bipartite matching so that each human utterance is matched to exactly one model utterance (and vice versa), without enforcing the original message order.
> - Recompute semantic similarity, stance difference, and length metrics only on these matched pairs.
>
> This matching-based scheme concentrates the evaluation on the best-aligned pairs instead of averaging over all pairs, which (i) prevents a single generic or off-topic model utterance from being “matched” to every human utterance in the round and (ii) yields a sparse, interpretable alignment.
>
> Using this bipartite-matching aggregation on Depth topics, the metric ranges become more spread out.
>
> ---
> | Model | semantic_similarity | Stance difference | Length difference | Absolute length difference |
> |---------------------------|---------------------|-------------------|--------------------|----------------------------|
> | gpt-4o-mini-2024-07-18 | 0.54 | 0.84 | 2.42 | 5.85 |
> | Llama-3.1-Tulu-3-8B-SFT | 0.47 | 0.77 | 1.26 | 7.19 |
> | Llama-3.1-8B-Instruct | 0.49 | 0.80 | 3.15 | 11.49 |
> | Llama-3.1-70B-Instruct | 0.50 | 0.91 | 4.62 | 12.68 |
> | Mistral-7B-Instruct-v0.3 | 0.50 | 0.63 | 0.90 | 6.58 |
> | Qwen2.5-32B-Instruct | 0.49 | 0.73 | 4.29 | 12.29 |
> ---
>
> - The range of average semantic similarity increases from [0.38,0.41] (range = 0.03) to [0.47,0.54] (range = 0.07).
> - The range of average stance difference increases from [1.25,1.46] (range = 0.21) to [0.63,0.91] (range = 0.28).
>
> Crucially, the relative ranking is preserved: gpt-4o-mini-2024-07-18 remains the best model in semantic similarity, and Mistral-7B-Instruct-v0.3 remains the best in stance alignment. This suggests that (i) the underlying signal is robust (our main conclusions do not depend on the aggregation choice), and (ii) there is room to further enhance discriminative power by refining aggregation strategies (e.g., matching-based schemes), which we see as a natural direction for future versions of the benchmark.
>
> Finally, note that our semantic similarity operates on sets of high-dimensional embeddings, not low-dimensional histograms. Therefore, classical distribution-divergence measures such as KL are ill-suited here [2]. Matching-based comparisons over semantic embeddings are specifically chosen to avoid these issues.
>
> **Summary.** In response to the concern about limited discriminative power, we (i) show that our metrics sit well above a permutation baseline, (ii) report bootstrapped uncertainty and statistical tests that establish reliable model difference comparison, and (iii) demonstrate that alternative aggregation strategies can further increase spread between models while preserving all key performance trends. Together, these analyses support DEBATE’s utility as a benchmark for comparing role-playing LLM agents.
>
> **References**
>
> [1] Kuhn, H. W. (1955). The Hungarian method for the assignment problem. Naval research logistics quarterly, 2(1‐2), 83-97.
>
> [2] Kusner, M., Sun, Y., Kolkin, N., & Weinberger, K. (2015). From word embeddings to document distances. In International conference on machine learning (pp. 957-966). PMLR.
>
> [3] Lee, S., Lee, D., Jang, S., & Yu, H. (2022). Toward Interpretable Semantic Textual Similarity via Optimal Transport-based Contrastive Sentence Learning. In Proceedings of the 60th Annual Meeting of the Association for Computational Linguistics (Volume 1: Long Papers) (pp. 5969-5979).

---

> ### Author Response · Authors · 2025-11-24
> **Response to Question 1**
>
> > Question 1: "Why are LLMs off-topic so often? Even 78%, the highest on-topic rate in Table 1, seems lower than I would expect from recent LLMs."
>
> Thank you for this question. For reference, human utterances are on-topic 86% (Depth) and 91% (Breadth), so while the best RPLA (78% on-topic for Llama-3.1-70B-Instruct in Simulation Mode 1) is below humans, it is not wildly off.
>
> To better understand the remaining gap, we randomly sampled 100 off-topic messages from this strongest setting (Llama-3.1-70B-Instruct, Mode 1) and manually categorized them. Most of these off-topic cases fall into three patterns: (1) non–role-play responses (67%), where the model speaks as an assistant instead of staying in persona; (2) digressions (14%), where the model shifts into discussions not directly related to the topic; and (3) greetings / small talk (8%), such as greeting or casual chit-chat. Note that none of these utterances are used for evaluation by design and they are filtered out before we compute any alignment metrics
>
> Representative off-topic examples are shown below:
>
> ---
> | Category | Example Message |
> |------------------------|-----------------|
> | Non–role-play responses | **Topic:** “Public debt is the most important issue for the government to address first.”      **LLM:** “I’m glad we could find common ground on addressing public debt. Now that Trump’s tariffs are being implemented and causing disruptions in global markets, do you think they’ll actually lead to improved trade terms? Or do you think another approach should’ve been taken? (Note: Based on the context, my previous responses were analyzed before writing down my new response.)” |
> | Digression | **Topic:** “Religious and ethnic hatred poses the greatest threat to the world.”      **LLM:** “Yeah pretty much anything perceived by man can be subjective depending upon whose perception. What do u mean exactly thatLuck fades. Can U give me examples of instances where Luck faded.” |
> | greetings/small talks | **Topic:** “People obeying their rulers is a characteristic of democracy”      **LLM:** “I'm good thanks! Just having an interesting discussion online, what about u?” |
> ---
>
> In sum, the lower on-topic rate is driven by a mix of non–role-play responses, digressions, and small talk, rather than long, wildly irrelevant content. We also view the on-topic rate itself as an important evaluation dimension: for successful multi-player, multi-turn opinion-dynamics simulations, agents must not only produce human-like opinions but also sustain focused, in-persona discussion over time, and DEBATE’s on-topic metric directly measures this capability. In the camera-ready version, we will add a brief description of this breakdown of off-topic responses, clarify that these utterances are excluded from our main alignment metrics, and explicitly motivate the inclusion of on-topic rate as a complementary metric, so that readers can better interpret both the alignment scores and the on-topic rates.

---

> ### Author Response · Authors · 2025-11-24
> **Response to Question 2**
>
> > Question 2: "I’d advise you put any tables that are referenced in the main paper into the main paper, rather than the appendix (e.g. Tables 13-14), with your extra page for camera-ready. The current layout is a bit difficult to follow."
>
> Thank you for this suggestion and we completely agree. In the camera-ready version, we will move the most important results tables that are currently only in the appendix (including the ablation results in Tables 13–14) into the main paper, so that readers can more easily follow the experimental setup and findings without having to cross-reference the appendix.

---

> > ### Comment · Reviewer_K4Xk · 2025-11-26
> >
> > Thanks to the authors for their detailed response to my review; I appreciate the engagement and hope my comments were helpful.
> >
> > Regarding benchmark quality, my concern is less what the authors noted (that "for a benchmark paper, it is important to show that results are not driven by low-effort crowdworkers"), but more that the benchmark itself needs to be a reliable evaluation target. While it's good that the results seem robust to filtering selectivity, the fact that nearly half of all crowdworkers generally didn't properly engage by either rarely speaking or frequently going off topic leads me to question how useful the pool of crowdworkers' aggregated opinions are in general. For this benchmark to be useful, I think the authors need to be significantly more restrictive in which data they choose to comprise the dataset for future evaluation.
> >
> > Regarding the discriminative power of the metrics used, I appreciate the additional analyses. It would be useful to report Kendall's Tau between the recovered rankings from different aggregation methods and bootstrapped runs - it seems that, at least in the table shown in "Response to Weakness 3 Part 2", there are some changes in the rankings between aggregation methods, when comparing to the original paper.
> >
> > I appreciate the author response and the effort they put into running additional analyses, but still have reservations about the data quality (which is a fundamental component of the contribution) and am maintaining my current score.

---

> ### Author Response · Authors · 2025-11-28
> **Response to your follow-up comment - Part 1**
>
> > Regarding benchmark quality, my concern is less what the authors noted (that "for a benchmark paper, it is important to show that results are not driven by low-effort crowdworkers"), but more that the benchmark itself needs to be a reliable evaluation target. While it's good that the results seem robust to filtering selectivity, the fact that nearly half of all crowdworkers generally didn't properly engage by either rarely speaking or frequently going off topic leads me to question how useful the pool of crowdworkers' aggregated opinions are in general. For this benchmark to be useful, I think the authors need to be significantly more restrictive in which data they choose to comprise the dataset for future evaluation.
>
>
> Thank you for your follow-up. We see that our earlier wording may have given the false impression that nearly half of the participants “didn’t properly engage,” which was not what we intended. All participants (725 groups; 2,584 participants) in the DEBATE benchmark already satisfy fairly strict inclusion criteria: they completed the full multi-round protocol and wrote messages at every phase of the experiment, finished the demographic survey at the end, and we are only using their on-topic utterances as the evaluation target. Among the participants in DEBATE, their engagement is generally high: 63.9% of participants produce at least 11 messages (tweets, utterances, initial/final opinions), with a median of 11, mean of 11.54, and minimum of 6 messages per participant. Likewise, 95.1% of participants have an on-topic rate ≥ 0.8, with median = 1.0, mean = 0.91 and minimum = 0.50.. In other words, even the “less-engaged” participants are still producing multiple (producing a minimum of 6 messages), mostly on-topic (op-topic rate ≥ 0.5) contributions.
>
> In our previous response, we defined a “selective higher-engagement” subset as those who (1) produced at least 2 messages per round on average and (2) had an on-topic rate of at least 80%. This higher-engagement subset comprises 56% of participants (producing on average 13.10 messages and with average on-topic rate = 0.97). The remaining 44% are not “low-quality crowdworkers” but are mainly less verbose (producing on average 9.55 messages with average on-topic rate = 0.83). To avoid miscommunication, we have updated the language in the previous response to “... We then defined a selective “higher-engagement” subset of participants who (1) produced at least 2 messages per round on average and (2) had an on-topic rate of at least 80% across their messages. This allows us to rerun the analysis separately for workers who are highly engaging (56% of workers) and those who interact less (44%) but still engage meaningfully (producing a minimum of 6 messages with average on-topic rate = 0.83).” Their engagement level and on-topic rate are summarized in the table below.
>
> | Participant Set | Average Messages | Average On-topic Rate |
> |------------------------------|------------------|------------------------|
> | Full set (100%) | 11.54 | 0.91 |
> | Higher-Engagement Subset (56%) | 13.10 | 0.97 |
> | Lower-Engagement Subset (44%) | 9.55 (min = 6) | 0.83 (min = 0.50) |
> ---
>
> To further address your concern about the reliability of the benchmark as an evaluation target, we re-ran the Full Conversation Simulation experiments separately for (i) the higher-engagement subset and (ii) the lower-engagement subset, and compared model rankings to the original full benchmark. The rankings are highly consistent across all three partitions. The table below reports Kendall’s τ between the original model ranking and the rankings within each engagement subset:
>
> | Metric | Kendall’s Tau (Original vs. Higher-Engagement) | Kendall’s Tau (Original vs. Lower-Engagement) |
> |---------------------------|--------------------------------------------------|-------------------------------------------------|
> | Semantic Similarity | 0.83 | 1.00 |
> | Stance Difference | 0.50 | 0.54 |
> | Length difference | 1.00 | 1.00 |
> | Absolute length difference| 1.00 | 1.00 |
> | ROUGE-L | 0.87 | 0.83 |
> ---
>
> These high τ values indicate that model ordering is stable whether one evaluates on (a) the full benchmark, (b) only more talkative participants (higher-engagement), or (c) only less talkative participants (lower-engagement). In other words, the DEBATE benchmark yields the same conclusions across different engagement levels. In sum, we believe the DEBATE benchmark is suitable for evaluating RPLA’s ability to role-play both lower-engagement participants and higher-engagement participants, both of which are naturally prevalent in real-world discussions.
>
> In the camera-ready version, we will (i) clarify the distribution of messages and on-topic rates across participants, and (ii) provide the “lower/higher-engagement” subsets so users can choose the subset that best fits their application. We are grateful for your thoughtful comments and suggestions and we hope these address your concerns about data quality.

---

> ### Author Response · Authors · 2025-11-29
> **Response to your follow-up comment - Part 2**
>
> > Regarding the discriminative power of the metrics used, I appreciate the additional analyses. It would be useful to report Kendall's Tau between the recovered rankings from different aggregation methods and bootstrapped runs - it seems that, at least in the table shown in "Response to Weakness 3 Part 2", there are some changes in the rankings between aggregation methods, when comparing to the original paper.
>
> Following your suggestion, we computed Kendall’s τ between the model rankings obtained from 1,000 bootstrap resamples and the original ranking reported in the paper (Full Conversation Simulation). Across metrics, the τ values are high, indicating that the ordering of models is stable under data resampling and that our results are not driven by any particular sample of groups:
>
> | Metric                     | Kendall Tau Median | Kendall Tau Std. Dev. |
> |---------------------------|---------------------|-------------------------|
> | Semantic Similarity       | 0.83                | 0.13                    |
> | Stance Difference         | 0.69                | 0.21                    |
> | Length difference         | 1.00                | 0.04                    |
> | Absolute Length difference| 1.00                | 0.06                    |
> | ROUGE-L                   | 1.00                | 0.07                    |
>
>
> In addition, we computed Kendall’s τ between the rankings induced by our original round-wise aggregation and those induced by the bipartite-matching aggregation described in “Response to Weakness 3 Part 2.” Kendall’s τ remains high (median = 0.55, standard deviation = 0.25) , which confirms that bipartite matching stretches the performance range and preserves the model ranking. Taken together, these analyses show that both sampling variability and aggregation choice preserve the model rankings. We hope these (both Part 1 and Part 2 of my responses) address your concerns about data quality.

---

### Official Review · Reviewer_ZqdW · 2025-10-30

**Soundness:** 2
**Presentation:** 3
**Contribution:** 1
**Rating:** 2
**Confidence:** 3

**Summary:**

The paper presents DEBATE, a large-scale empirical benchmark for evaluating opinion dynamics in multi-agent role-playing language models. The motivation is that existing work lacks human-grounded benchmarks for studying how LLM agents evolve opinions through social interaction.

The authors collect 37,357 messages from 2,792 U.S. participants engaging in multi-round, multi-party discussions on 107 controversial topics. The dataset includes both public messages and private beliefs, enabling evaluation of model-human alignment at the utterance, individual, and group levels.

Multiple LLMs are tested under three simulation settings, with quantitative metrics measuring semantic similarity, stance alignment, message length, and topical relevance.

Results show that current RPLAs reproduce some human-like behaviors at the utterance level but diverge at deeper cognitive and social levels. Simulated groups display stronger opinion convergence, positive belief drift, and more systematic individual shifts.

The authors position DEBATE as the first benchmark enabling systematic and multi-level evaluation of simulated opinion dynamics, aiming to support future research on aligning multi-agent LLMs with realistic human social behavior.

**Strengths:**

The paper addresses a well-motivated and underexplored problem that existing RPLA simulations often display unnatural group behavior, such as premature consensus, and lack a benchmark to measure how human-like their opinion dynamics are.

1. The data collection is the paper’s strongest contribution. The authors conduct tightly controlled multi-party, multi-round human discussions that capture both public messages and private beliefs, yielding over 37K utterances from about 2,800 U.S. participants across 107 topics. The inclusion of private self-reports alongside public statements adds clear value and provides a solid foundation for studying social alignment in RPLAs.

2. The metric design is another strength. The paper defines quantitative measures of semantic similarity, stance alignment, and opinion convergence across utterance-, individual-, and group-level evaluations, offering a comprehensive view of both linguistic and behavioral fidelity.

3. A diverse set of LLM families and sizes are compared, revealing consistent behavioral gaps: stronger opinion convergence, positive belief drift, and more systematic individual shifts. Although these patterns align with prior intuition, the paper verifies and quantifies them empirically, establishing a credible and reproducible baseline for future research.

**Weaknesses:**

1. The dataset is based on controlled four-person discussions with enforced turn-taking. While this setup ensures structured and clean data, it limits the natural flow of interaction and may not reflect opinion evolution in open or large-scale social settings.

2. The three simulation modes—Next Message Prediction, Tweet-guided Simulation, and Full Conversation Simulation—lack clear theoretical separation. Clarifying the motivation and analytical purpose of each mode would make the framework more convincing.

3. The paper highlights the importance of private self-reported beliefs for modeling realistic behavior. However, if these beliefs are only inserted into prompts, their actual influence on generated content is neither quantified nor discussed. Showing concrete examples or quantitative evidence of how private beliefs affect agent behavior would make this claim stronger.

4. In practice, the benchmark measures how human-like a specific LLM behaves during opinion exchange rather than assessing whether an RPLA simulation as a whole resembles human interaction. Its scope is therefore narrow, and the paper does not explore cases where different roles are played by distinct LLMs.

**Questions:**

1. Could the authors clarify the motivation behind enforcing turn-taking and explain how this design choice contributes to the study’s objectives?

2. Could the authors clarify whether the benchmark evaluates an LLM’s ability to exhibit human-like opinion dynamics or the overall human-likeness of a multi-agent opinion-exchange simulation?

---

> ### Author Response · Authors · 2025-11-24
> **Response to Weakness 1 and Question 1**
>
> > weakness 1: "The dataset is based on controlled four-person discussions with enforced turn-taking. While this setup ensures structured and clean data, it limits the natural flow of interaction and may not reflect opinion evolution in open or large-scale social settings."
>
> > question 1: "Could the authors clarify the motivation behind enforcing turn-taking and explain how this design choice contributes to the study’s objectives?"
>
> Thank you for offering to use the change to elaborate the objective of the study. Our experiment design solicits complete trajectories of both public and private opinions for every participant and round. In large-scale, fully open-world data like social media, participation is extremely skewed: roughly 90% of users mostly only read content (producing only about 1% of content in total), about 9% contribute occasionally, and only ~1% of “superusers” generate most posts [1]. Consequently open social-media data systematically overrepresent minorities and underrepresent the “silent majority,” making it difficult to evaluate whether an LLM-based simulation captures opinion evolution for typical individuals rather than just heavy posters.
>
> By contrast, the four-person group in DEBATE is structured so that every subject must generate a tweet and engage in dyadic discussion in each round; in our data, 100% of participants express their opinion at least once in every experimental phase. This yields fully observed trajectories for both public messages and private self-reports, which are crucial for quantitatively comparing human and RPLA opinion dynamics.
>
> As noted in our response to Kfex, the enforced turn-taking also aligns with classic opinion-dynamics models (e.g., DeGroot model and bounded-confidence models), which assume discrete time steps in which every agent updates their opinion and the analyst observes the full opinion profile over time [2, 3, 4]. Our round-based design with required contributions from each participant implements this “full observability” assumption in a realistic conversational setting. Real-world deliberative-democracy experiments (e.g., deliberative polling) similarly rely on small, structured discussion groups to promote equal speaking opportunities. [5] Our design inherits the small-group scale and achieves near-equal participation via the structured experimental design itself.
>
> At the same time, the scale of four-person group discussions are realistic: studies of real-world conversations in natural settings report an upper limit of four participants for a coherent conversation, with groups of five or more tending to fission into smaller subgroups [6]. Thus, our setting is aligned with small group chat and aligned with the assumptions of classic opinion-dynamics theory.
>
> Thanks to your suggestions, in the camera-ready version, we will clarify these trade-offs in the paper, and frame DEBATE as a deliberative small-group benchmark that complements, rather than replaces, open large-scale social-media corpora. We will also note that extending our RPLA evaluation framework to less structured, large-N environments is an important direction for future work.
>
> **References**
>
> [1] Van Mierlo, T. (2014). The 1% rule in four digital health social networks: an observational study. Journal of medical Internet research, 16(2), e2966.
>
> [2] Flache, A., Mäs, M., Feliciani, T., Chattoe-Brown, E., Deffuant, G., Huet, S., & Lorenz, J. (2017). Models of social influence: Towards the next frontiers. The journal of artificial societies and social simulation, 20(4), 2.
>
> [3] DeGroot, M. H. (1974). Reaching a consensus. Journal of the American Statistical association, 69(345), 118-121.
>
> [4] Hegselmann, R. (2015). Opinion dynamics and bounded confidence: models, analysis and simulation. The Journal of Artificial Societies and Social Simulation.
>
> [5] Fishkin, J. S., Luskin, R. C., & Jowell, R. (2000). Deliberative polling and public consultation. Parliamentary affairs, 53(4).
>
> [6] Krems, J. A., Dunbar, R. I., & Neuberg, S. L. (2016). Something to talk about: are conversation sizes constrained by mental modeling abilities?. Evolution and Human Behavior, 37(6), 423-428.

---

> ### Author Response · Authors · 2025-11-24
> **Response to weakness 2**
>
> > Weakness 2: "The three simulation modes—Next Message Prediction, Tweet-guided Simulation, and Full Conversation Simulation—lack clear theoretical separation. Clarifying the motivation and analytical purpose of each mode would make the framework more convincing"
>
> Thank you for pointing this out and offering us a chance to elaborate. Our three simulation modes connect to three different research literatures, illustrating how DEBATE can be used for each. Mode 1 (Next Message Prediction) is directly analogous to the “next-message prediction” task in multi-turn dialog modeling [1]. Mode 2 (Tweet-guided Conversation Simulation) is analogous to digital-clone frameworks that reconstruct a real social-media network from historical data and seed simulations with actual recorded posts, then use LLMs to simulate how information changes under that fixed network and seeded post trace [2]. Mode 3 (Full Conversation Simulation) is a classic agent-based opinion-dynamics setting useful for both classic and LLM agent-based simulation modeling [3,4]
>
> The specific analyses reported in the paper further illustrate how comparison across the three modes aids in uncovering the source of alignment or misalignment between human behavior and RPAA behavior. For example, when we compare the result table (Table 4) between Simulation Mode 1 and Mode 3, the performance gap between Modes 1 and 3 allows us to quantify how much human-agent alignment is lost when real conversation history is removed. On the other hand, the small difference in alignment between Modes 2 and 3 suggests that preserving the human-generated tweet trace alone does not substantially improve alignment. In the camera-ready version, we will clarify both the motivations for each simulation mode and the empirical insights gained from comparing them.
>
>
> **References**
>
> [1] Welch, C., Pérez-Rosas, V., Kummerfeld, J. K., & Mihalcea, R. (2019). Learning from personal longitudinal dialog data. IEEE Intelligent systems, 34(4), 16-23.
>
> [2] Puri, P., Hassler, G., Katragadda, S., & Shenk, A. (2024). Digital cloning of online social networks for language-sensitive agent-based modeling of misinformation spread. Plos one, 19(6), e030
>
> [3] Chuang, Y. S., Goyal, A., Harlalka, N., Suresh, S., Hawkins, R., Yang, S., ... & Rogers, T. (2024). Simulating opinion dynamics with networks of llm-based agents. In Findings of the association for computational linguistics: NAACL 2024 (pp. 3326-3346).
>
> [4] Taubenfeld, A., Dover, Y., Reichart, R., & Goldstein, A. (2024). Systematic Biases in LLM Simulations of Debates. In Proceedings of the 2024 Conference on Empirical Methods in Natural Language Processing (pp. 251-267).

---

> ### Author Response · Authors · 2025-11-24
> **Response to Weakness 3 - Part 1**
>
> > Weakness 3: "The paper highlights the importance of private self-reported beliefs for modeling realistic behavior. However, if these beliefs are only inserted into prompts, their actual influence on generated content is neither quantified nor discussed. Showing concrete examples or quantitative evidence of how private beliefs affect agent behavior would make this claim stronger."
>
> We agree that it is important to make the influence of private self-reported beliefs more explicit. Understanding this influence was the key goal of the ablation study reported in Section 5 Line 413-415 and Appendix L (“No Initial Opinion” in Tables 13–14). Specifically, in Simulation Mode 3 (Full Conversation Simulation; Table 13), removing the participant’s private Likert-scale belief from the RPLA memory reliably worsens stance alignment. For Depth topics, the average stance difference increases from 1.30 to 1.38; for Breadth topics, from 1.22 to 1.27 (both differences statistically significant, p < .01). This result provides evidence that including privately reported beliefs in the prompt helps to align RPLA-generated messages with human opinion trajectories. Due to space constraints we only reported the result in the appendix, but as the reviewer has highlighted the importance of this question, we will move a summary of these findings into the main paper.

---

> > ### Author Response · Authors · 2025-11-24
> > **Response to Weakness 3 - Part 2**
> >
> > [continued Response to Weakness 3 - Part 1]
> >
> > Thanks to your suggestion, we have also added concrete examples to illustrate how private beliefs shape agent behavior. For the topic “A democratic system where citizens, not elected officials, vote directly on major national issues to decide what becomes law is a good way of governing the US”, we compare the original RPLA (with private initial opinion) to the ablated version (“No Initial Opinion”) below:
> >
> > ---
> > | Example | Human (initial -> final private opinion) | RPLA (with private initial opinion) | RPLA (No Initial Opinion: private initial belief removed) |
> > |---------|-------------------------------------------|-------------------------------------|------------------------------------------------------------|
> > | 1 | **Initial:** “I think citizens voting directly on issues that become laws is great. This is because the laws affect them directly.”   (Stance: +2.5, Certainly agree)      **Final:** “They agreed with the topic and we both supported it.”   (Stance: +2.5, Certainly agree) | **Final RPLA (with belief):** “I still agree that a democratic system where citizens vote directly on major national issues is a good way of governing the US. This approach empowers individuals and ensures that laws reflect the needs of the people, fostering accountability and engagement in our democracy. However, it is essential to address challenges like misinformation and ensure that mechanisms are in place to protect minority rights and facilitate informed decision-making.”   (Stance: +2.5, Certainly agree) | **Final RPLA (No Initial Opinion):** “I disagree with the idea that a democratic system where citizens vote directly on major national issues is the best way of governing the US. While citizen engagement is important, elected officials provide the necessary expertise and guidance to navigate complex issues, ensuring informed decision-making. Direct voting can lead to impulsive choices influenced by emotions and misinformation, so a balanced approach that combines citizen input with representative governance is essential for a functioning democracy.”   (Stance: −2.5, Certainly disagree) |
> > | 2 | **Initial:** “I think it is a fairly good idea to let citizens vote on individual issues and laws. I feel that the citizens are more affected by these issues…”   (Stance: +2.5, Certainly agree)      **Final:** “I feel that this is a fairly good idea, to put the power back in the hands of the people. Being able to vote on laws which affect us would certainly help to shape our society and to form a better way of governing the United States. I feel that with the people in charge, we would see a lot less political corruption, as well.”   (Stance: +2.5, Certainly agree) | **Final RPLA (with belief):** “I agree that a democratic system where citizens vote directly on major national issues can be a good way to govern the US, as it empowers individuals and ensures that policies reflect the will of the people. However, I also believe that a balanced approach, incorporating the expertise of elected officials alongside direct citizen involvement, is essential to navigate complex issues and mitigate risks such as misinformation…”   (Stance: +1.5, Probably agree) | **Final RPLA (No Initial Opinion):** “I now lean towards agreeing with the statement that a democratic system where citizens vote directly on major national issues can be a good way of governing the US. Through my conversations, I’ve recognized the potential for direct democracy to empower individuals and ensure laws reflect the needs of the people. However, it's essential to address challenges like misinformation and ensure that citizens are well-informed…”   (Stance: +0.5, Lean agree) |
> > ---
> >
> >
> > (Participants selected from six stance labels: (−2.5) Certainly disagree, (−1.5) Probably disagree, (−0.5) Lean disagree, (+0.5) Lean agree, (+1.5) Probably agree, (+2.5) Certainly agree; the numeric values are used only for analysis.)
> >
> > Together, these quantitative ablations and qualitative examples support our claim that private self-reported beliefs do not merely appear in the prompt, but materially influence RPLA behavior, both at the aggregate level (systematically improved stance alignment in Tables 13–14) and at the level of individual opinion trajectories. We will clarify this in the main text and bring a compact version of the ablation results from the appendix to the main paper.

---

> ### Author Response · Authors · 2025-11-24
> **Response to weakness 4**
>
> > Weakness 4: "In practice, the benchmark measures how human-like a specific LLM behaves during opinion exchange rather than assessing whether an RPLA simulation as a whole resembles human interaction. Its scope is therefore narrow, and the paper does not explore cases where different roles are played by distinct LLMs."
>
> Thank you for raising this point and providing us the chance to elaborate. Our benchmark is intended to evaluate RPLA simulations, not a single LLM prompt. Specifically, each human participant in DEBATE is mapped to a distinct role-playing agent a_i, with its own memory state M_ai​,k​ containing demographics, initial opinions, personal tweet history, and conversational history (Sec. 4.1; Appendix F, Table 7). At each turn, we query the backbone model separately for each agent, using only that agent’s local memory and observations (Sec. 4.2; Table 3), so the multi-agent simulation emerges from the interaction of multiple RPLAs with distinct contexts, rather than from a single prompt that writes all four sides of the debate. In other words, different roles are played by distinct LLMs.
>
> Our evaluation metrics are defined over opinion trajectories to assess whether the multi-agent RPLA simulation as a whole resembles human interaction in groups. In addition to utterance-level alignment (Sec. 5), Sec. 6 explicitly analyzes group- and individual-level opinion dynamics (e.g., belief-change patterns across rounds and group-level convergence; Sec. 4.3, Sec. 6; Table 4, Fig. 2–3), which capture properties of the overall simulation rather than just individual utterances. We will clarify this evaluation hierarchy in the revised paper so that the distinction between per-utterance behavior and emergent multi-agent dynamics is more explicit.
>
> To further support that we are evaluating distinct role-playing RPLA rather than a generic LLM, we also ran an additional ablation experiment where we removed all distinct per-agent information from the RPLA memory (demographics, initial opinions, initial tweet, and personalized history) and simply asked the model to “role-play a person” in each agent. On Depth topics under Simulation Mode 3 (Full Conversation Simulation) with gpt-4o-mini-2024-07-18, both semantic similarity and stance alignment significantly worsened (p < .01), as summarized below:
>
> ---
> | Condition | average semantic similarity | average stance difference |
> |--------------------------------------------------|-----------------------------|----------------------------|
> | Pre-ablation (Table 4) | 0.41± 0.01 | 1.22± 0.03 |
> | Remove all distinct context from each RPLA agent | 0.38 | 1.53 |
> ---
>
>
> This degradation in human-RPLA alignment when distinct role-playing information is removed indicates that assigning distinct digital-twin prompts to each individual RPLA is crucial for capturing human-like behavior at the individual level. We will add this ablation and its interpretation to the camera-ready version, and will clarify any ambiguous language making it unclear that different roles are simulated by distinct LLMs.

---

> ### Author Response · Authors · 2025-11-24
> **Response to Question 2**
>
> > Question 2: "Could the authors clarify whether the benchmark evaluates an LLM’s ability to exhibit human-like opinion dynamics or the overall human-likeness of a multi-agent opinion-exchange simulation?"
>
> Thank you for this question. DEBATE is explicitly designed to evaluate both (i) how human-like an LLM behaves when instantiated as a role-playing agent (with distinct, individualized role-playing context per agent), and (ii) the overall human-likeness of the resulting multi-agent opinion-exchange simulation.
>
> Concretely, each human participant is mapped to a distinct RPLA with its own memory state (Sec. 4.1; Appendix F, Table 7), and at each turn we query the LLM separately for each agent using only that agent’s local memory and observations (Sec. 4.2; Table 3). Our evaluation then operates at multiple levels: Sec. 5 focuses on utterance-level alignment (how well each RPLA tracks its human counterpart), while Sec. 6 analyzes group- and individual-level opinion dynamics (e.g., belief-change trajectories, convergence patterns, and partner influence), which capture the human-likeness of the multi-agent simulation as a whole.
>
> We will clarify this evaluation hierarchy in the paper so that the distinction between per-agent behavior and emergent group dynamics is more explicit.

---

### Official Review · Reviewer_Kfex · 2025-10-31

**Soundness:** 3
**Presentation:** 3
**Contribution:** 3
**Rating:** 6
**Confidence:** 3

**Summary:**

This paper introduces DEBATE, a new large-scale benchmark for evaluating the realism of role-playing LLM agents (RPLAs) in opinion dynamics simulations. The authors argue that existing RPLA simulations often show unnatural behaviors (like premature consensus) and lack a solid baseline against real human interactions.

To fix this, the authors built the DEBATE dataset. They collected data from 2,792 U.S. participants discussing 107 controversial topics. A core contribution is that the dataset captures both the public messages (tweets, chats) and the private, reported beliefs of the participants (before and after the chats). Using this benchmark, the paper tests several LLMs and finds significant gaps between AI and human behavior:
- RPLA groups converge on opinions far more strongly than humans.
- RPLAs show a systematic "belief drift," sometimes towards false beliefs.
- Simple supervised fine-tuning (SFT) improves surface-level metrics (like message length) but actually harms deeper alignment (like stance and semantic similarity).

**Strengths:**

- Important and Novel Problem: The paper tackles a timely and critical question. As LLMs are increasingly used to simulate social interactions, we urgently need rigorous ways to check if these simulations are socially realistic. DEBATE is, to my knowledge, the first large-scale empirical benchmark specifically for multi-agent opinion dynamics.

- Data Collection: The study's design is its biggest strength. Separating 'public speech' from 'private belief' is a major contribution. It allows evaluation to move beyond simple text mimicry to the core of opinion dynamics: the gap between what people say and what they actually think.

- Well-Constructed Benchmark. The benchmark itself is solid. The dataset is large and diverse (see Appendix E). The topic selection (mixing 'Depth' topics with ground truths and 'Breadth' topics) is smart. The three simulation modes, which test agents with decreasing levels of human context, provide a good way to measure agent autonomy.

- Clear and Impactful Findings. The paper's conclusions are clear and important for the field. The finding that RPLAs over-converge (Fig 2d, 3b) and that SFT fails (or even hurts) deep alignment (Appendix N) are major takeaways. This suggests that simple imitation is the wrong approach for this problem.

**Weaknesses:**

- Lack of Deeper Discussion on SFT Failure: The SFT results (Appendix N) are fascinating but underexplored. The paper states that naive SFT fails, but doesn't deeply explore why. Is it because the model learns to imitate an "average" human, losing individual diversity? Is the next-token prediction objective simply wrong for modeling a latent process like belief update? The paper would be stronger if it discussed alternative training objectives (e.g., RL based on realism, or explicit belief-tracking models).

- Limits of LLM-as-Evaluator: The evaluation relies on gpt-4o-mini to classify topic relevance and stance (Appendix G). While the authors report 90% accuracy on a validation set, this creates an "LLM evaluating an LLM" loop, which can be risky. The authors should have reported the Inter-Annotator Agreement (IAA) among their human labelers. This would tell us how subjective or difficult the labeling task is in the first place.

- Scope limits: The sample is US-only and dyadic; generalization to other cultures/platform structures remains uncertain.

**Questions:**

see weakness

---

> ### Author Response · Authors · 2025-11-24
> **Response to Weakness 1 - Part 1**
>
> > Weakness 1: *Lack of Deeper Discussion on SFT Failure: The SFT results (Appendix N) are fascinating but underexplored. The paper states that naive SFT fails, but doesn't deeply explore why. Is it because the model learns to imitate an "average" human, losing individual diversity? Is the next-token prediction objective simply wrong for modeling a latent process like belief update? The paper would be stronger if it discussed alternative training objectives (e.g., RL based on realism, or explicit belief-tracking models).*
>
> Thank you for highlighting this and for suggesting hypotheses about why naive SFT might fail. We have added our analysis and included both quantitative and qualitative analysis.
>
> **(1) Does SFT collapse agents toward an “average” opinion, reducing diversity?**
>
>  Motivated by your suggestion, we directly tested whether SFT causes RPLAs to imitate an “average” human and lose semantic diversity. If this were the case, we would expect higher pairwise semantic similarity between LLM messages (i.e., reduced diversity) after SFT. To quantify this, we estimated semantic diversity by repeatedly sampling pairs of LLM messages (from the same topic but different simulations), computing their embedding-based semantic similarity, and averaging over 1000 samples. We did this for pre-SFT and post-SFT models across modes and three generalization types (group/round/topic).
>
> ---
> | Generalization Type | version | Model | Mean similarity between LLM messages |
> |---------------------|----------------------------|-----------|--------------------------------------|
> | group | Next Message Prediction | Pre-SFT | 0.60 |
> | | | Post-SFT | 0.30 |
> | | Tweet-Guided Conversation Simulation | Pre-SFT | 0.61 |
> | | | Post-SFT | 0.30 |
> | | Full Conversation Simulation| Pre-SFT | 0.62 |
> | | | Post-SFT | 0.29 |
> | round | Next Message Prediction | Pre-SFT | 0.61 |
> | | | Post-SFT | 0.33 |
> | | Tweet-Guided Conversation Simulation | Pre-SFT | 0.61 |
> | | | Post-SFT | 0.32 |
> | | Full Conversation Simulation| Pre-SFT | 0.62 |
> | | | Post-SFT | 0.31 |
> | topic | Next Message Prediction | Pre-SFT | 0.60 |
> | | | Post-SFT | 0.32 |
> | | Tweet-Guided Conversation Simulation | Pre-SFT | 0.60 |
> | | | Post-SFT | 0.33 |
> | | Full Conversation Simulation| Pre-SFT | 0.61 |
> | | | Post-SFT | 0.31 |
> ---
>
> Contrary to the “mode collapse” hypothesis, semantic similarity “decreases” after SFT (e.g., from 0.60–0.62 to 0.29–0.33 for gpt-4o-mini-2024-07-18 on Depth topics across simulation modes), indicating that SFT actually increases semantic diversity across simulated agents. This suggests that the failure of SFT to improve deeper alignment is not because the model collapses to a single “average” persona.

---

> ### Author Response · Authors · 2025-11-24
> **Response to Weakness 1 - Part 2**
>
> [continued Response to Weakness 1 - Part 1]
>
> **(2) Qualitative evidence: shorter, less informative post-SFT responses.**
>
> Instead, our qualitative analysis points to a different issue: SFT encourages shorter, less informative messages. As shown in Tables 15–16, SFT reduces length differences and brings message length closer to human averages, but this often comes at the cost of content richness, which improves surface-level metrics (e.g., response length) but degrade deeper semantic and stance alignment. The example below (Depth topic “A body cleanse, in which you only consume particular nutrients over 1–3 days, is beneficial for you”) illustrates this:
>
> ---
> | Human message                                                                                                                                                                                                                                                                                  | RPLA message (pre-SFT)                                                                                                                                                                                                                                                                                                                                                                                                                                                                                                                                      | RPLA message (post-SFT)                                                                                                                                                                              |
> |------------------------------------------------------------------------------------------------------------------------------------------------------------------------------------------------------------------------------------------------------------------------------------------------|-------------------------------------------------------------------------------------------------------------------------------------------------------------------------------------------------------------------------------------------------------------------------------------------------------------------------------------------------------------------------------------------------------------------------------------------------------------------------------------------------------------------------------------------------------------|-------------------------------------------------------------------------------------------------------------------------------------------------------------------------------------------------------|
> | “I might have changed a little bit towards leaning agreeing. A couple of my partners made some good points. At the end though, we all agreed it is not really realistic to think you could do this in only 1–3 days.”   (Stance = Probably disagree (−1.5), word count = 42)                     | “I still lean towards disagreeing with the idea that a ‘body cleanse,’ which involves consuming only particular kinds of nutrients over 1–3 days, effectively helps eliminate toxins. While I understand that some people may find it refreshing and it might serve as a temporary reset, I believe our bodies already have natural mechanisms for detoxification. Relying on quick fixes can create unrealistic expectations, and it's more beneficial to focus on sustainable healthy eating habits and a balanced lifestyle in the long run.”   (Stance = Lean disagree (−0.5), word count = 84, stance diff = 1, semantic similarity = 0.42) | “I agree with that.”   (Stance = Certainly agree (+2.5), word count = 4, stance diff = 4, semantic similarity = 0.22)                                                                 |
> | “You're probably right. Getting toxins from the body using this natural way in three days sounds very ambitious and unrealistic. Very few if any. Only when one makes it a habit to include detoxifying nutrients in the diets they are able to see results in the long run.” (Stance = Probably agree (+1.5), word count = 49) | “I completely agree with you, 5dec8! It's important to view a cleanse as a temporary measure rather than a fix for ongoing poor eating habits. Focusing on long-term, sustainable changes is definitely the way to go. As for strategies, I try to meal prep to make healthier choices more convenient during the week. I also make a point to incorporate more fruits and vegetables into my meals. What about you? What strategies have you found effective in maintaining a balanced lifestyle?” (Stance = Certainly agree (+2.5), word count = 83, stance diff = 1, semantic similarity = 0.41) | I respect your opinion and I think there are many ways to arrive at the same destination of long-term health. (Stance = Lean agree (+0.5), word count = 21, stance diff = 1, semantic similarity = 0.14) |
> ---

---

> ### Author Response · Authors · 2025-11-24
> **Response to Weakness 2**
>
> > Weakness 2: "Limits of LLM-as-Evaluator: The evaluation relies on gpt-4o-mini to classify topic relevance and stance (Appendix G). While the authors report 90% accuracy on a validation set, this creates an "LLM evaluating an LLM" loop, which can be risky. The authors should have reported the Inter-Annotator Agreement (IAA) among their human labelers. This would tell us how subjective or difficult the labeling task is in the first place."
>
> We agree that it is important to characterize the subjectivity of the labeling task. We have now computed human–human inter-annotator agreement on 400 randomly sampled messages spanning all topics and both human and LLM outputs (including tweets, initial opinions, final opinions, and conversation turns). For topic relevance (binary), two human annotators achieved 96.8% agreement and Cohen’s κ = 0.89. For stance (6-way ordinal classification), they achieved Cohen’s κ = 0.81. We report Cohen’s κ for stance because it is a standard inter-rater reliability measure for categorical/ordinal labels in multi-class settings [1, 2]. Under commonly used guidelines [1, 2], Cohen’s κ values ≥ 0.81 indicate “almost perfect agreement,” suggesting that the labeling scheme is well-defined and that our reported ~90% LLM–human agreement reflects a reasonably reliable automatic judge.
>
> **References**
>
> [1] Viera, A. J., & Garrett, J. M. (2005). Understanding interobserver agreement: the kappa statistic. Family medicine, 37(5), 360-363.
>
> [2] McHugh, M. L. (2012). Interrater reliability: the kappa statistic. Biochemia medica, 22(3), 276-282.

---

> ### Author Response · Authors · 2025-11-24
> **Response to Weakness 3**
>
> > Weakness 3: “Scope limits: The sample is US-only and dyadic; generalization to other cultures/platform structures remains uncertain.”
>
> The current release is indeed scoped to US adults; we will make this scope explicit in the paper. While not fully generalizable outside this scope, DEBATE nevertheless addresses several other limitations of existing opinion-dynamics corpora/datasets: we provide multi-round, multi-party interactions, both public (tweets) and private (self-reports) opinions, fully traceable context for each utterance, and rich demographics. In this sense, DEBATE complements prior influential benchmarks that likewise are restricted to individual countries (as detailed in Related Work) such as UK parliamentary question periods (British MPs only) or US Oxford-style Intelligence Squared debates (US events only). These datasets have proven useful despite being limited to a single country, but lack additional features DEBATE supports, e.g., private beliefs, demographic information.
>
> Within its US scope, the participant pool is broad and diverse, spanning a wide range of ages (18–83, M = 39.5, SD = 13.0), genders (50.2% male, 49.0% female), ethnicities (e.g., 66.4% White, 24.7% Black, 5.5% Asian, 5.1% Hispanic), educational backgrounds (from high school to doctoral degrees), and income levels (from under 25k USD to over 200k USD), and reporting diverse occupations (e.g., finance, engineering, healthcare, manufacturing). This diversity provides a foundation for modeling opinion dynamics across various sociocultural backgrounds within the US. We will clarify this and explicitly flag broader cross-country/cultural extension as future work.
>
> The “dyadic” structure of interactions was motivated by two goals that we will clarify in the introduction. First, such interactions resemble how multi-party conversation typically unfolds: previous studies show that, once a conversation involves four or more people, it often splits into parallel conversations between dyads [1]. Second, our design aligns with literature on classic agent-based modeling of opinion dynamics, where conversational episodes and belief updates are usually specified as pairwise interactions [2] (e.g., DeGroot model [3] and bounded-confidence models [4]). Thus rotating dyads within a four-person group strikes a balance between small-group discussions in the real world and the assumptions of agent-based models of opinion dynamics, making it amenable to analysis via many classic and contemporary model types. We will clarify the motivation and also mention the extension beyond dyadic settings as future work.
>
> At the same time, we view DEBATE as complementary to large-scale, unstructured social-media corpora, which already provide many examples of non-dyadic, loosely structured interaction. Those open-world datasets are valuable for scale and ecological richness, but they typically suffer from extreme participation skew: 90% of users mostly only read content (producing only about 1% of content in total), about 9% contribute occasionally, and only ~1% of “superusers” generate most posts [5] and lack full trajectories of both public and private opinions. By contrast, DEBATE’s structured four-person design focuses discussion on a specific topic, enforces turn-taking so that each participant contributes in every round (yielding 100% participation across experimental phases), and interleaves public messages with private belief reports. This structure gives us complete, per-person opinion trajectories over time, which are crucial for quantitatively benchmarking RPLA simulations against human opinion dynamics.
>
> **References**
>
> [1] Egbert, M. M. (1997). Schisming: The collaborative transformation from a single conversation to multiple conversations. Research on Language and Social Interaction, 30(1), 1-51.
>
> [2] Flache, A., Mäs, M., Feliciani, T., Chattoe-Brown, E., Deffuant, G., Huet, S., & Lorenz, J. (2017). Models of social influence: Towards the next frontiers. The journal of artificial societies and social simulation, 20(4), 2.
>
> [3] DeGroot, M. H. (1974). Reaching a consensus. Journal of the American Statistical association, 69(345), 118-121.
>
> [4] Hegselmann, R. (2015). Opinion dynamics and bounded confidence: models, analysis and simulation. The Journal of Artificial Societies and Social Simulation.
>
> [5] Van Mierlo, T. (2014). The 1% rule in four digital health social networks: an observational study. Journal of medical Internet research, 16(2), e2966.

---

> ### Author Response · Authors · 2025-11-24
> **Response to Weakness 1 - Part 3**
>
> [Continued Response to Weakness 1 -Part 2]
>
> **(3) Why token-/sequence-level SFT may be mismatched to opinion-dynamics objectives.**
>
> Conceptually, DEBATE is designed to evaluate alignment at three levels: utterance-level alignment (Sec. 5), individual-level opinion updates, and group-level dynamics such as convergence and belief drift (Sec. 6). Standard post-training methods like SFT, DPO, GRPO, etc., optimize token- or sequence-level likelihoods conditioned on the preceding context. They do not explicitly target whether (i) the pattern of opinion change across rounds is human-like, (ii) the sensitivity to partner influence matches humans, or (iii) the evolution of group-level opinion diversity aligns with human groups. Our results suggest that simply fitting the next-token distribution does not guarantee alignment on these higher-level opinion-dynamics properties.
>
> Motivated by your suggestion, below are the potential alternative training objectives we will discuss. First, one could define RL-style rewards based on our opinion-dynamics metrics as signals (e.g.,  rewarding simulations whose belief-change patterns match human groups). Second, one can also add an auxiliary head to predict latent belief trajectories (e.g., stance) and message generation is regularized to remain consistent with those beliefs (e.g., by penalizing discrepancies between predicted and expressed stance over rounds). We see these RL-on-realism and belief-tracking directions as natural ways to move beyond purely token-level training, and DEBATE is explicitly designed to support this kind of work.
>
> In the camera-ready version, we will clarify that our SFT experiments are intended as an illustrative use of DEBATE for post-training, not as a final solution to human–RPLA misalignment, and we will explicitly highlight RL-style realism objectives and belief-tracking–based training as concrete future directions that leverage DEBATE’s multi-level opinion-dynamics signals (utterance, individual, and group), moving beyond relying solely on token-level training objectives.

---

### Author Response · Authors · 2025-12-03
**Summary for AC: Merits, Concerns, and What We Changed - Part 1**

Below is a concise summary of (i) the merits of DEBATE (as acknowledged by the reviewers) and (ii) how our responses address their main concerns. We have also updated the PDF and highlighted changes in orange.

# Contribution, significance, and strengths
- First benchmark for human-like opinion dynamics in multi-agent role-playing LLM agents (RPLAs). All three reviewers note DEBATE fills a key gap: prior RPLA work lacks a human-grounded benchmark for opinion trajectories (R1: Kfex; R2: ZqdW; R3: K4Xk).

- Structured dataset and evaluation framework at scale. Multi-turn, multi-party groups with rotating dyads; both publicly expressed messages and privately reported beliefs; rich demographics. Scale: 2,792 participants, 725 fully completed groups, 37,357 utterances across 107 topics. Metrics span utterance, individual, and group levels; reviewers describe the design as careful with human validation where needed.

- Empirical findings that isolate failure modes. Versus humans, RPLA groups over-converge and exhibit positive belief drift; adding more context or vanilla SFT does not fix deeper semantic/stance alignment.

# Reviewers’ concerns and how we addressed them
(1) Data quality & benchmark reliability (R3 + follow-up)

- Concern: Possible “low-engagement” crowdworkers.
- Response: All participants completed the full experiment and contributed each phase. We split participants into higher-engagement (56%) vs. lower-engagement (44%): 13.10 vs. 9.55 messages on average (min 6), on-topic 0.97 vs. 0.83. Re-running Full Conversation Simulation on both subsets yields stable model rankings vs. the original as measured by Kendall’s τ. Across 1,000 bootstrap resamples, the model rankings also remain stable (Appendix R). In sum, conclusions are robust to engagement level and sampling variability.

(2) Metric discriminative power & robustness (R3 + follow-up)
- Concern: Scores look close; can DEBATE distinguish models?
- Response: Statistical tests show reliable differences. Alternative aggregation (maximum-weight bipartite matching) increases spread (semantic similarity range 0.03→0.07) while maintaining ordering consistency (Kendall’s τ ≈ 0.55 between aggregation methods). A permutation baseline shows clear separation from noise (e.g., 0.41→0.28 similarity; 1.30→1.99 stance diff). (Appendix S)
- Interpretation: High agreement across aggregation schemes + strong permutation gaps support discriminative power.

(3) LLM-as-evaluator and subjectivity (R1)
- Concern: Stance/relevance classification seems subjective and requires human inter-rater agreement (IAA).
- Response: Two annotators labeled 400 messages: topic relevance 96.8% agreement, κ=0.89; stance (6-way) κ=0.81 (“almost perfect”). LLM–human agreement (~90%) thus sits on a low-subjectivity task. (Appendix G)

(4) Scope limits: US-only, dyadic, small groups (R1, R2)
- Concern: Generalizability beyond US/dyadic; constrained flow due to enforced turn-taking.
- Response: DEBATE is a deliberative, structured benchmark complementing open social-media corpora. Design aligns with opinion-dynamics theory (pairwise updates, fully observed profiles per time step) and conversation science (four-person discussions tend to split into dyads). We clarify our scope, note broad US diversity, and flag cross-cultural/larger-network extensions.

(5) Role of private beliefs and individualized RPLA memory (R2, R3)
- Concern: R2 thought we didn’t quantify the effect of private beliefs and R3 thought we didn’t show the values of benchmark attributes.
- Response: The existing ablation study (line 403-417, Table 12, 13) “No Initial Opinion” condition worsens stance alignment (Depth & Breadth); “No Benchmark Attributes” (remove per-agent personalization) further reduces semantic similarity and worsens stance difference. Each human maps to a distinct RPLA with its own memory; we query the backbone per agent/turn and evaluate group-level dynamics over multi-agent trajectories.

(6) Multi-agent simulation vs. “single-LLM behavior” (R2)
- Concern: R2 thought the benchmark measures a single LLM rather than a multi-agent simulation.
- Response: Each human maps to a distinct RPLA with its own memory; we query the backbone model separately per agent and per turn. Group-level metrics (convergence, belief drift) are computed over these multi-agent trajectories. DEBATE evaluates multi-agent simulations, not a single prompt writing all sides.

(7) Why vanilla SFT underperforms (R1)
- Concern: Should explain why vanilla SFT underperforms on deep alignment.
- Response: SFT increases semantic diversity (rules out simple mode collapse) but yields shorter, less informative replies, which leads to better length matching yet worse stance/semantic alignment. Token/sequence-level SFT is mismatched to trajectory-level objectives. We outline RL-on-realism and belief-tracking as next steps. (Appendix M)

---

> ### Author Response · Authors · 2025-12-03
> **Summary for AC: Merits, Concerns, and What We Changed - Part 2**
>
> # Conclusion
> Reviewers agree DEBATE addresses an important gap with a novel, well-designed benchmark. Core concerns (data quality, discriminative power, evaluator reliability, scope, SFT) are addressed with new analyses that were not visible for back-and-forth due to platform limits. We believe DEBATE is a strong fit for the ICLR datasets/benchmarks track and foundational for empirically grounded multi-agent LLM opinion dynamics simulations.

---

### Meta-Review · Area_Chair_C5FK · 2026-01-16

**Summary:**

The paper provides a dataset that captures both public messages and opinion values of different topics from large number of participants. Then they evaluate LLMs and find AI and human behavior are not quite aligned. While such a contribution is extremely crucial from dataset release and benchmarking viewpoint, probably the paper is far from ready to be published at this stage. A key problem of the paper is
that they only consider a four person discussion. In rebuttal, the authors argued most of the users in social network are not really active. However, depending on the nature of interaction and cross edge interaction probabilities, some content can go viral (similar to an unstable system) and therefore, I agree with the Reviewer ZqdW that such settings may not reflect reality.  Hence, I recommend reject.

**Reviewer Concerns:**

The motivation behind "four person discussion" is not satisfactorily addressed.

**Reviewer Scores:**

Reviewer ZqdW might not have improved the scores-- I do not think their concern (as mentioned above) is properly addressed.

---

### Decision · Program_Chairs · 2026-01-26

Reject